# Diffusion-based deep learning method for augmenting ultrastructural imaging and volume electron microscopy

Chixiang Lu[1], Kai Chen[1,2], Heng Qiu[1], Xiaojun Chen[2], Gu Chen[1], Xiaojuan Qi ®[3] ✉ & Haibo Jiang ®[1] ✉

Electron microscopy (EM) revolutionized the way to visualize cellular ultra-structure. Volume EM (vEM) has further broadened its three-dimensional nanoscale imaging capacity. However, intrinsic trade-offs between imaging speed and quality of EM restrict the attainable imaging area and volume. Isotropic imaging with vEM for large biological volumes remains unachievable. Here, we developed EMDiffuse, a suite of algorithms designed to enhance EM and vEM capabilities, leveraging the cutting-edge image generation diffusion model. EMDiffuse generates realistic predictions with high resolution ultra-structural details and exhibits robust transferability by taking only one pair of images of 3 megapixels to fine-tune in denoising and super-resolution tasks. EMDiffuse also demonstrated proficiency in the isotropic vEM reconstruction task, generating isotropic volume even in the absence of isotropic training data. We demonstrated the robustness of EMDiffuse by generating isotropic volumes from seven public datasets obtained from different vEM techniques and instruments. The generated isotropic volume enables accurate three-dimensional nanoscale ultrastructure analysis. EMDiffuse also features self-assessment functionalities on predictions' reliability. We envision EMDiffuse to pave the way for investigations of the intricate subcellular nanoscale ultra-structure within large volumes of biological systems.

Electron microscopy (EM) is an essential tool for obtaining high-resolution images of biological specimens and has made a tremendous impact on cell biology, revealing highly complex cellular structures at the nanometer scale. Over the decades, EM has catalyzed numerous breakthroughs in life sciences, such as discovering novel cellular organelles, elucidating membrane structures, and generating detailed visualization of the macromolecular complexes[1–3]. Volume electron microscopy (vEM), a more recent innovation, arose from overcoming limitations of conventional two-dimensional electron microscopy techniques in studying 3D cellular structures[4–7]. vEM techniques such as serial section-based tomography techniques with transmission electron microscopy (TEM) and scanning electron microscopy (SEM),

serial block-face SEM (SBF-SEM), and focused ion beam SEM (FIB-SEM) have facilitated investigations of 3D structures in cells, tissues and even small model organisms at nanometer resolution. Particularly, recent efforts in mapping brain connectomics with serial section-based vEM techniques[8–12] and the developments of enhanced FIB-SEM for the acquisition of isotropic vEM data with improved robustness and throughput[13–15] have further reinvigorated the field of EM and its applications.

However, the potential of EM and vEM is hindered by intrinsic limitations. It is essential to achieve a sufficient signal-to-noise ratio (SNR) to accurately map nanoscale cellular structures, which will inevitably reduce the imaging speed and prolong the dwell time per

[1]Department of Chemistry, The University of Hong Kong, Hong Kong, China. [2]School of Molecular Sciences, The University of Western Australia, Perth, WA, Australia. [3]Department of Electrical and Electronic Engineering, The University of Hong Kong, Hong Kong, China. ✉e-mail: xjqi@eee.hku.hk; hbjiang@hku.hk

pixel, restricting the area and volume of imaging. Robust isotropic vEM also requires advanced instrumentation, such as enhanced FIB-SEM, which is only available at certain institutions or specialized centers. More importantly, the dogma of vEM is no viable method for generating isotropic datasets from large volumes. Currently, vEM is either limited to producing isotropic data from volumes of up to approximately 300 μm × 300 μm × 300 μm[16], using state-of-the-art enhanced FIB-SEM, or capturing larger volumes aiming for cubic millimeter scale through serial section-based techniques[10,17], which, however, suffers from unsatisfactory axial resolution due to the limited section thickness of diamond knife cuts.

In recent years, computational methods and deep learning techniques have been applied to address the technical limitations of microscopy methods to expedite imaging processes and enhance the image quality of both light microscopy[18–21] and electron microscopy[22–24]. Significant advances have been achieved in denoising and super-resolution tasks with deep learning techniques[23,25–28]. However, three major limitations still exist in the current methods. Firstly, most methods[20,21,23,26–29] use a regression-based deep learning model, which emphasizes the learning of low-frequency mode[30]. Additionally, these frameworks employ L1 or L2 training objectives to train the model towards approximating the minimum mean square error (MMSE) of possible true prediction distribution, i.e., the average/median of all possible data interpretations of the learned posterior. This results in excessively smooth predictions with reduced resolution – critical in microscopy – defined as the smallest discernible distance between two distinct points on a specimen. A few methods utilize CycleGAN-based approaches, which may be susceptible to unstable training processes and limited capability in handling geometric transformations[31]. Secondly, many current methods are designed for light microscopy[21,25–27], whereas EM images contain complex nanoscale structures, underexplored noise models, and single-channel limited image information, rendering image quality enhancement tasks more difficult. Third, few of the current methods can provide reliability assessments for their predictions. Since denoising and super-resolution are inherently ill-posed problems with multiple possible solutions, biologists may doubt the validity of the processed images without any uncertainty measures. CARE assesses aleatoric uncertainty by training networks to predict the mean and variance and assesses epistemic uncertainty by evaluating the agreement across five distinct networks, which introduce a significant increase in training overhead.

In this study, we propose EMDiffuse, a diffusion model-based package for EM applications, aiming to enhance EM ultrastructural imaging and expand the realm of vEM capabilities. Diffusion models have demonstrated superiority over regression-based models[32–34] and exhibit greater stability in training than GAN-based models[35] regarding image generation and restoration tasks due to their distinctive diffusion-based training and inference schemes. Diffusion models also generate outputs with high-resolution details, which is critical for imaging intricate nanoscale cellular structures with EM. Here, we adopted the diffusion model for EM applications and developed EMDiffuse-n for EM denoising, EMDiffuse-r for EM super-resolution, and vEMDiffuse-i and vEMDiffuse-a for generating isotropic resolution data from anisotropic volumes for vEM. Moreover, we demonstrate the self-assessment capability of EMDiffuse, taking advantage of the inherent ability of diffusion models to generate an arbitrary number of diverse samples. The code is available in https://github.com/Luchixiang/EMDiffuse/[36].

## Results

### EMDiffuse exhibited outstanding denoising performance
We developed EMDiffuse-n, the noise reduction pipeline of EMDiffuse, to denoise EM ultrastructural images and accelerate EM imaging (Fig. 1a). The EMDiffuse-n comprises three stages, which are data collection, image processing, and diffusion model (Fig. 1a, "Methods" section). The first stage involved obtaining EM training pairs with different acquisition times (Supplementary Fig. 1a, "Methods" section). Then, a hierarchical approach was employed to precisely align and register the noisy images with the reference image (Supplementary Fig. 1b, "Methods" section). Finally, a diffusion model, namely UDiM (Supplementary Fig. 2, "Methods" section), was trained to diminish the noise in well-aligned raw images. We observed extremely noisy training patterns lead to large gradients and subsequently cause drastic and undesired changes in model weights as well as instabilities during model training. To address this issue, we employed the difficulty-aware loss function (Supplementary Table 1, "Methods" section)[37], which downweights the contributions of such cases through a learned difficulty map, effectively mitigating the impact of these instances.

In the inference stage, given an input image, UDiM samples one plausible solution from the learned solution distribution at each test time. Thus, it can generate an unlimited number of plausible outputs. Multiple outputs can be sampled and ensembled to improve the predictions of EMDiffuse-n or analyzed individually since recovering the noise-free images is an ill-posed problem (Fig. 1a, Supplementary Fig. 3a, b, "Methods" section). Specifically, we optimized the prediction generation methods based on the assessment of FSIM[38] (Supplementary Fig. 3c) and prediction resolution (Supplementary Fig. 3d). We used K outputs (K = 2 in our experiments) for each raw input and computed the mean of these outputs, samples from the learned distribution, as the ultimate prediction ("Methods" section, Supplementary Fig. 3). Of note, we assessed the predictions with FSIM instead of PSNR due to the impact of random noise in ground truth EM images (Supplementary Fig. 4).

Next, a comprehensive examination was conducted to evaluate the performance of EMDiffuse-n and other denoising methods (Fig. 1b, c). We compared EMDiffuse-n with three widely adopted supervised denoising methods (i.e., CARE[28], RCAN[39], PSSR[23]) and two self-supervised methods (i.e., Noise2Noise[29] and Noise2Void[19]). EMDiffuse, as well as the baselines, were trained and tested on a mouse brain cortex dataset acquired in-house (Supplementary Information). EMDiffuse-n surpassed other denoising methods in generating images with intricate ultrastructural information. This allows for adequately distinguishing mitochondria, cellular vesicles, endoplasmic reticulum (ER), and proximal plasma membranes (Fig. 1b, Supplementary Fig. 5, and Supplementary Movie 1 and 2). Example line profiles (Fig. 1b) revealed that EMDiffuse-n successfully separated ultrastructure that is difficult to discriminate, such as mitochondria cristae and proximal plasma membrane, and produced results closest to the GT, while other denoising methods (i.e., CARE[28], RCAN[39], PSSR[23]) were incompetent in discerning these fine details. Of note, the Fourier power spectrum and resolution measurements[40] indicated that EMDiffuse-n generates outputs with high-resolution details (Fig. 1b). By contrast, other supervised and self-supervised deep learning models introduced smoothness[41] (white line across the center of power spectrum) as they approximated the MMSE of possible true prediction distribution, and partially lost intricate structural details (Fig. 1b and Supplementary Fig. 5). In terms of quantitative evaluations, compared to other denoising methods, EMDiffuse-n demonstrated superior performance across three evaluation metrics, including LPIPS[42], FSIM[38], and resolution ratio, that assess models' capability in generating accurate and high-resolution predictions (Fig. 1c).

EMDiffuse-n addressed the common concern on the reliability of deep learning-generated data for scientific data processing (Fig. 1d). EMDiffuse-n was designed with the capability to reflect and self-assess the reliability of its predictions by calculating the standard deviation of K outputs for each raw image (Fig. 1a, "Methods" section). EMDiffuse-n produced low uncertainty values when it was confident about its predictions (Fig. 1d and Supplementary Fig. 6). The overall uncertainty

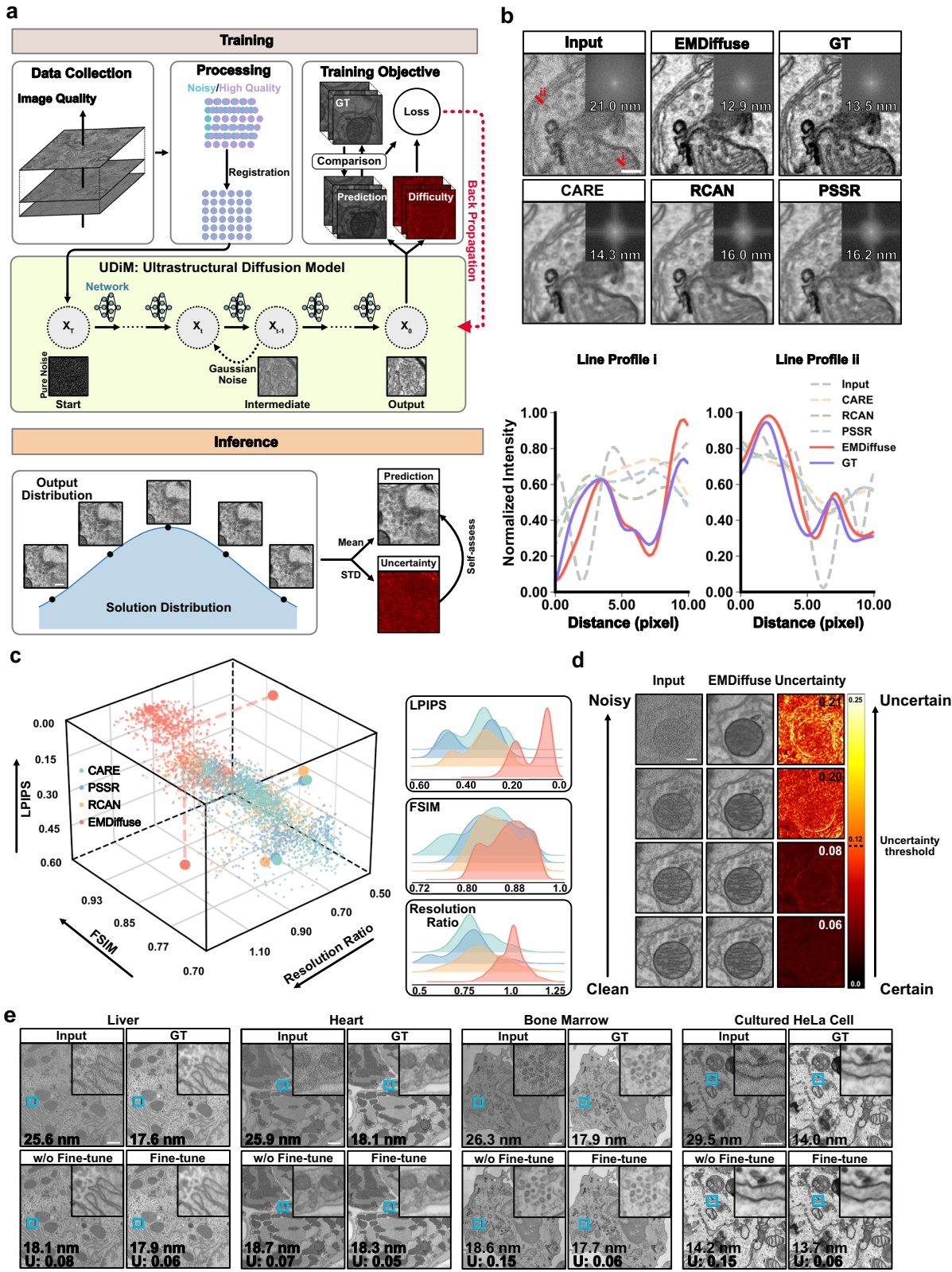

value of each input was calculated (Fig. 1d, "Methods" section). We performed a study on a dataset with varying noise levels and identified an uncertainty value threshold, 0.12, that could assist biologists and microscopists in assessing the reliability of predictions (Methods). Predictions with higher uncertainty values indicate the high variance between distinct outputs from EMDiffuse and potential inaccuracies in predicted structures.

Finally, EMDiffuse-n achieved strong generalizability and transferability (Fig. 1e). The deep learning model's performance is susceptible to the domain gap between test and training data, ranging from disparities in the structures to variations in noise levels. We explored the possibility of generalizing and transferring the pre-trained model to new types of biological samples by introducing diverse structures into the model's fine-tuning process to mitigate the domain gap

**Fig. 1 | EMDiffuse exhibits excellent denoising capability and generates images with high-resolution ultrastructural details. a** Schematic of EMDiffuse-n. In the training stage, paired data of various noise levels were collected and processed with a difficulty-aware training objective implemented. In the inference stage, EMDiffuse-n estimates and reduces the noise of raw EM images. Each test time, given an input image, EMDiffuse-n samples one potential output from the learned solution distribution. The average of the various outputs represents the final prediction, while the variance indicates the level of uncertainty. The uncertainty reflects the reliability of the prediction. **b** EMDiffuse reduces the noise and enhances the resolution of EM images with a representative region of a mouse brain cortex tissue with a pixel size of 3.3 nm processed by EMDiffuse-n, alongside a comparison of the resolution and artifacts in the same image processed with denoising algorithms. GT: ground truth image; Input: input image, CARE, PSSR, and RCAN: generated images by CARE, PSSR, and RCAN algorithms from the input image, respectively. Top right of each panel is the Fourier power spectrum. The resolution of processed images is shown on the Fourier transform image. Bottom panels are line profiles across the lines drawn in the Input panel. Min-max normalization was employed to scale the intensity values within each line plot.

**c** Quantitative performance assessment of images processed with EMDiffuse, CARE, PSSR, and RCAN algorithms using the Learned Perceptual Image Patch Similarity (LPIPS, lower values indicate superior performance) and the resolution ratio (ground truth resolution/prediction resolution, higher values indicate higher image resolution). Each point represents a unique test image of the mouse brain cortex, $n = 960$. **d** Uncertainty map for EMDiffuse processed images at different noise levels enables self-assessment of uncertainty values of the prediction. Predictions with uncertainty values of the image below 0.12 ("Methods" section) are counted as reliable outputs. **e** EMDiffuse allows robust transferability. Representative input, EMDiffuse output without fine-tuning (w/o fine-tune), EMDiffuse fine-tuning with one training image (fine-tune), and GT images from the mouse liver, heart, bone marrow tissues with a pixel size of 4.4 nm and cultured HeLa cell sample with a pixel size of 2.1 nm. A magnified region is positioned in the top right corner. The resolution of each panel is shown at the bottom left corner with nm as the unit. The uncertainty value of the results is shown in the bottom left corner. U uncertainty value. Scale bars, **a**, **d**, **e** 0.3 μm; **b** 0.1 μm. Source data are provided as a Source Data file.

(Supplementary Fig. 7a, "Methods" section). EMDiffuse-n trained on mouse brain cortex dataset was fine-tuned and tested on three different mouse tissue images (i.e., liver, heart, bone marrow) and cultured HeLa cell images acquired in-house (Supplementary Information). EMDiffuse-n had excellent generalizability and high transferability on images from biological samples different from training data (Fig. 1e and Supplementary Movie 3). Moreover, we also showed that the performance of the pre-trained EMDiffuse-n model could be easily improved by fine-tuning the decoder with only a single data pair of images of 3 megapixels from new domains (Supplementary Fig. 7). After fine-tuning, the predictions from all four datasets showed low uncertainty values and high reliability (Fig. 1e). In contrast, hundreds of pairs of training data were required to train the model from scratch on a new domain. This opens the possibility of applying EMDiffuse-n in various EM application scenarios.

## EMDiffuse achieved superior super-resolution capability

We also applied EMDiffuse for the super-resolution task (EMDiffuse-r) that reconstructs a high-resolution image, which requires a long acquisition time, from a noisy low-resolution image, which needs only a short acquisition time (Fig. 2a and Supplementary Fig. 8). A mouse brain cortex super-resolution dataset composed of paired noisy input with a pixel size of 6.6 nm and ground truth with a pixel size of 3.3 nm was obtained in-house to train EMDiffuse-r and other baselines. We optimized the prediction generation and computed the mean of two outputs for each raw input as the prediction (Supplementary Fig. 9a, "Methods" section) and computed the uncertainty values for assessing reliability. Firstly, EMDiffuse-r has exhibited superior qualitative performance, particularly in improving resolution and resolving intricate details, while other tested methods introduced unwanted smoothness (Fig. 2b, Supplementary Fig. 9b Supplementary Movie 4 and Movie 5), which was consistent from low-resolution input images with different noise levels (Supplementary Fig. 9c). For example, EMDiffuse-r successfully discerned and separated the proximal synaptic vesicles (Fig. 2b) and mitochondria cristae (Supplementary Fig. 9c) while other methods failed to differentiate them. Then, the resolution values and Fourier power spectrum further confirmed that EMDiffuse-r did not introduce the undesirable smoothness and artifacts into predictions (Fig. 2b and Supplementary Fig. 9c). Quantitative comparisons in LPIPS, FSIM, and resolution ratio metrics (Fig. 2c) reveal that EMDiffuse-r outperformed other models in all three metrics and generated the most accurate and high-resolution predictions. The Fourier ring correlation plot further indicates that EMDiffuse captures the intricate details present in the high-frequency components (Fig. 2d). Furthermore, the uncertainty value below the threshold indicated that the prediction was reliable except for extremely noisy raw inputs

(Fig. 2b and Supplementary Fig. 9c). The generalizability and robust adaptability of EMDiffuse-r also enabled easy adaptation of the proposed model to a brand-new dataset with a single training data pair of images of 3 megapixels (Fig. 2e, Supplementary Fig. 10, and Supplementary Movie 6). Notably, by super-resolving a noisy 6-nm pixel size image into a clean 3-nm pixel size image, EMDiffuse-r doubled the image resolution and facilitated a 36× increase in EM imaging speed for the specific imaging setup used in this study.

## vEMDiffuse – EMDiffuse for enhancing volume electron microscopy

Isotropic resolution reconstruction from anisotropic vEM data can significantly accelerate vEM and expand its potential applications. In this work, we further extended EMDiffuse to vEM data and developed vEMDiffuse, which can generate isotropic volumes from anisotropic ones to reduce the number of layers to be captured for volume generation and accelerate vEM data acquisition (Figs. 3 and 4). Specifically, we developed vEMDiffuse-i which incorporates a channel embedding ("Methods" section) to generate layers between the 1st and Nth layers by learning from a small volume of isotropic training data[13,15,43] (Fig. 3a). In the inference phase, vEMDiffuse-i generates vEM data of isotropic resolution from an anisotropic one by generating intermediate layers between two consecutive layers of the anisotropic volume (Fig. 3a, "Methods" section). Uncertainty values were calculated for predictions of each layer to ensure reliability.

We firstly validated the isotropic reconstruction capability of vEMDiffuse-i using downsampled volumes on two opensource isotropic datasets: OpenOrganelle mouse liver dataset (jrc_mus-liver)[44] and OpenOrganelle mouse kidney dataset (jrc_mus-kidney)[15]. For each dataset, we held out an isotropic subvolume (8 nm × 8 nm × 8 nm resolution) for training and downsampled another to an anisotropic volume (8 nm × 8 nm × 48 nm resolution) for testing by removing the axial layers. First, the XY view example of the generated liver volume (Fig. 3b, Supplementary Movie 7 and 8) indicated that vEMDiffuse-i could generate a volume with similar ultrastructural information and comparable axial resolution as the original isotropic volume. Then, an examination of a series of XY views showed that vEMDiffuse-i effectively maintained axial continuity and accurately and reliably (i.e., uncertainty values below the uncertainty threshold) replicated the ultrastructural changes of organelles within the isotropic volume (Supplementary Figs. 11, 12, and 13). Further, the YZ views (Fig. 3c, d) and XZ views (Supplementary Fig. 14) of the volume corroborated that vEMDiffuse-i accurately reconstructed the intricate structures of organelles, including mitochondria, ER, and Golgi apparatus along the axial axis. By contrast, these ultrastructural details were lost in anisotropic volume and were not restored in ITK-cubic interpolated[45]

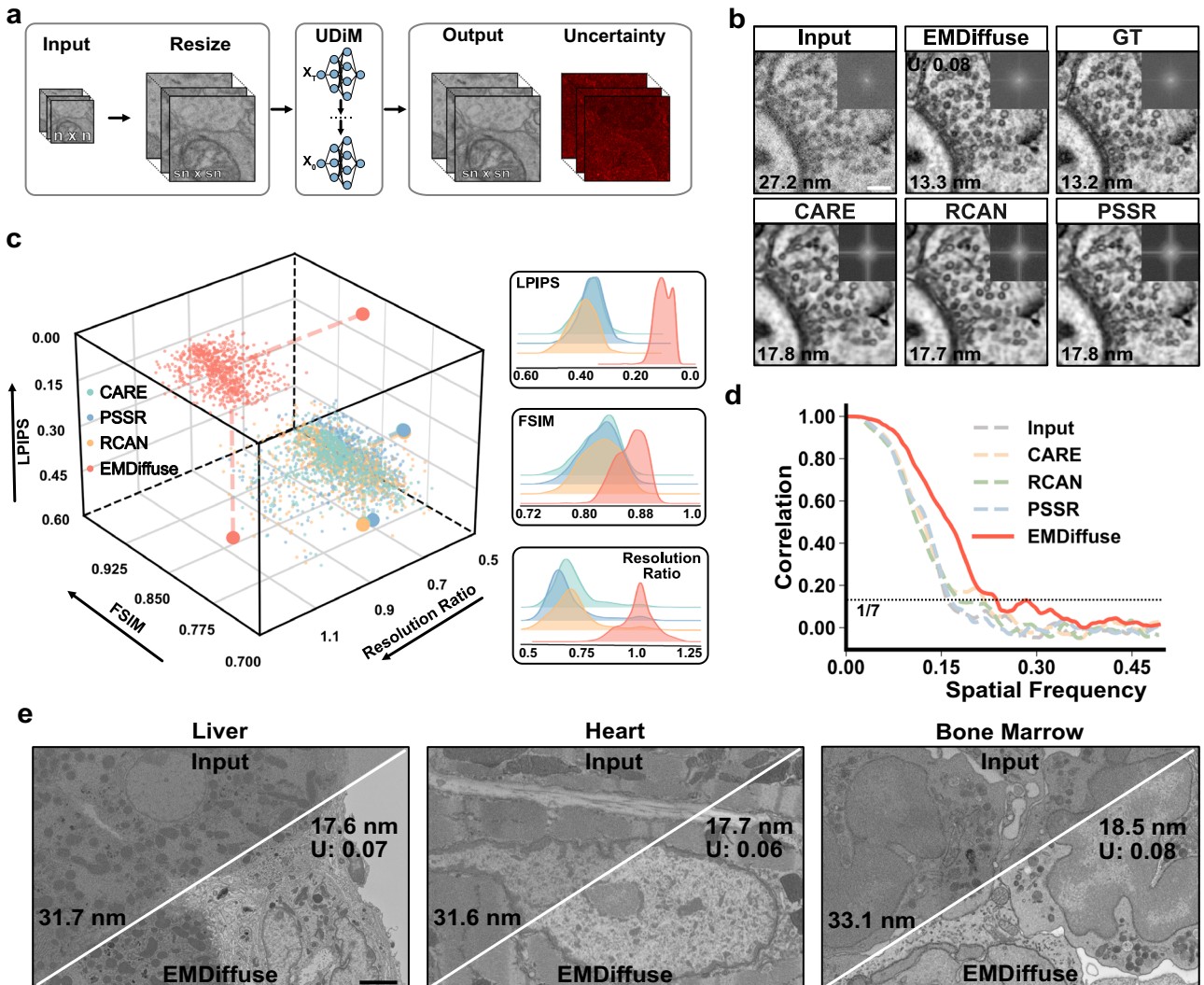

**Fig. 2 | EMDiffuse achieves superior performance in super-resolution.**
**a** Schematic of EMDiffuse-r for super-resolution of EM images. **b** Representative region from mouse brain sections (input with a pixel size of 6.6 nm, GT with a pixel size of 3.3 nm) for comparing the super-resolution capability of EMDiffuse-r with CARE, PSSR, and RCAN. Top right of each panel is the Fourier power spectrum. The resolution of processed images is shown on the bottom left of the image. The uncertainty value is shown in the top left corner. **c** The 3D scatter plot and distribution plots show the quantitative performance assessment of EMDiffuse, CARE, PSSR, and RCAN for the super-resolution task. Each point represents a unique test image of the mouse brain cortex, $n = 960$. **d** The Fourier ring correlation plot between predictions in (**b**) and ground truth image. **e** EMDiffuse for super-resolution is robust and transferable to other tissues, including the liver, heart, and bone marrow. Resolution values and uncertainty values are indicated on each panel. U uncertainty value. Scale bar, **b** 0.1 μm; **d** 0.3 μm. Source data are provided as a Source Data file.

volume (Fig. 3c, d, and Supplementary Fig. 14). What's more, we compared EMDiffuse-i with 3D-SRU-Net[46], another supervised deep learning method for isotropic reconstruction (Supplementary Information), using isotropic OpenOrganelle mouse kidney dataset (jrc_mus-kidney). 3D-SRU-Net restored most of the organelles effectively but failed to accurately generate fine structures, such as the Golgi apparatus and mitochondria cristae (Supplementary Fig. 15a, b). Quantitative evaluation using resolution ratio, LPIPS, and FSIM metrics further elaborated that vEMDiffuse-i held an improved performance over 3D-SRU-Net and ITK-cubic interpolation, both in terms of image resolution and fidelity (Supplementary Fig. 15c). Moreover, we also trained and applied vEMDiffuse-i on the OpenOrganelle T-cell[47] (jrc_ctl-id8-2, Supplementary Figs. 16 and 17) and EPFL mouse brain datasets (Supplementary Fig. 18)[48]. vEMDiffuse-i consistently generated high-quality results with low uncertainty values and accurate organelle ultrastructure (Supplementary Fig. 16 and Supplementary Fig. 18a), suggesting its universal applicability on a variety of vEM datasets with a small volume of isotropic training data.

Segmentation of ultrastructural details (e.g., organelles) from vEM data is an essential task for vEM analysis, which often requires high-quality isotropic volumes[43]. Herein, we showed that vEMDiffuse-i predictions from anisotropic input volumes enabled accurate segmentation of organelles, achieving a similar quality as that using the isotropic resolution vEM captured with FIB-SEM as input (Fig. 3e, f). We trained mitochondria and ER segmentation models on the isotropic OpenOrganelle mouse liver volume with corresponding masks (Supplementary Fig. 19, "Methods" section)[43,49]. The trained models were applied to interpolated volume, vEMDiffuse-i generated volume, and ground truth isotropic volume ("Methods" section). The vEMDiffuse-i generated volume achieved a similar Intersection over the Union (IoU) score as isotropic volume (Fig. 3e), indicating precise organelle prediction by vEMDiffuse-i. In contrast, the IoU scores of the anisotropic volume and interpolated volume were low (Fig. 3e). The 3D rendering visualizations (Fig. 3f and Supplementary Movie 9) demonstrated the accurate reconstruction of organelle ultrastructure in vEMDiffuse-i generated volume. In contrast, anisotropic volume and interpolated

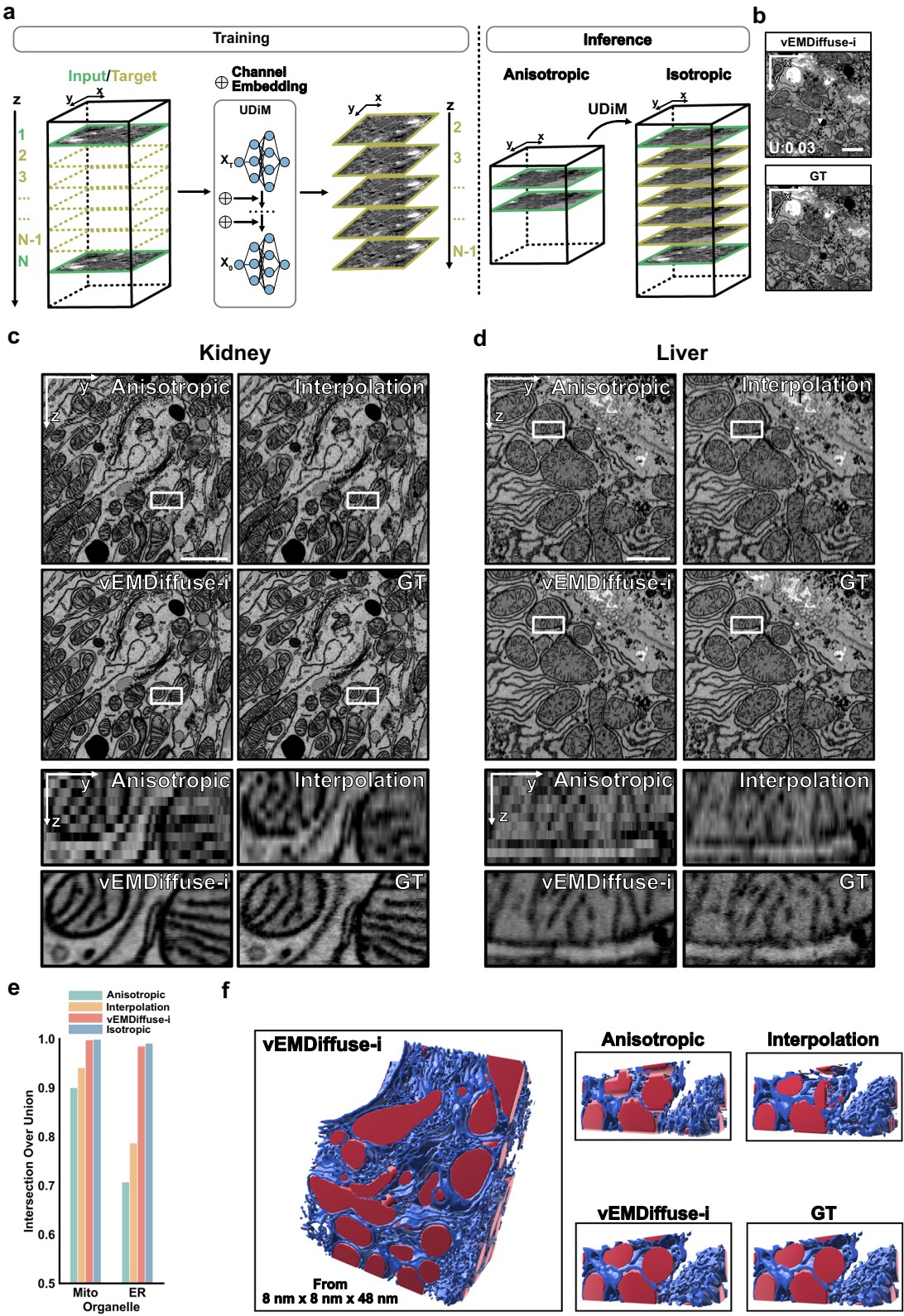

volume failed the task, which represents a major limitation of serial section-based vEM techniques.

We evaluated the capability of vEMDiffuse-i to generate isotropic volumes from anisotropic volumes of different axial resolutions from 48 nm to 96 nm (Supplementary Fig. 20). vEMDiffuse-i successfully generated fine structures of organelles with axial resolution up to 64 nm, demonstrated by high IoU scores of ER and mitochondria

segmentation masks above 0.9 (Supplementary Fig. 20a) and uncertainty values below the uncertainty threshold (Supplementary Fig. 20b). With axial resolution of 96 nm, we observed increased artifacts in generated volume, reduced IoU scores of segmentation masks (Supplementary Fig. 20a), increased uncertainty values above the uncertainty threshold (Supplementary Fig. 20b), and decreased FSIM metrics (Supplementary Fig. 20c).

**Fig. 3 | Accelerating isotropic volume electron microscopy (vEM) using vEM-Diffuse for isotropic volume reconstruction from anisotropic data. a** Schematic of vEMDiffuse-i. vEMDiffuse-i learns to predict consecutive z-slices (*yellow*) with the preceding and following slices (*green*) as input. In the inference stage, vEMDiffuse-i achieves an isotropic resolution of vEM by generating intermediate layers between two layers of anisotropic volume. **b** Representative generated XY view and the corresponding ground truth (GT) data. **c**, **d** Representative vEMDiffuse-i generated YZ views of the Openorganelle mouse kidney dataset (jrc_mus-kidney) and the Openorganelle mouse liver dataset (jrc_mus-liver). Bottom panels are enlarged boxed regions of the anisotropic volume (8 nm × 8 nm × 48 nm resolution), interpolation volume, vEMDiffuse-i generated volume, and the ground truth (GT) isotropic volume (8 nm × 8 nm × 8 nm resolution). **e** Comparison of organelle segmentation results between the anisotropic mask (downsampled from ground truth masks, 8 nm × 8 nm × 48 nm resolution), the segmentation mask of interpolated volume, vEMDiffuse-i generated volume, and ground truth isotropic volume (upper bound, 8 nm × 8 nm × 8 nm resolution) on Openorganelle mouse liver dataset. The Intersection over Union (IoU) was calculated against the ground truth mask. **f** Reconstructed 3D view based on the segmentation mask of vEMDiffuse-i generated volume (*left*) and comparison (*right*) of the 3D view of a zoomed-in volume between the anisotropic mask, the segmentation mask of interpolated volume, the segmentation mask of vEMDiffuse-i generated volume and ground truth mask. Red, mitochondria; blue, endoplasmic reticulum (ER). U uncertainty value. Scale bar, **b** 2 μm; **c**, **d** 1 μm.

## vEMDiffuse reconstructs isotropic resolution vEM volumes without isotropic resolution vEM training data

Acquiring high-quality isotropic vEM data from FIB-SEM is often impractical for most research laboratories, and the total volume that can be imaged in 3D by FIB-SEM is often small. To democratize access to isotropic vEM data and enable isotropic reconstruction of large tissue samples, we further developed vEMDiffuse-a which reconstructed isotropic volumes with only anisotropic training data (Fig. 4a). The division of input and target volumes for producing the training data in vEMDiffuse-a was the same as that of vEMDiffuse-i but occurred along the y-axis (as opposed to the z-axis in vEMDiffuse-i) ("Methods" section). The inference stage was identical to vEMDiffuse-i, where vEMDiffuse-a generated intermediate layers between two consecutive XY layers of an anisotropic volume.

To consolidate our assumption that information in the lateral axis is transferable to the axial axis, we first trained and tested vEMDiffuse-a on the manually downgraded anisotropic OpenOrganelle mouse kidney (jrc_mus-kidney) dataset[15] and downgraded anisotropic OpenOrganelle mouse kidney (jrc_mus-liver) dataset, where some layers along the z-axis have been removed to simulate an anisotropic vEM data (8 nm × 8 nm × 48 nm resolution, "Methods" section). First, although the XZ view training images contain undesired distortions (Supplementary Fig. 21a), we find that the vEMDiffuse-a model, given two consecutive XY view images as input, could produce intermediate layers with similar quality and resolution as the ground truth ones (Fig. 4b, Supplementary Fig. 21b, Supplementary Movie 10 and 11). The uncertainty value of generated layers also indicated that the predictions were reliable (Supplementary Fig. 21b). Further, the YZ and XZ views of the generated isotropic volume demonstrated that vEMDiffuse-a accurately captured the lateral information and leveraged it to enhance the axial resolution (Fig. 4c, d Supplementary Fig. 22 and Supplementary Movie 12). For example, the continuity of ER and contacts between ER and mitochondria (Fig. 4c, d) was accurately restored by vEMDiffuse-a. In contrast, in anisotropic volume, such information was lost, limiting investigations of organelle-organelle interactions in 3D. We further compared vEMDiffuse-a with CARE (Supplementary Information), neither of which required isotropic training volumes. Specifically, CARE and vEMDiffuse-a were trained and tested on downsampled 8 nm × 8 nm × 48 nm anisotropic Openorganelle mouse kidney volumes (jrc_mus-kidney). CARE and ITK-cubic interpolation faced challenges with fine ultrastructure (Supplementary Fig. 23a, b). The quantitative evaluation using resolution ratio, LPIPS, and FSIM again elaborated the superior capabilities in generating isotropic volumes of vEMDiffuse-a (Supplementary Fig. 23c). Moreover, vEMDiffuse-a improved vEM organelle reconstruction, which will facilitate 3D investigation at the organelle level from an anisotropic volume (Fig. 4e, f and Supplementary Movie 13). Again, anisotropic volume failed to reconstruct accurate 3D structures of mitochondria and ER. The mitochondria membrane segmentation and reconstruction on the Openorganelle mouse liver dataset (jrc_mus-liver) also demonstrated vEMDiffuse-a's capability to facilitate 3D ultrastructural analysis (Supplementary Fig. 24).

vEMDiffuse-a can be trained on any existing tissue array tomography type of anisotropic volumes and produce reliable isotropic volumes. We applied vEMDiffuse-a to two large open anisotropic tissue array tomography-type datasets: the MICrONS multi-area dataset[50] (Fig. 4g) and the FANC dataset[17] (Fig. 4h). vEMDiffuse-a could generate an 8 nm × 8 nm × 8 nm voxel isotropic volume from the 8 nm × 8 nm × 40 nm resolution MICrONS multi-area volume (Fig. 4g, Supplementary Fig. 25) and a 4 nm × 4 nm × 4 nm voxel volume from the 4 × 4 × 40 nm FANC volume (Fig. 4h, Supplementary Fig. 26). The lateral image quality was comparable to the original volume (Supplementary Movie 14 and 15) with uncertainty values below the threshold (Supplementary Fig. 25b, Supplementary Fig. 26b), while the axial resolution was boosted to an isotropic level, enabling researchers to observe seamless organelle ultrastructural changes along the z-axis with tissue array tomography data (Fig. 4g, h and Supplementary Movie 16). By reconstructing an anisotropic volume into an isotropic one, vEMDiffuse-a enabled the visualization of cellular structures in 3D, paving the way for researchers to map the isotropic 3D ultrastructure of organelles within large tissue samples and pushing the boundaries of applications with vEM.

We demonstrated the application of vEMDiffuse-a in neurite tracing on the downsampled MANC dataset[51] (8 nm × 8 nm × 48 nm resolution), using a Flood-Filling Network (FFN)[52] model (Supplementary Information). Indicated by high IoU scores, vEMDiffuse-a generated volume (Supplementary Fig. 27a) showed improved capabilities in generating fine neurite details and capturing complex branching structures, when compared to the over-smoothed anisotropic masks and the segmentation masks of interpolated volume (Supplementary Fig. 27b).

## Discussion

This study introduces EMDiffuse, a suite of deep learning-based methods that enhance the imaging power of ultrastructural imaging with EM and vEM. EMDiffuse expedites imaging processes and improves imaging quality through denoising and super-resolution tasks. More importantly, EMDiffuse enables robust isotropic vEM volume generation, even without the isotropic training data.

Leveraging the diffusion model, EMDiffuse exhibited superior performance in augmenting EM imaging while generating the images with nanoscale biological structure details. Contrary to the baseline models that predict the average or median of all feasible solutions, thereby inducing smoothness, EMDiffuse opts to sample a single plausible solution at each testing instance. Consequently, it enables users to produce unlimited high-resolution images for examinations of areas of interest. In our experiments, we generated two predictions and used their mean as the final output. Of note, the outputs of different test times might exhibit variability with notably noisy inputs. We have configured the EMDiffuse to allow the generation of multiple plausible solutions for future users to analyze them individually.

Moreover, EMDiffuse attempted to address the reliability conundrum in bioimage processing deep learning applications by

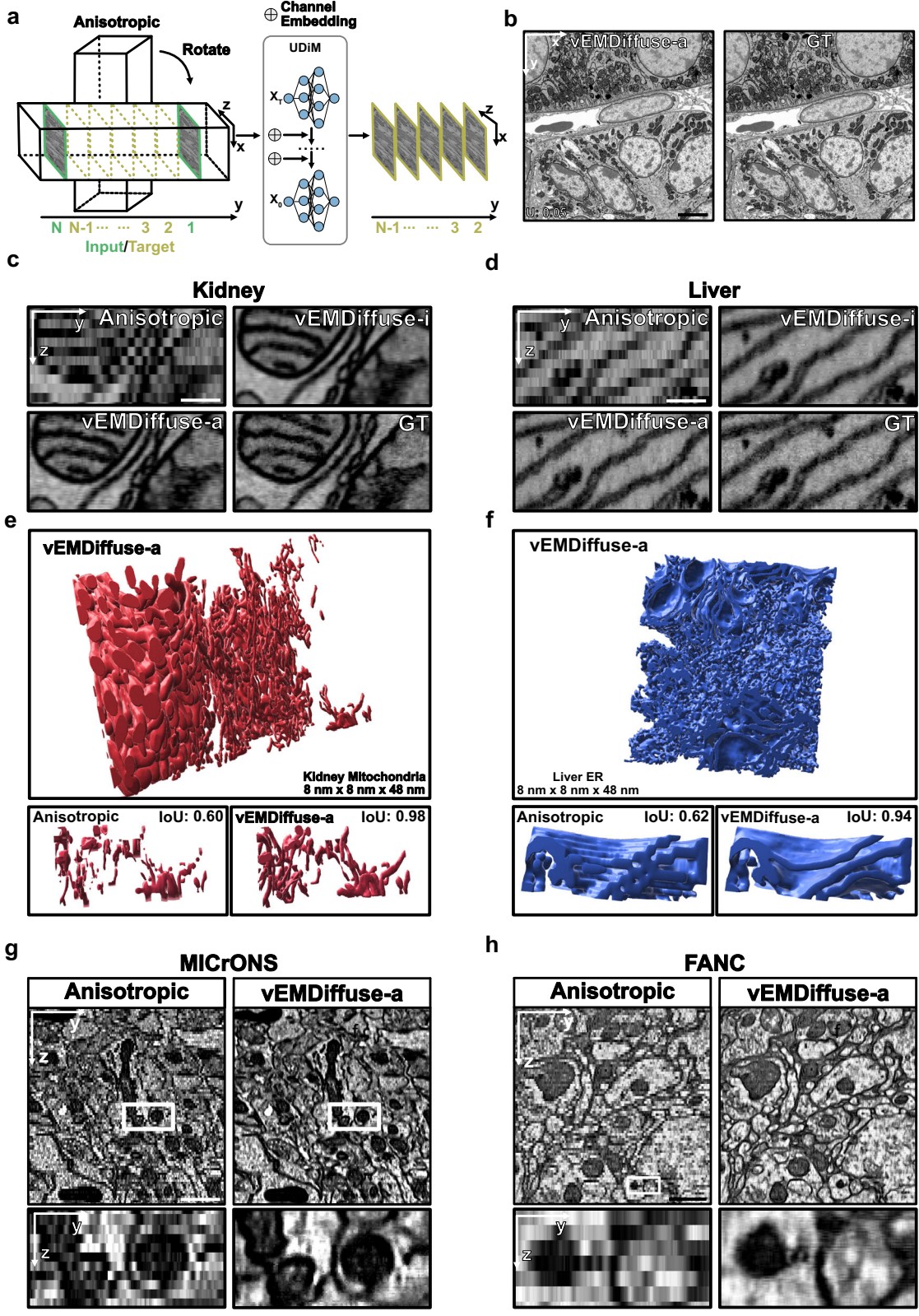

assessing the variance between different outputs. We have provided a reference threshold of 0.12, which has been applicable in assessing all tasks demonstrated in this work. However, considering the wide application range of electron microscopy, establishing a universally applicable threshold remains an elusive goal. Future users are encouraged to fine-tune the threshold by analyzing their dataset's uncertainty maps and predictions.

EMDiffuse demonstrated robustness and transferability to facilitate adaptation to other EM datasets with only one pair of fine-tuning images. Practically, microscopists may acquire a single training pair of images of 3 megapixels comprising a clean and noisy image of the target tissue to fine-tune the model. Of note, we observed the differences in the denoising performance for the direct application without fine-tuning, which likely represents the in-domain constraint of the

**Fig. 4 | Expanding vEM realm with vEMDiffuse by generating isotropic resolution volumes with tissue array tomography type of data. a** Schematic of vEMDiffuse-a. During training, vEMDiffuse-a is trained to predict successive y slices (*yellow*) conditioned on the front and back y layers of anisotropic volume (*green*). For inference, same as vEMDiffuse-i, vEMDiffuse-a generates middle layers along the z-axis between two consecutive XY images from the anisotropic volume. **b** Representative vEMDiffuse-a generated XY view of Openorganelle mouse kidney dataset (jrc_mus-kidney). **c, d** Representative vEMDiffuse-a generated YZ view of Openorganelle mouse kidney dataset (jrc_mus-kidney, **c**) and Openorganelle mouse liver dataset (jrc_mus-liver, **d**). Shown are the anisotropic volume (8 nm × 8 nm × 48 nm resolution), vEMDiffuse-i reconstruction volume, vEMDiffuse-a

reconstruction volume, and isotropic volume (8 nm × 8 nm × 8 nm resolution). **e, f** 3D reconstruction of mitochondria and endoplasmic reticulum of the two datasets. Top panels are the reconstruction of the test volume. The bottom panels exhibit comparisons of an enlarged 3D reconstruction from the anisotropic mask and the segmentation mask of vEMDiffuse-a generated volume. The overlap ratios with the ground truth (IoU) are shown in the top right corner. **g, h** Representative vEMDiffuse-a generated YZ view (*top*) and enlarged region (*bottom*) of the MICrONS multi-area dataset (8 nm × 8 nm × 40 nm resolution) and FANC dataset (4 nm × 4 nm × 40 nm). U uncertainty value. Scale bar, **b** 1.5 μm; **c, d** 0.2 μm; (MICrONS in **g**), 0.8 μm; (FANC in **h**), 0.6 μm.

deep learning model. The number of pixels or images used for fine-tuning the model will vary depending on how different the target dataset is from the training dataset.

In our experiments, EMDiffuse can faithfully restore high-quality images from noisy images and reduce the data acquisition time by a factor of 18. In the super-resolution task, EMDiffuse reduced the data acquisition time by a factor of 36 by restoring noisy and lower-resolution images to high-quality and resolution images. However, the expedition factor may vary among different EM instruments, depending on the instrument's setup, particularly the detectors' sensitivity. Nevertheless, it is possible to combine the acceleration capabilities of EMDiffuse and state-of-the-art EM instruments to push the limitations of attainable imaging areas of EM – potentially more biological insights within the same acquisition time.

vEMDiffuse broadens the capabilities of vEM imaging. EMDiffuse-i and EMDiffuse-a were designed to tackle vEM data from the most commonly used vEM techniques, FIB-SEM, and serial section-based tomography techniques. EMDiffuse-i demonstrated its robustness with isotropic generation of four volumes from distinctive biological samples of various nanoscale structures and contrast (i.e., mouse brain, mouse kidney, mouse liver, and cultured T-cell and cancer cell). vEMDiffuse-a transfers the knowledge acquired from the anisotropic volume lateral axes to the axial axes. It can be applied to any extant tissue array tomography-type anisotropic vEM data from serial section-based techniques without additional data for training. Isotropic reconstruction and axial resolution enhancement on two public extensive vEM datasets–MICrONS multi-area and FANC–have exemplified vEMDiffuse-a's capability to perform universal image transformations in existing large datasets. vEMDiffuse-a enables accurate isotropic reconstruction at the organelle scale for serial section-based tomography vEM datasets, enhancing our ability to perform the three-dimensional reconstruction of numerous nanoscale organelles within tissues that were unfeasible to investigate. The combination of vEMDiffuse-a and serial section-based tomography methods is poised to democratize the isotropic vEM.

Although the inherent model of vEMDiffuse is 2D, we incorporated axial information into the 2D diffusion model through channel embedding. This technique enables the model to accurately discern and generate specific layers within the 3D structure. What's more, compared with the 3D model, the 2D network is much more computationally and memory efficient, which makes it feasible to increase the size of the network (e.g., the number of layers) and enhance the capability of the model to process complex biological structures.

The capability of EMDiffuse is dependent on the axial resolution of the anisotropic volume, which is recommended to be as fine as the diameter of interested structures to reconstruct. For example, an axial resolution below 56 nm can reliably reconstruct fine ultrastructural details of organelles, including ER and mitochondria cristae. We expect that vEMDiffuse can also reconstruct other fine structures accurately, given an axial resolution that approximates the sizes of interested structures, such as synaptic vesicles and synapses, to facilitate connectomics studies. A poorer axial resolution significantly risks fine structures being indiscernible or inaccurately reconstructed,

appearing smeared across successive layers, thus undermining the reconstruction.

Further refinements to EMDiffuse are anticipated to fully harness the potential impact of this deep learning model on biological imaging and cell biology. The stable diffusion model[33] could be adopted as the backbone of EMDiffuse to accelerate the inference processes. The integration of algorithms of the EMDiffuse package can be investigated for further enhancements of EM imaging, particularly to accelerate the data acquisition and reconstruction of large isotropic volumes. The continued development of EMDiffuse will pave the way for investigations of three-dimensional ultrastructural changes in large volumes of biological systems with isotropic resolution.

## Methods
### Denoise data pre-processing
The inherent distortions and horizontal-and-vertical drifts associated with EM imaging[53] may result in misaligned training data (Supplementary Fig. 1a). Therefore, it is crucial to align the data properly for training the EMDiffuse model. In this study, we presented a two-stage registration strategy for coarse-to-fine alignment of the raw image and the ground truth reference image. The coarse stage employed a keypoint matching-based registration technique[54], which estimated a global homography matrix between them based on ORB (Oriented FAST and Rotated BRIEF) key points and feature descriptors[55]. The homography matrix was used to warp the raw image and aligned it with the reference image. However, the coarse stage could only align the two images globally, and local misalignments still existed (Supplementary Fig. 1b). Hence, in the fine stage, we employed optical flow estimation[56] to estimate dense correspondences between pixels in two image patches which were further used to warp the raw image to achieve better local alignment with the reference image.

Additionally, for the sake of training efficiency, we cropped 256 × 256 image patches with a stride of 196 to construct the data for training and inference, resulting in around 20,000 patches for training, 1000 patches for validation, and 1000 patches for inference. In the inference stage, for one raw image, the corresponding output patches from the model were stitched together using Imaris Stitcher (Oxford Instruments Group) to be the final output.

### EMDiffuse architecture, diffusion process, and training objective
The architecture of UDiM and the diffusion process of EMDiffuse are included in Supplementary Information. We observed that EMDiffuse training often got stuck at the hard cases (e.g., extremely noisy raw images in the denoising task). Therefore, inspired by an existing method[37], we added an additional prediction head that produced a difficulty assessment map $\boldsymbol{\varphi}_i$ and used it to weight the prediction error. This resulted in a difficulty-aware loss function[37]:

$$L = \frac{1}{N}\sum_i \left( \frac{|\boldsymbol{\epsilon} - \boldsymbol{\epsilon}_\theta(\sqrt{\overline{\alpha_t}}\mathbf{x}_0 + \sqrt{1 - \overline{\alpha_t}}\boldsymbol{\epsilon}, t, \boldsymbol{c}_r)|^2}{2\boldsymbol{\varphi}_i^2} + \log(\boldsymbol{\varphi}_i) \right) \quad (1)$$

Where $\mathbf{x}_0$ is the ground truth, $\mathbf{c}_r$ is the raw input image, $\boldsymbol{\epsilon} \sim \mathcal{N}(0,1)$, $\boldsymbol{\epsilon}_\theta$ is the network, t is the timepoint, $\bar{\alpha}_t = \prod_{i=1}^{t} \alpha_i$ and $\alpha_i$ is the noise schedule (more details are included in Supplementary Information).

## Inference stage

To generate samples from the target distribution, we start from a random noise sample $x_T$ and iteratively apply the denoising function $p_\theta(x_{t-1}|x_t)$ in reverse order, from step T to step 0:

$$\mathbf{x}_{t-1} = \frac{1}{\sqrt{\alpha_t}} \left( \mathbf{x}_t - \frac{1-\alpha_t}{\sqrt{1-\bar{\alpha}_t}} \boldsymbol{\epsilon}_\theta(\mathbf{x}_t, t, \mathbf{c}_r) \right) + \mathcal{N}(0, \boldsymbol{\Sigma}_t),$$
$$\text{where } \Sigma_t = (1 - \alpha_t) \cdot \boldsymbol{I} \tag{2}$$

Also, each step is also guided by the raw input $\mathbf{c}_r$. Through step-by-step denoising, we can approximate the target distribution $p(x_0)$ in the final step $x_0$ as the final prediction from the diffusion model. As for uncertainty prediction, we adopt the last step (i.e., $t = 0$) of the diffusion process as the final variance map.

## Denoising neural networks training

Five neural networks, CARE[28], RCAN[39], PSSR[23], Noise2Noise[29], and Noise2Void[19], were compared with our EMDiffuse-n on the denoising task. Each model was trained and tested on the aligned and cropped multiple noise levels mouse brain cortex SEM dataset. To prevent overfitting, the training was stopped after the validation loss did not decrease for 20 epochs. The checkpoint with the best performance on the validation set was selected for testing. Data augmentation techniques such as random flip and rotation were applied during training. All experiments were done on a computer workstation equipped with four RTX 3090Ti NVIDIA graphic processing cards. All the plots were generated using Matplotlib and Seaborn libraries in Python. More details of baseline implementation are included in Supplementary Information.

EMDiffuse-n implementation was done using Python version 3.8 and Pytorch version 1.7.0[57]. The Adam optimizer[58] with an initial learning rate of 5e-5 was employed to train EMDiffuse-n. The learning rate remained constant throughout the training processes since we observed no overfitting except in transfer learning experiments. The batch size was 64. Gaussian blur was included in data augmentation, which proved beneficial in SR3[59] (Supplementary Table 1). Given paired high-quality image $\mathbf{x}^{gt}$ and noisy image $\mathbf{c}_n$, the training objective for EMDiffuse-n is:

$$L = \frac{1}{N} \sum_i \left( \frac{|\boldsymbol{\epsilon} - \boldsymbol{\epsilon}_\theta \left( \sqrt{\bar{\alpha}_t} \mathbf{x}_0^{gt} + \sqrt{1-\bar{\alpha}_t} \boldsymbol{\epsilon}, t, \mathbf{c}_n \right)|^2}{2\boldsymbol{\varphi}_i^2} + \log(\boldsymbol{\varphi}_i) \right) \tag{3}$$

## Optimization of prediction generation

When the noise level of a raw image is high, denoising and super-resolution tasks become notorious ill-posed one-to-many problems. In regions where noise dominates, multiple potential solutions exist (Supplementary Fig. 3a). To model such inherent ambiguities, we harnessed the power of diffusion models in sampling one plausible solution at each test time. Specifically, given a single raw image $\mathbf{c}_r$, we sampled from the learned solution distribution multiple times to produce inputs $\{\mathbf{x}_T^1, \mathbf{x}_T^2, \ldots, \mathbf{x}_T^K\}$ for the diffusion model, which in turn produces multiple results $\{\mathbf{x}_0^1, \mathbf{x}_0^2, \ldots, \mathbf{x}_0^K\}$. These diverse outputs captured the inherent multi-modal solutions. To alleviate the ambiguities in the predictions, we used the mean of these diverse outputs to be the final prediction $\mathbf{x}_0$.

$$\mathbf{x}_0 = \frac{1}{K} \sum_{m=1}^{K} (\mathbf{x}_0^m) \tag{4}$$

As the task becomes increasingly ill-posed with a higher noise level in the raw input image, we increased the number of $K$ during inference. However, if the noise level of the raw image is low, incorporating a larger $K$ to calculate the mean may introduce undesired smoothness into the prediction, potentially undermining the prediction resolution (Supplementary Fig. 3c, d). Therefore, to achieve a balance of image quality and prediction robustness, we empirically determined the optimal number of $K$ for a given input based on its noise level using the multi-noise-level mouse brain denoise dataset. We processed the data with different values of $K$ and evaluated the quality of the outputs using FSIM as the metric (Supplementary Fig. 4a). We observed that $K = 2$ achieved superior FSIM scores in most cases and enhanced the predictions in most noise cases (noise level > 50). At extreme noise cases, using a larger $K$ might contribute only marginal enhancement, and the resultant outputs might lack reliability, as the raw image is too noisy to restore some intricate structures (Supplementary Fig. 6). Users should be cautious about the predictions if the input is excessively noisy. This could also be reflected in our uncertainty score as described in the following. We measured the noise level of the raw image using BRISQUE which will be covered later.

## Assessing the reliability of predictions

**Uncertainty value.** Estimating the epistemic uncertainty of the model predictions is vital for enhancing reliability and aiding biologists in making informed decisions. We employed the variance of the $K$ outputs of the diffusion model as the model's epistemic uncertainty (Supplementary Fig. 6), as the prediction becomes increasingly trustworthy if multiple outputs agree. Our methodology evaluates uncertainty on a patch-by-patch basis for large images. Specifically, we divide large images into smaller patches of $256 \times 256$ pixels. For each patch, min-max normalization was applied to enhance the local contrast. Then, we calculated the standard deviation (STD) of K outputs $\{\mathbf{x}_0^1, \mathbf{x}_0^2, \ldots, \mathbf{x}_0^K\}$ ($K = 2$ in our experiments):

$$\boldsymbol{\delta}_i = \sqrt{\frac{\sum_{m=1}^{K} (\mathbf{x}_0^m - \mathbf{x}_0)^2}{K-1}} \tag{5}$$

Where the $\mathbf{x}_0$ is the average of outputs. Given the uncertainty map $\boldsymbol{\delta}_i$, we obtained the uncertainty value of the prediction by extracting the 99th percentile pixel value of $\boldsymbol{\delta}_i$. This percentile was chosen to exclude unreliable predictions even when the model was uncertain only in a minor section of the image. Instead of using the maximum pixel value, this strategy helped avoid undesired high uncertainty values caused by outliers and high contrast pixels. The high contrast indicates sudden shifts in pixel intensities which often happens in structural boundaries, such as organelle membranes will induce elevated standard deviations and thus high uncertainty value (Supplementary Fig. 6). However, this does not necessarily mean incorrect predictions of those regions. After empirical analysis of approximately 200 image predictions and their corresponding ground truths from our mouse brain cortex denoising dataset, 1% of the maximum uncertainty pixels were empirically disregarded in uncertainty estimation to avoid the above issues (Supplementary Fig. 6). Nevertheless, because electron microscopy has a wide range of applications, the quantity of disregarded pixels may vary across different datasets and can be modified by users through examination of the uncertainty map and estimation of the unreasonably high number of uncertain pixels. Upon reassembly, if any individual patch exceeds the predetermined uncertainty threshold, EMDiffuse flags the entire image and the specific patch as potentially problematic.

## Uncertainty threshold

We established an uncertainty threshold $\tau$ to guide biologists in practice. For one input, the prediction was considered reliable if the uncertainty value 99th percentile pixel value of $\boldsymbol{\delta}_i$) was below $\tau$. We

determined the uncertainty threshold by examining the prediction accuracies of our multi-noise-level mouse brain denoise dataset using FSIM and LPIPS. We observed there is a transition valley between the low LPIPS and high LPIPS regions (Fig. 1c), which we found corresponds to images that are at the boundary of being accurate predictions (low LPIPS and high FSIM). We thus calculated the uncertainty of these samples and used their average to be the uncertainty threshold $\tau$ which was found to be 0.12.

## EMDiffuse-n transfer learning

EMDiffuse-n data-efficient fine-tuning. First, we examined EMDiffuse-n's generalization capability by directly applying the pre-trained model from the mouse brain cortex dataset to these new datasets (Fig. 1e). Second, we investigated the effectiveness of transferring the pre-trained EMDiffuse-n model to new tissues in a data-efficient manner by fine-tuning the model using few-shot tissue samples. To help the model avoid overfitting to the limited training samples and achieve better performance, we only fine-tuned the decoder and bottleneck layer parameters, as opposed to all model parameters (Supplementary Fig. 7a). We also investigated the performance of the model after fine-tuning using the different numbers of training samples (Supplementary Fig. 7b).

## EMDiffuse-n fine-tuning details

During training, we used a learning rate of 1e-5 and fine-tuned EMDiffuse-n over 200 epochs on each dataset. All other configurations, such as the optimizer, remained consistent with those used in training EMDiffuse-n. To eliminate the impact of randomness, all experiments were replicated five times using the same training configurations.

## EMDiffuse-r for super-resolution

In the case of EMDiffuse-r, raw images were captured with a magnification factor of 20,000× with a pixel size of 6.6 nm, utilizing dwell time of 2 μs, 4 μs, and 6 μs. The ground truth reference images were acquired at a 40,000× magnification factor with a pixel size of 3.3 nm and 36 μs dwell time, covering half the region size of the raw images. Images were acquired using the FEI Verios SEM with an acceleration voltage of 2 kV. For data pre-processing, the same registration and cropping pipeline were employed, resulting in 15,000 well-aligned pairs for training EMDiffuse-r and other super-resolution models, 800 pairs for validation, and 800 pairs for testing. The raw, low-resolution 128 × 128 image patch was rescaled to match its target high-resolution size of 256 × 256. The same training strategy and the same loss function were adopted as EMDiffuse-n. Similarly, we set parameter $K$ to 2 for calculating the mean and uncertainty map as output (Supplementary Fig. 9a). The loss function for EMDiffuse-r was:

$$L = \frac{1}{N}\sum_i\left(\frac{|\boldsymbol{\varepsilon} - \boldsymbol{\epsilon}_\theta\left(\sqrt{\overline{\alpha_t}}\mathbf{x}_0^{gt} + \sqrt{1-\overline{\alpha_t}}\boldsymbol{\varepsilon}, t, \mathbf{c}_n\right)|^2}{2\boldsymbol{\varphi}_i^2} + \log(\boldsymbol{\varphi}_i)\right) \quad (6)$$

$\mathbf{x}^{gt}$ was high-resolution image and $\mathbf{c}_n$ was rescaled low-resolution image.

## EMDiffuse-r transfer learning

To validate the transferability of EMDiffuse-r, we transferred EMDiffuse-r trained on mouse brain cortex dataset to mouse liver, mouse heart, mouse bone marrow, and cultured HeLa cell datasets (the same datasets in EMDiffuse-n transfer learning). Since these three datasets were designed for transfer learning in the denoising task, the raw input was acquired with the same magnification rate as GT. Inspired by PSSR[23], we addressed the scarcity of low-resolution images by manually downsampling the raw input with a short acquisition time. However, unlike PSSR, since we already had noisy images with low dwell time, we didn't have to manually add Gaussian noise.

Remarkably, we also achieved superior performance with a single pair of noisy downsampled images and GT (Fig. 2d) with the previously introduced transfer learning method.

## Super-resolution baseline training

For the super-resolution task, CARE[28], RCAN[39], PSSR[23] were compared with EMDiffuse-r. Each model was trained and tested on the same well-aligned brain cortex dataset consisting of low-resolution noisy input and high-resolution ground truth image. The low-resolution image was rescaled to match the target image size. The implementation and training details were kept the same as the denoising task.

## vEMDiffuse-i for isotropic reconstruction

vEMDiffuse-i was designed to restore an isotropic volume (voxel size 8 nm × 8 nm × 8 nm in our experiments) from an anisotropic one (voxel size 8 nm × 8 nm × 48 nm in our experiments) by interpolating intermediate layers $\{\mathbf{x}_0^1, \mathbf{x}_0^2, \ldots, \mathbf{x}_0^R\}$ between two adjacent layers $\mathbf{c}^u$ (the upper layer), $\mathbf{c}^l$ (the lower layer) in an anisotropic volume (Fig. 3a).

## Channel embedding

As the axial resolution of vEM fluctuates across various datasets, channel embedding ensures the model can produce the appropriate number of interpolation layers (R) without architectural modification, which allows model architecture to be reused across different datasets. The number of interpolation layers R is determined by the resolution of the training dataset to generate isotropic volumes. Channel embedding in vEMDiffuse employs sinusoidal positional encoding to embed an integer j $\epsilon$[1,R] into a feature. This integer signifies the relative position between the target layer and the upper layer $\mathbf{c}^u$. vEMDiffuse is designed to produce one layer per iteration. In each training iteration, an integer j $\epsilon$[1,R] is randomly selected as the input for channel embedding. The model is then trained to generate the $j^{th}$ layer between the upper layer $\mathbf{c}^u$ and the lower layer $\mathbf{c}^l$. In the inference stage, vEMDiffuse employs channel embedding to embed the index ranging from 1 to R and generates corresponding intermediate layers sequentially.

## vEMDiffuse-i training

During each training iteration, vEMDiffuse-i randomly selected one layer as the upper layer $\mathbf{c}^u$ in training isotropic volume and concatenated it with $\mathbf{c}^{u+R+1}$ layer of the volume (i.e., $\mathbf{c}^l = \mathbf{c}^{u+R+1}$). The model was required to generate $\{\mathbf{x}_0^{u+1}, \mathbf{x}_0^{u+2}, \ldots, \mathbf{x}_0^{u+R}\}$ layers of the volume. The training objective of vEMDiffuse-i was:

$$L = \frac{1}{N}\sum_{i=0}^{N}\sum_{j=1}^{R}\left(\frac{||\boldsymbol{\epsilon}_t - \boldsymbol{\epsilon}_\theta\left(\sqrt{\overline{\alpha_t}}\mathbf{x}_0^{u+j} + \sqrt{1-\overline{\alpha_t}}\boldsymbol{\epsilon}, t, \mathbf{c}^u, \mathbf{c}^l, j\right)||^2}{2\sigma_i^2} + \log(\boldsymbol{\sigma}_i)\right) \quad (7)$$

No overfitting was observed in our dataset, and we chose the checkpoint when vEMDiffuse-i's performance stabilized on the validation set. For each dataset, we trained a model for it.

## vEMDiffuse-i inference

During the inference stage, given an anisotropic volume $\{\mathbf{c}^1, \mathbf{c}^2, \ldots, \mathbf{c}^M\}$ with M layers, we traverse every pair of adjacent layers $\{\mathbf{c}^u, \mathbf{c}^{u+1}\}$ of the volume: $[\{\mathbf{c}^1, \mathbf{c}^2\}, \{\mathbf{c}^2, \mathbf{c}^3\}, \ldots, \{\mathbf{c}^{M-1}, \mathbf{c}^M\}]$. For each pair of input, vEMDiffuse-i generated R images $\{\mathbf{x}_0^1, \mathbf{x}_0^2, \ldots, \mathbf{x}_0^R\}$ between them:

$$\mathbf{x}_0^j = model(\mathbf{c}^u, \mathbf{c}^{u+1}, j)\, j = 1, 2, \ldots, R \quad (8)$$

Then we inserted the generated layers between $\{\mathbf{c}^u, \mathbf{c}^{u+1}\}$. Thus, the final generated volume $\{\mathbf{c}^1, \mathbf{x}_0^1, \mathbf{x}_0^2, \ldots, \mathbf{c}^2, \ldots, \mathbf{c}^M\}$ had ((M −1) × R + 2) layers. To enhance the performance of vEMDiffuse-i, we

obtained two outputs ($K = 2$) for each input and use their mean as the final prediction $\mathbf{x}_0^j$.

### vEMDiffuse-i inference dataset preparation

In the inference stage, we firstly cropped $1024 \times 1024 \times 1024$ subvolume (not used in the training stage) from OpenOrganelle Kidney isotropic volume[15] and OpenOrganelle Liver isotropic volume[44] as test datasets (voxel size is 8 nm × 8 nm × 8 nm). We manually removed some layers to simulate the anisotropic volumes. Starting from $\mathbf{c}^1$, we retained only the layers with indices whose remainder is 1 after divided by 6 while discarding the others:

$$\mathbf{c}^j = \begin{cases} \mathbf{c}^j \; if \; j \; mod \; 6 = 1 \\ NULL \; if \; j \; mod \; 6! = 1 \end{cases} \tag{9}$$

$mod$ means the remainder after division. This reduces the axial voxel size from 8 nm to 48 nm.

### vEMDiffuse-a for isotropic reconstruction

In vEMDiffuse-i, we extracted layers along the z-axis of isotropic volume to train, and thus we required isotropic volume for training. However, obtaining isotropic volume is impractical for many research groups. In vEMDiffuse-a, we trained the model's isotropic reconstruction only with anisotropic volume.

### vEMDiffuse-a model training

In vEMDiffuse-a, we extracted layers along the y-axis of the anisotropic volume (equivalent to the z-axis of isotropic volume in vEMDiffuse-i) to generate input and target pair to train (Fig. 4a). In the training stage, vEMDiffuse-a was required to generate $\{\mathbf{xz}_0^{u+1}, \mathbf{xz}_0^{u+2}, \ldots, \mathbf{xz}_0^{u+R}\}$ XZ layers of the volume given input pair of upper and lower XZ layer pair $\{\mathbf{c}_{xz}^u, \mathbf{c}_{xz}^l\}$. Of note, $\{\mathbf{c}_{xz}^u, \mathbf{c}_{xz}^l\}$ and $\{\mathbf{xz}_0^{u+1}, \mathbf{xz}_0^{u+2}, \ldots, \mathbf{xz}_0^{u+R}\}$ are both XZ view images exhibiting reduced image quality with line artifacts (Supplementary Fig. 25a and Supplementary Fig. 26a). We find the model is robust to these artifacts. We used the same training strategy as vEMDiffuse-i, but we noticed severe overfitting after a long training time, resulting in XZ-like predictions with clear line artifacts. To avoid overfitting, we terminated the training after 1200 epochs. Again, for each dataset, we trained a model for it.

### vEMDiffuse-a inference

The inference stage of vEMDiffuse-a was equivalent to that of vEMDiffuse-i. vEMDiffuse-a interpolated intermediate layers between upper and lower input pair $\{\mathbf{c}_{xy}^u, \mathbf{c}_{xy}^l\}$. Again, of note, in the training stage, the input $\{\mathbf{c}_{xz}^u, \mathbf{c}_{xz}^l\}$ were XZ view images and the model was trained to interpolate layers along the y-axis. However, in the inference stage, the input $\{\mathbf{c}_{xy}^u, \mathbf{c}_{xy}^l\}$ were XY view images of the anisotropic volume and vEMDiffuse-a interpolated layers along the z-axis.

### vEMDiffuse-a training and inference dataset preparation

To prove our concept that vEMDiffuse-a could transfer knowledge learned from the lateral axis (XZ views along the y-axis of anisotropic volumes at the training stage) to the axial axis (XY views along the z-axis of anisotropic volumes at inference stage), we firstly manually downgraded isotropic volumes: Openorganelle Kidney dataset[15] and OpenOrganelle Liver isotropic volume[44], and trained vEMDiffuse-a on the downgraded anisotropic volume. Given the isotropic volume $\{\mathbf{c}_{xy}^1, \mathbf{c}_{xy}^2, \ldots, \mathbf{c}_{xy}^M\}$, we reduced the axial resolution by removing layers:

$$\mathbf{c}_{xy}^j = \begin{cases} \mathbf{c}_{xy}^j \; if \; j \; mod \; 6 = 1 \\ NULL \; if \; j \; mod \; 6! = 1 \end{cases} \; j = 0, 1, \ldots M \tag{10}$$

Of note, in vEMDiffuse-i, we only downgraded the volume in the inference stage while in the training stage, we still used isotropic volumes. Conversely, in vEMDiffuse-a, we downgraded the volume in both the training and inference stage. Then, we chopped along the y-axis to form training data: $\{\mathbf{c}_{xz}^u, \mathbf{c}_{xz}^l\}$ and $\{\mathbf{xz}_0^{u+1}, \mathbf{xz}_0^{u+2}, \ldots, \mathbf{xz}_0^{u+R}\}$ as elucidated above.

For original anisotropic datasets such as the MICrONS multi-area and FANC multi-area datasets, we directly sliced along the y-axis. For both MICrONS multi-area and FANC datasets, we reserved $2048 \times 2048 \times 128$ subvolume as test datasets (not used in the training stage). In practice, researchers can also use the same anisotropic volume in the training and inference stage.

### Organelle segmentation

The organelle segmentation was trained with the isotropic volume and corresponding official segmentation masks provided by Openorganelle[15]. A 3D U-Net segmentation model was trained on isotropic data. More details of implementation are included in Supplementary Information. Once trained, we applied the model to (1) ground truth isotropic volume, (2) vEMDiffuse generated volume, and (3) interpolated volume. The Intersection over Union (IoU) ratio was calculated against the ground truth mask (isotropic resolution). We applied erosion and dilation post-processing techniques to remove small false positive regions and fill in small hollow holes. As for the anisotropic mask, we directly downsampling the ground truth mask, the same as our protocol for volume downsampling. We utilized Imaris (by Oxford Group) to interpolate the resultant anisotropic mask. We rendered the 3D visualization with Imaris, automatically filtering out some small regions.

### Statistics and reproducibility

For each denoising and super-resolution experiment, we repeated experiments three times and showed the representative image from 960 images (Figs. 1b, d–e, 2b, e). Additionally, each vEM reconstruction experiment (including vEMDiffuse-i and vEMDiffuse-a) was independently repeated at least three times for each dataset (Fig. 4b–d, 4g–h).

### Reporting summary

Further information on research design is available in the Nature Portfolio Reporting Summary linked to this article.

## Data availability

Denoising and super-resolution training and test data used in EMDiffuse are available at https://zenodo.org/records/10205819. For vEM, all OpenOrganelle datasets are downloaded from the OpenOrganelle website (https://openorganelle.janelia.org). The Openorganelle Kidney dataset is available at https://doi.org/10.25378/janelia.16913035.v1. The Openorganelle Liver dataset is available at https://doi.org/10.25378/janelia.16913047.v1. The Openorganelle T-Cell dataset is available at https://doi.org/10.6084/m9.figshare.14447541.v1. The EPFL mouse brain dataset is available at https://www.epfl.ch/labs/cvlab/data/data-em/. The MICrONS multi-area dataset can be downloaded from https://www.microns-explorer.org/. The FANC dataset can be downloaded from https://bossdb.org/project/phelps_hildebrand_graham2021. The MANC dataset can be downloaded from https://www.janelia.org/project-team/flyem/manc-connectome. Source data are provided with this paper.

## Code availability

The source codes of EMDiffuse, several representative pre-trained models as well as some example images for testing are publicly accessible via https://github.com/Luchixiang/EMDiffuse.

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

## Acknowledgements

This work was supported by the Research Grants Council of Hong Kong (17102722, 17202422, 27209621) and the Australian Research Council (LP190100433). The work was conducted in the JC STEM Lab of Molecular Imaging, funded by The Hong Kong Jockey Club Charities Trust. We also thank Dr. Jing Guo for her assistance in image processing.

## Author contributions

H.J., X.Q., and C.L. designed the experiments and wrote the paper. C.L., K.C., H.Q., X.C., and G.C. performed, collected, and assembled the experiments. H.J. and X.Q. secured funding. All have commented on and edited the manuscript.

## Competing interests

The authors have declared that no conflict of interest exists.
