## [Peer Review File · Nature Communications]

Diffusion-based deep learning method for augmenting ultrastructural imaging and volume electron microscopyREVIEWER COMMENTS

Reviewer #1 (Remarks to the Author):

In the manuscript "EMDiffuse: a diffusion-based deep learning method augmenting ultrastructural imaging and volume electron microscopy", Chixiang Lu and colleagues describe their experiments using Stable Diffusion models to perform

1. 2D image denoising (EMDiffuse-n)
2. 2D super-resolution (EMDiffuse-r)
3. 3D slice in-painting to achieve isotropic resolution (vEMDiffuse-i and -a)

The authors use the now popular 2D-U-Net architecture with global attention for their training. They added a difficulty channel, and for 3D slice in-painting, they provide two inputs and a slice index. They train using perceptual losses to overcome the smoothing that comes with training for PSNR. They explored variability, and decided empirically that averaging two independent predictions works best (compared to just one prediction or averaging many more).

For -n, they acquire image pairs with different SNR and align them. For -r, they simulate low resolution data from high resolution data. For -i, they train from simulated anisotropic resolution data from isotropic resolution data, and for -a, they train from re-sliced non-isotropic data.

They directly compared 2D-denoising and 2D-super-resolution with the state of the art methods CARE, PSSR, and RCAN. For 3D slice in-painting, they show that organelle segmentation works better with their upscaled data than with naively upscaled data. Some details are missing to fully understand this.

They provide their new denoising training data under restricted access (per request) on Zenodo, all other datasets that they used were freely available under permissive licenses by others. The GitHub code repository as listed in the manuscript does not exist or is not openly accessible.

I find the described work useful and worth publishing such that others can build on these results. Using diffusion models for denoising and super-resolving scanning EM data is new and has not been sufficiently explored. The proposed approaches for isotropic super-resolution (particularly -a) are very interesting. The manuscript in its current form, however, should be substantially rewritten such that readers can fully understand the method and comparison experiments, the language used is often inconsistent and difficult to disentangle, there are frequent unspecific speculative statements that do not contribute to a better understanding, streamlining this and sticking to consistent terminology would help a lot, some serious hands-on editing help by the publisher would probably be a good idea.

I have a few major concerns that should be addressed:

The comparison with other methods (CARE, PSSR, N2V, N2N) lacks details about whether the other algorithms had been trained on the same data, and if so, how. Please add all details that explain whether this is an apples to apples or and apples to oranges comparison.

Using generative deep learning to produce isotropic super-resolution is very interesting but has been explored before, e.g. by the earlier mentioned CARE and PSSR methods, differences in the approach would be interesting to understand for the reader (e.g. 2D reslicing vs. 2D slice in-painting). The 3D SR-U-Net by Heinrich et al. (https://link.springer.com/chapter/10.1007/978-3-319-66185-8_16) is likely worth discussing as well (e.g. to highlight 3D vs 2D interpolation U-Net architectures). A comparison with CARE, PSSR, and 3D SR-UNet for generating isotropic 3D volumes from non-isotropic volumes is missing but would add a lot to this paper.

The details of the segmentation comparison are not clear to me. Have the 3D U-Nets been trained on both downsampled and isotropic data or did you use one training and applied it to downsampled and isotropic data? Is the IoU metric on the downsampled results over the downsampled pixels against downsampled GT or over upsampled results and isotropic GT?

Data and code are, at this time, not available. Please make them available, similar to the publicly available data and code that have been used to create this work.

Incomplete list of minor comments as noted while reading

565 remove "the" before "training"

581 Fix sentence

727 what is the fidelity enhancement module?

729 e.g. -> i.e.

730 is K again just 2? Please elaborate. Supplement? Isn't σ_i the uncertainty?

733--742 unclear, please rewrite simpler, 1% of the image being ignored sounds a lot, not little, or I am misunderstanding something? What maximum value? Just the maximum uncertainty pixel? Why just one! And how does this relate to the ignored 1%?

744 for σ ?

804-820 This description sounds a little bit as if the desired resolution could be arbitrary, but if I understand the approach correctly, the step size between channel indices c is defined by the resolution of the training data (int both $-i$ and $-a$ experiments, correct? Could you please clarify this? Otherwise, a floating point precision c in $[0:1]$ would be more logical, wouldn't it? But there's no training data for that?

836 "enhancethe"

Reviewer #2 (Remarks to the Author):

The manuscript proposes to use diffusion models for EM image enhancement comments. It tackles four different tasks:

- Denoising of EM Images, to enable acquisition with lower dwell times
- Super-resolution of EM Images, to improve the resolution
- Resampling EM data to isotropic resolution (with paired training data)
- Resampling EM data to isotropic resolution (without paired training data)

The diffusion model is trained and applied conditioned on the noisy / lower resolution / anisotropic data. Further, the authors use the generative properties of the diffusion model to predict uncertainties and prescribe empirical thresholds that can be used to judge whether the model predictions can be trusted. They compare favorably to several baseline methods (but see below for caveats).

Overall, I believe that this work constitutes an important contribution to image enhancement for volume EM, as it presents a unified framework for several practically important related tasks, and shows very promising performance.

However, I have a few concerns that should be addressed before the manuscript is published:

1. It fails to cite important prior work. In particular Heinrich et al.

(https://link.springer.com/chapter/10.1007/978-3-319-66185-8_16), which has introduced a method for predicting isotropic from anisotropic EM data. It is one of the first methods to tackle EM enhancement tasks, and also constitutes prior work for one of the tasks the authors tackle.

Consequently, the claim that this work is the first to tackle this task (Line 291, "More importantly, EMDiffuse ventures into an uncharted territory of isotropic vEM volume generation and broadens the horizons for vEM applications.") is not true and must be correct.

The authors should also cite Buchholz et al.

(<https://www.sciencedirect.com/science/article/abs/pii/S0091679X19300706>), who apply CARE to EM, similar to the CARE baseline used by the authors. I would also suggest to check this work for special considerations when applying CARE to EM, which may be used to improve baseline experiments. (See next point).

2. The authors do not describe the baseline methods (CARE, Noise2Void, etc.) in any detail. It is unclear how these methods are used by the authors, and hence it is unclear whether the improved image quality compared to them is due to actual method improvements or due to insufficient training of the baselines.

The following points must be addressed for all baselines:

- How are the baseline methods trained? Are they trained on the same training volumes as the proposed method? Is the training budget similar to the proposed method?
- How are the hyperparameter of the baselines tuned? Is similar effort spend on this compared to the proposed method?

Addressing these points in a new detailed methods section is crucial to understand the exact

contribution of EMDiffuse.

3. The github link (<https://github.com/Luchixiang/EMDiffuse>) given in the manuscript does currently not work. Hence I could not evaluate the state of the software itself, since it was not provided with the supplementary material either. Given that the practical impact of the method could be very significant, it would be of high importance to provide user-friendly and well documented software. The authors should submit a version of their software with the next submission (either by uploading it to github or adding it to the supplementary material), so that this can be reviewed.

Reviewer #3 (Remarks to the Author):

This manuscript introduced algorithms designed to enhance EM and vEM capabilities, EMDiffuse, leveraging the cutting-edge image generation diffusion model. EMDiffuse demonstrates superior denoising and super-resolution performance, generates realistic predictions without unwarranted smoothness. EMDiffuse perhaps has a powerful tool in reducing imaging time for vEM dataset. The results are significant and interesting. Figures are sufficient. This reviewer would like to comments some.

“Resolution” is a key word in this manuscript, so it should be defined at beginning of the main text for reader’s better understanding. How to measure “resolution” should be described concretely in methods, though it is roughly indicated in lines 689-693 and cited reference 39.

I understand that results related to the Figures 1 and 2 may be based on their original serial section-based tomography vEM datasets obtained with either FE-SEM, and Figures 3 and 4 may be based on open source vEM datasets obtained with FIB-SEM, which are indicated in line 922-933 at the end of Methods. It is unclear that which dataset was used to obtain each result, such as EMDiffuse-n, EMDiffuse-n training, EMDiffuse-n transfer learning, EMDiffuse-r and so on, shown in the Figure panels. These should be clearly indicated in main text or figure legends, with datasets conditions: thickness of the serial section, pixel size, FIB-SEM, FE-SEM, open source or home made.

In line 485 – 486, “Sections of mouse tissues, measuring 500 nm in thickness, were mounted onto silicon wafers.” This sentence is ambiguous. Whether it means either “serial sections of mouse tissues of 500 nm thick sections were mounted onto silicon wafers”, or “serial sections of mouse tissues of 50 nm thick section, measuring 500 nm in total thickness, were mounted onto silicon wafers”. It should be clearly described.

The GitHub page (<https://github.com/Luchixiang/EMDiffuse>) is currently inaccessible, rendering it impossible to assess the effectiveness of this technology for EM data in general. It is imperative that the GitHub page be made accessible, and the associated code should be made available.

While I find this technique to be highly effective for denoising and super resolution of typical structures, I have concerns regarding potential artifacts in the case of rare structures.

Specifically:

- When a new sample is trained with limited data using transfer learning.
- When the size of the structure aligns closely with the z step of the original image.

Could you discuss the possibility of artifacts being introduced in these scenarios? If such a possibility exists, please elaborate on the nature of these potential artifacts.

Since many users will conduct 3D segmentation post-application of this method, it is crucial to validate and demonstrate the efficacy of this method for 3D segmentation. I am particularly interested in understanding its effectiveness when applied to SSTEM data. While it may be highly effective for mitochondrial segmentation, how about its application to mitochondria-associated membranes? Addressing these questions and discussing the effectiveness and limitations of this method in these contexts is essential.

One of the motivations for applying this method is automating neurite tracing. If possible, could you provide insights into how the accuracy of neurite auto-tracing improves with and without the application of this method? Additionally, I would appreciate understanding the effectiveness of this method for SSTEM data with large z steps.

The manuscript mentions SEM imaging magnification but lacks information on the pixel size (nm/pixel), which is now standard manner to describe a magnification of electron micrograph. Please include details about the pixel size.

Acceleration voltage is a matter for scanning electron micrograph to collect tissue image signals from the depth. It should be described in Method.

The authors compare their proposed methods, EMDiffuse-n and EMDiffuse-r, with state-of-the-art methods such as CARE in Figures 1 and 2. To clarify the effectiveness of vEMDiffuse-i for vEM image segmentation, it would be helpful to provide the following:

- The intersection over union (IoU) value of segmentation using "Interpolation" in Figure 3e.
- A segmentation example in Figure 3f.

The authors use channel embedding to generate images in vEMDiffuse-i. Is there any relationship between the number of interpolated images R and the segmentation accuracy?

The users of vEMDiffuse-i would be appreciated if you provide the relationship. Is there a risk of overfitting if R is too large?

Non-machine learning experts would appreciate a more detailed explanation of the difficulty assessment map (L573), including equations, in the main text.

In Extended Data Figure 16c, the input appears to be the same image as the input for Case 1 in Figure 16b. Is this correct?

The loss function is stated to follow Ho et al. (2020) in L565. However, the equation in L569 specifies that the noise is a function of time t. Is it correct to understand that $\epsilon_t = N(0, 1)$? If so, for non-expert readers in machine learning, it would be appreciated if t were removed.

Non-expert readers in machine learning would appreciate it if you explicitly stated the objective functions for EMDiffuse-n and EMDiffuse-r, as is done for vEMDiffuse-i in L826.

In Extended Data Figure 4, the PSNR values are higher for 6Mean than for Single. This suggests that PSNR is a good metric for evaluating the quality of denoised images. Is this correct?

Reviewer #4 (Remarks to the Author):

EMDiffuse: a diffusion-based deep learning method augmenting ultrastructural imaging and volume electron microscopy

Summary:

=====

This manuscript proposes ways of applying diffusion models on Electron Microscopy (EM) and volume EM (vEM) tasks. The authors claim that by applying their method, EM and vEM tasks can significantly increase the imaging speed without compromising on the quality of acquired data and also enable the isotropic imaging of large biological volumes.

In particular, the paper shows the application of their diffusion model setup on (i) denoising, (ii) super-resolution (SR), and (iii) volume reconstruction without using isotropic training data. The manuscript demonstrates the applicability of the proposed method by performing experiments on public data and introduces a self-assessment method (based on uncertainty readouts) that users can utilize to judge the quality and reliability of obtained results.

The proposed method (the DiReP Network) relies on a combination of existing diffusion model training strategies, which to the best of our knowledge are for the first time applied to the presented tasks and data.

In terms of validation of their results, the manuscript notes that the PSNR metric should not be used, since it favors over-smoothed predictions. As an alternative, the authors propose to use FSIM (Feature Similarity Index), BRISQUE (Blind/Referenceless Image Spatial Quality Evaluator), and LPIPS (Learned Perceptual Image Patch Similarity). With respect to those metrics, the proposed method outperforms several baselines, i.e. CARE, RCAN PSSR, but when suitable also N2N and N2V.

All presented approaches (quickly summarize below) are based on well established Diffusion Model training and inference procedures. In particular, solutions for the following four tasks are presented:

1. Image Denoising:

A variant of DiReP Network, called EMDiffuse-n, is used in their denoising pipeline. During inference, an input is fed to the trained DiReP network and two predictions are sampled

from the diffusion posterior. Those two samples are averaged (pixel-wise) to produce the final prediction.

For quality assessment FSIM (compared against imaged ground truth data) was used. The presented method is reported to improve 30% over other baselines.

Additionally, EMDiffuse-n also outputs an uncertainty map, i.e. the pixel-wise difference between the 2 above mentioned samples (the std if more than 2 samples are used). The uncertainty map can be used to reject predictions whenever an uncertainty threshold (in their case 0.12) in the average of all pixel-wise uncertainties is superseded.

The authors also show that EMDiffuse-n can be used in a transfer learning setting, where a model trained on mouse brain cortex data was finetuned to denoise data from mouse liver, heart, or bone marrow data. Finetuning on a single image reduces the uncertainty score mentioned above considerably.

2. Image Super-Resolution:

The paper then applied a variant of the same diffusion model pipeline called the EMDiffuse-r to perform a SR task. First, the lower resolution images gets resized to its desired dimensions, then a trained DiReP network is used to predict the higher resolution result. Again, the mean of two samples is the final prediction.

The authors demonstrated superior performance in restoring fine structures when compared with other baseline methods, i.e. CARE, RCAN, and PSSR.

Additionally, the authors calculate the uncertainty for the predictions and show that the outputs are reliable.

Finally, the authors show that the model trained on mouse cortex data is able to generalize to other datasets, again finetuned on a single additional image, i.e. from the mouse heart, liver, and bone marrow data used throughout the presented work.

3. Volume reconstruction with isotropic vEM data:

The manuscript proposes to reconstruct the axial volume of a section using the first and n-th slice of the section using another variant called the vEMDiffuse-i.

To this end, the model takes two consecutive raw data slices and a suitably encoded

“channel embedding”, i.e. the index of the slice [1,R] between the two given raw slices.

The uncertainty calculation method previously mentioned is also used here.

The authors use no direct metrics to assess the reached performance, but go on showing that on a downstream segmentation task, evaluated in terms of IoU, their predicted isotropic volume outperforms the same segmentation applied on the raw (anisotropic) data.

4. Volume reconstruction with anisotropic vEM data:

While for vEMDiffuse-i isotropic data is required during training, the authors finally introduce a variation called vEMDiffuse-a, which can operate on anisotropic training data alone.

This is accomplished by training on XZ slices instead of XY slices, the advantage being, that in the remaining Y direction, we have isotropic resolution and hence the required training data to replicate the vEMDiffuse-i procedure (on a surrogate XY interpolation task). Once trained, the trained model is then applied directly to XY slices, thereby predicting the missing data to make the original volume isotropic.

As before, the authors use no direct metrics to assess the reached performance, but go on showing that on a downstream segmentation task, evaluated in terms of IoU, their predicted isotropic volume outperforms the same segmentation applied on the raw (anisotropic) data.

Strengths:

=====

1. The manuscript addresses an important area in EM and vEM reconstruction and has the potential to find broad application.
2. The authors propose a single diffusion model-based setup that is applicable to multiple practical tasks.
3. The authors provide a way for users to assess the uncertainty in the predicted data.

4. The authors show that their method is good at transfer-learning to other (yet somewhat similar) data.
5. The authors compare their method against several relevant baselines and show that their method outperforms them.
6. The authors also use suitable perceptual metrics such as FSIM, BRISQUE, and LPIPS for measuring the performance of their method. It is regrettable though, that PSNR and/or SSIM evaluations are not at all available in the manuscript.

Weaknesses/Open Questions:

=====

1. The authors propose a method based on existing DL technologies that promises to be of great utility for life-scientists that acquire EM data. In the light of this setup, the reviewer thinks it is required to make the full source of the method publicly available. The link given in the section “Code Availability” leads to an error 404, making it impossible for this reviewer to evaluate the quality of the codebase to be made available. I consider this to be the biggest problem to propose acceptance of this manuscript.
2. From a purely DL-engineering perspective, it is very regrettable that there are no ablation studies presented. The reviewer is certain that multiple slightly different approaches have been tried and it would be very interesting to hear how much worse they have performed. One example: instead of interpolating R-fold (in vEMDiffuse models), the “channel embedding” could encode the relative position between the two given slices, which would even make it possible to transfer learn between datasets that obey a different value for R. Or: how much worse are results without the difficulty-aware loss term? In the CV field, such ablations are considered essential to tell a complete story about a proposed method. The reviewer will leave it to the editor to decide how important such information is for a venue like Nature Communication.
3. The authors rely on an uncertainty metric threshold that is determined on some given

data. Will 0.12 now work for all data users would like to use? The reviewer thinks this is not the case, posing the question how future users should go about evaluating a suitable threshold for their own data. Being more explicit about this in the manuscript seems desirable.

4. In Figure 3(e), quantitative results are only shown for the anisotropic volume and the prediction using vEMDiffuse-i. The reviewer would appreciate to also see the same quantification for GT (as an upper bound) and the interpolated volume (as a baseline).

5. The paper deals with various mouse datasets, presumably being more similar than other samples where cells and organelles have much diverging sizes and the subcellular milieu can even contain structures not present in mice. It would be interesting to see how trained models would transfer and how much finetuning would improve predictions. Side note: the fact that vEMDiffuse-a works well indicates that a trained model is relatively robust to quite drastic changes in data appearance. Still, a more quantitative approach would be desirable.

6. Reporting an improvement in percent (e.g. the reported 30% in line 130) was more confusing to the reviewer than helpful. 30% in what? It might be better to state the respectively used metric and the absolute numbers that lie 30% apart.

7. The image quality assessment metric LPIPS, the pre-trained neural network is usually trained with natural images. In this case, it is unclear how this would affect the performance reporting of the metric.

8. Not at all reporting on classical and widely used metrics such as PSNR or SSIM is regrettable.

9. The text is overall well written, but slightly deteriorates in later sections (and in more technical sections). A thorough pass over all text sections is highly recommended.

10. Please correct derivation to deviation on line 729

11. Please correct LIPSI to LIPS on line 748.

Point-by-point response to the reviewers' comments

REVIEWER COMMENT

Reviewer #1 (Remarks to the Author):

In the manuscript "EMDiffuse: a diffusion-based deep learning method augmenting ultrastructural imaging and volume electron microscopy", Chixiang Lu and colleagues describe their experiments using Stable Diffusion models to perform 1. 2D image denoising (EMDiffuse-n) 2. 2D super-resolution (EMDiffuse-r) 3. 3D slice in-painting to achieve isotropic resolution (vEMDiffuse-i and -a). The authors use the now popular 2D-U-Net architecture with global attention for their training. They added a difficulty channel, and for 3D slice in-painting, they provide two inputs and a slice index. They train using perceptual losses to overcome the smoothing that comes with training for PSNR. They explored variability, and decided empirically that averaging two independent predictions works best (compared to just one prediction or averaging many more).

For -n, they acquire image pairs with different SNR and align them. For -r, they simulate low resolution data from high resolution data. For -i, they train from simulated anisotropic resolution data from isotropic resolution data, and for -a, they train from re-sliced non-isotropic data.

They directly compared 2D-denoising and 2D-super-resolution with the state of the art methods CARE, PSSR, and RCAN. For 3D slice in-painting, they show that organelle segmentation works better with their upscaled data than with naively upscaled data. Some details are missing to fully understand this. They provide their new denoising training data under restricted access (per request) on Zenodo, all other datasets that they used were freely available under permissive licenses by others. The GitHub code repository as listed in the manuscript does not exist or is not openly accessible.

I find the described work useful and worth publishing such that others can build on these results. Using diffusion models for denoising and super-resolving scanning EM data is new and has not been sufficiently explored. The proposed approaches for isotropic super-resolution (particularly -a) are very interesting.

The manuscript in its current form, however, should be substantially rewritten such that readers can fully understand the method and comparison experiments, the language used is often inconsistent and difficult to disentangle, there are frequent unspecific speculative statements that do not contribute to a better understanding, streamlining this and sticking to consistent terminology would help a lot, some serious hands-on editing help by the publisher would probably be a good idea.

RESPONSE: We thank the reviewer for acknowledging the strength of the manuscript. With regard to the code and training data, we have made the code (<https://github.com/Luchixiang/EMDiffuse>) and dataset (<https://zenodo.org/records/10205819>) public. We have also undertaken an extensive revision to enhance the understandability of the methods and comparison experiments, removed unspecific speculative statements, and ensured consistent terminology throughout.

I have a few major concerns that should be addressed:

The comparison with other methods (CARE, PSSR, N2V, N2N) lacks details about whether the other algorithms had been trained on the same data, and if so, how. Please add all details that explain whether this is an apples to apples or apples to oranges comparison.

RESPONSE: Thank you for this suggestion. In all experiments, baseline methods were trained on the same dataset as EMDiffuse. We have included the details of the comparison methods in the revised manuscript in Lines 642-691 (for denoising) and Lines 878-883 (for super-resolution).

Using generative deep learning to produce isotropic super-resolution is very interesting but has been explored before, e.g. by the earlier mentioned CARE and PSSR methods, differences in the approach would be interesting to understand for the reader (e.g. 2D reslicing vs. 2D slice in-painting). The 3D SR-U-Net by Heinrich et al. (https://link.springer.com/chapter/10.1007/978-3-319-66185-8_16) is likely worth discussing as well (e.g. to highlight 3D vs 2D interpolation U-Net architectures). A comparison with CARE, PSSR, and 3D SR-UNet for generating isotropic 3D volumes from non-isotropic volumes is missing but would add a lot to this paper.

RESPONSE: Thank you for the suggestion on comparing our approach with CARE, PSSR, and 3D-SRU-Net⁴⁵. We have now included additional results comparing 3D-SRU-Net with vEMDiffuse-i (Extended Data Fig. 15, *see below*) in the revised manuscript, both of which used isotropic volumes as ground truth for training. Specifically, we compared the use of 3D-SRU-Net and vEMDiffuse-i for the reconstruction of the Openorganelle mouse kidney dataset (jrc_mus-kidney), and both were trained on the 8 nm x 8 nm x 8 nm isotropic volume. Subsequent testing was done on a held-out downsampled 8 nm x 8 nm x 48 nm anisotropic volume from the same dataset. The YZ (Extended Data Fig. 15a) and XZ (Extended Data Fig. 15b) corroborated that although 3D-SRU-Net was able to restore most of the organelles effectively, for fine structures like the Golgi apparatus and mitochondria cristae, 3D-SRU-Net failed to accurately reconstruct them. Quantitative evaluations regarding resolution ratio, LPIPS, and FSIM metrics further manifested that vEMDiffuse-i held an improved performance over 3D-SRU-Net and conventional interpolation techniques, both in terms of image resolution and fidelity (Extended Data Fig. 15c). The details of 3D-SRU-Net training method used in the comparison are included in Lines 931-940.

We have also included the comparison between CARE and vEMDiffuse-a for isotropic reconstruction (Extended Data Fig. 23, *see below*), neither of which used isotropic training volumes. Specifically, CARE and vEMDiffuse-a were trained and tested on downsampled 8 nm x 8 nm x 48 nm anisotropic Openorganelle mouse kidney volumes (jrc_mus-kidney). While CARE showcased proficiency in restoring large organelles, it also faced challenges with fine structures, similar to the limitations observed with 3D-SRU-Net (Extended Data Fig. 23a and 23b). The quantitative evaluation using resolution ratio, LPIPS, and FSIM again elaborated the superior reconstruction capabilities of vEMDiffuse-a (Extended Data Fig. 23c). The details of the CARE 3D reconstruction training method are included in Lines 986-995 in the revised manuscript.

We have not included the comparison with PSSR for isotropic 3D reconstruction, as its primary design was to enhance lateral resolution rather than focusing on isotropic reconstruction.

Extended Data Fig. 23 vEMDiffuse-a outperforms other unsupervised isotropic reconstruction methods.

a, b. Example XZ view and YZ view and enlarged regions of the anisotropic volume (8 nm x 8 nm x 48 nm resolution), interpolated volume (ITK-cubic), CARE generated volume, vEMDiffuse-a generated volume, and ground truth isotropic volume (GT, 8 nm x 8 nm x 8 nm resolution) on the Openorganelle mouse kidney dataset (jrc_mus-kidney). Scale bar, 1 μ m

c. The violin plots of three metrics (LPIPS, FSIM, and resolution ratio) show the quantitative performance assessment of vEMDiffuse-a for the isotropic reconstruction task compared with ITK-cubic interpolation and CARE.

The details of the segmentation comparison are not clear to me. Have the 3D U-Nets been trained on both downsampled and isotropic data or did you use one training and applied it to downsampled and isotropic data? Is the IoU metric on the downsampled results over the downsampled pixels against downsampled GT or over upsampled results and isotropic GT?

RESPONSE: Thank you for highlighting the need for clarity regarding our segmentation comparison. The 3D U-Net segmentation model was trained on isotropic data. Once trained, we applied the model to (1) ground truth isotropic volume, (2) vEMDiffuse-i generated volume, and (3) interpolated volume. The Intersection over Union (IoU) ratio was calculated against the ground truth mask (isotropic resolution). As for the anisotropic mask, we directly downsampled the ground truth mask. We utilized Imaris (Oxford Group) to interpolate the resultant anisotropic mask. Subsequent IoU calculations for this mask were performed against the ground truth mask (isotropic resolution). The vEMDiffuse-i generated volume achieved a similar IoU score as isotropic volume (**Fig. 3e**), indicating precise organelle prediction by vEMDiffuse-i. In contrast, the IoU scores of the anisotropic volume and interpolated volume were low (**Fig. 3e**). The 3D rendering visualizations (**Fig. 3f**) demonstrated the accurate reconstruction of organelle ultrastructure in vEMDiffuse-i generated volume. In contrast, anisotropic volume and interpolated volume failed the task, which represents a major limitation of serial section-based vEM techniques. Accordingly, we have updated the manuscript and figures. Please refer to new **Fig. 3e** and **Fig. 3f**, and Lines 238-248 and Lines 997-1016 of the manuscript.

Fig. 3 Accelerating isotropic volume electron microscopy (vEM) using vEMDiffuse for isotropic volume reconstruction from anisotropic data.

e. Comparison of organelle segmentation results between the anisotropic mask (downsampled from ground truth masks, 8 nm x 8 nm x 48 nm resolution), the segmentation mask of interpolated volume, vEMDiffuse-i generated volume, and ground truth isotropic volume (upper bound, 8 nm x 8 nm x 8 nm resolution) on Openorganelle mouse liver dataset. The Intersection over Union (IoU) was calculated against the ground truth mask. **f.** Reconstructed 3D view based on the segmentation mask of vEMDiffuse-i generated volume (left) and comparison (right) of the 3D view of a zoomed-in volume between the anisotropic mask, the segmentation mask of interpolated volume, the segmentation mask of vEMDiffuse-i generated volume and ground truth mask. Red, mitochondria; blue, endoplasmic reticulum (ER).

Data and code are, at this time, not available. Please make them available, similar to the publicly available data and code that have been used to create this work.

RESPONSE: Thanks for your comments. We have made the code (<https://github.com/Luchixiang/EMDiffuse>) and dataset (<https://zenodo.org/records/10205819>) public.

Incomplete list of minor comments as noted while reading

565 remove "the" before "training"

RESPONSE: Thanks for noting it. We have revised it in our updated manuscript.

581 Fix sentence

RESPONSE: Thank you for pointing that out. The sentence has been fixed in the revised manuscript.

727 what is the fidelity enhancement module?

RESPONSE: Thanks for noticing this. The fidelity enhancement module was an intermediary step during our method's developmental phase. However, it was not used in the final EMDiffuse. We have removed it from the manuscript.

729 e.g. -> i.e.

RESPONSE: Thank you for pointing it out. We have revised it.

730 is K again just 2? Please elaborate. Supplement? Isn't σ_i the uncertainty?

RESPONSE: Thanks for the comment. K is consistently set at 2 in EMDiffuse, and σ_i is the uncertainty. We have revised the manuscript accordingly in Lines 797-805.

733--742 unclear, please rewrite simpler, 1% of the image being ignored sounds a lot, not little, or I am misunderstanding something? What maximum value? Just the maximum uncertainty pixel? Why just one! And how does this relate to the ignored 1%?

RESPONSE: Thank you for pointing out the need for clarification in this section. Given the uncertainty map δ_i , we obtained the uncertainty value of the prediction by extracting the 99th percentile pixel value of δ_i . This percentile was chosen to exclude unreliable predictions even when the model was uncertain only in a minor section of the image. Instead of using the maximum pixel value, this strategy helped avoid undesired high uncertainty values caused by outliers and high contrast pixels. The high contrast indicates sudden shifts in pixel densities which often happens in structural boundaries, such as organelle membranes will induce elevated standard deviations and thus high uncertainty value (Extended Data Fig. 6). However, this does not necessarily mean incorrect predictions of those regions. Thus, 1% of the maximum uncertainty pixels were disregarded in uncertainty estimation to avoid the above issues. We have modified the paragraph in the revised manuscript (Lines 806-815)

744 for σ ?

RESPONSE: Yes. For each input, the prediction was considered reliable if the uncertainty value (99th percentile value of δ_i) was below the uncertainty threshold, τ .

804-820 This description sounds a little bit as if the desired resolution could be arbitrary, but if I understand the approach correctly, the step size between channel indices c is defined by the resolution of the training data (int both -i and -a experiments, correct? Could you please clarify this? Otherwise, a floating point precision c in $[0:1]$ would be more logical, wouldn't it? But there's no training data for that?

RESPONSE: Thank you for pointing out this confusion. For each dataset, the step size between channel indices c is defined by the resolution of that dataset. The channel embedding is designed to allow the model architecture to be flexibly reused across different datasets without modifying the model architecture. We have clarified the methodology of channel embedding, please refer to Lines 891-902 in the revised manuscript.

836 "enhancethe"

RESPONSE: Thank you for pointing out the typo. We have corrected it in the manuscript.

Reviewer #2 (Remarks to the Author):

The manuscript proposes to use diffusion models for EM image enhancement comments. It tackles four different tasks:

- Denoising of EM Images, to enable acquisition with lower dwell times
- Super-resolution of EM Images, to improve the resolution
- Resampling EM data to isotropic resolution (with paired training data)
- Resampling EM data to isotropic resolution (without paired training data)

The diffusion model is trained and applied conditioned on the noisy / lower resolution / anisotropic data. Further, the authors use the generative properties of the diffusion model to predict uncertainties and prescribe empirical thresholds that can be used to judge whether the model predictions can be trusted. They compare favorably to several baseline methods (but see below for caveats). Overall, I believe that this work constitutes an important contribution to image enhancement for volume EM, as it presents a unified framework for several practically important related tasks, and shows very promising performance.

RESPONSE: We thank the reviewer for acknowledging the strength of our manuscript.

However, I have a few concerns that should be addressed before the manuscript is published:

1. It fails to cite important prior work. In particular Heinrich et al. (https://link.springer.com/chapter/10.1007/978-3-319-66185-8_16), which has introduced a method for predicting isotropic from anisotropic EM data. It is one of the first methods to tackle EM enhancement tasks, and also constitutes prior work for one of the tasks the authors tackle. Consequently, the claim that this work is the first to tackle this task (Line 291, "More importantly, EMDiffuse ventures into an uncharted territory of isotropic vEM volume generation and broadens the horizons for vEM applications.") is not true and must be correct. The authors should also cite Buchholz et al. (<https://www.sciencedirect.com/science/article/abs/pii/S0091679X19300706>), who apply CARE to EM, similar to the CARE baseline used by the authors. I would also suggest to check this work for special considerations when applying CARE to EM, which may be used to improve baseline experiments. (See next point).

RESPONSE: Thank you for drawing our attention to these critical references. We have revised the statement in the manuscript and included these key references. We have now included additional results comparing 3D-SRU-Net with vEMDiffuse-i (Extended Data Fig. 15, *see above*), which both use isotropic training volumes. Specifically, we compared the use of 3D-SRU-Net and vEMDiffuse-i for the reconstruction of the Openorganelle mouse kidney dataset (jrc_mus-kidney), and both were trained on the 8 nm x 8 nm x 8 nm isotropic volume. Subsequent testing was done on a held-out downsampled 8 nm x 8 nm x 48 nm anisotropic volume from the same dataset. The YZ (Extended

Data Fig. 15a) and XZ (Extended Data Fig. 15b) corroborated that although 3D-SRU-Net was able to restore most of the organelles effectively, for fine structures like the Golgi apparatus and mitochondria cristae, 3D-SRU-Net failed to accurately reconstruct them. Quantitative evaluation using resolution ratio, LPIPS, and FSIM metrics further elaborated that vEMDiffuse-i held an improved performance over 3D-SRU-Net and conventional interpolation techniques, both in terms of image resolution and fidelity (Extended Data Fig. 15c). The details of 3D-SRU-Net training method used in the comparison are included in Lines 931-940.

We have reviewed and cited the work by Buchholz *et al.* on applying CARE to EM, ensuring our baseline aligns with established approaches. We have confirmed that our training configuration aligns with their application of CARE to EM, and included the training details of CARE in the Methods section, please refer to Lines 653-656.

2. The authors do not describe the baseline methods (CARE, Noise2Void, etc.) in any detail. It is unclear how these methods are used by the authors, and hence it is unclear whether the improved image quality compared to them is due to actual method improvements or due to insufficient training of the baselines. The following points must be addressed for all baselines:

- How are the baseline methods trained? Are they trained on the same training volumes as the proposed method? Is the training budget similar to the proposed method?
- How are the hyperparameter of the baselines tuned? Is similar effort spend on this compared to the proposed method?

Addressing these points in a new detailed methods section is crucial to understand the exact contribution of EMDiffuse.

RESPONSE: We appreciate this suggestion. All baseline methods were trained on the same dataset as EMDiffuse. We ensured consistent and sufficient training for these methods, optimizing their performance on the validation sets. Hyperparameters were optimized for each baseline method. For all models, we adjusted key parameters, including learning rate, batch size, and learning rate schedule to guarantee optimal performance. We have included the training and implementation details of five baselines and EMDiffuse in the Methods section, please refer to Lines 642-691 (for denoising) and Lines 878-883 (for super-resolution) in the revised manuscript.

3. The github link (<https://github.com/Luchixiang/EMDiffuse>) given in the manuscript does currently not work. Hence I could not evaluate the state of the software itself, since it was not provided with the supplementary material either. Given that the practical impact of the method could be very significant, it would be of high importance to provide user-friendly and well documented software. The authors should submit a version of their software with the next submission (either by uploading it to github or adding it to the supplementary material), so that this can be reviewed.

RESPONSE: Thanks for your comments. We have made the code public (<https://github.com/Luchixiang/EMDiffuse>)

Reviewer #3 (Remarks to the Author):

This manuscript introduced algorithms designed to enhance EM and vEM capabilities, EMDiffuse, leveraging the cutting-edge image generation diffusion model. EMDiffuse demonstrates superior denoising and super-resolution performance, generates realistic predictions without unwarranted smoothness. EMDiffuse perhaps has a powerful tool in reducing imaging time for vEM dataset. The results are significant and interesting. Figures are sufficient. This reviewer would like to comments some.

RESPONSE: We thank the reviewer for acknowledging the strength of the manuscript.

“Resolution” is a key word in this manuscript, so it should be defined at beginning of the main text for reader’s better understanding. How to measure “resolution” should be described concretely in methods, though it is roughly indicated in lines 689-693 and cited reference 39.

RESPONSE: Thank you for the comments. We have now elucidated the term "resolution" in the main text, please refer to Lines 66-68. In the Methods section, we’ve expanded on the technique used to measure resolution, please refer to Lines 754-768 in the revised manuscript.

I understand that results related to the Figures 1 and 2 may be based on their original serial section-based tomography vEM datasets obtained with either FE-SEM, and Figures 3 and 4 may be based on open source vEM datasets obtained with FIB-SEM, which are indicated in line 922-933 at the end of Methods. It is unclear that which dataset was used to obtain each result, such as EMDiffuse-n, EMDiffuse-n training, EMDiffuse-n transfer learning, EMDiffuse-r and so on, shown in the Figure panels. These should be clearly indicated in main text or figure legends, with datasets conditions: thickness of the serial section, pixel size, FIB-SEM, FE-SEM, open source or home made.

RESPONSE: Thank you for your feedback. We have included the details of datasets used in each task in the main texts or figure legends.

Specifically, for the denoising task, EMDiffuse, as well as the baselines, were trained and tested on a mouse brain cortex dataset acquired in-house with a 3.3 nm pixel size. For transfer learning, we fine-tuned and tested EMDiffuse on FE-SEM datasets from three different mouse tissues (*i.e.*, liver, heart, bone marrow) with a pixel size of 4.4 nm, and cultured HeLa cell images with a pixel size of 2.1 nm. For the super-resolution task, a mouse brain cortex dataset composed of paired noisy input with a pixel size of 6.6 nm and ground truth with a pixel size of 3.3 nm was obtained to train EMDiffuse-r and other baselines. Images were acquired in-house using a backscatter detector of FEI Verios SEM. For vEMDiffuse-i and vEMDiffuse-a experiments, we used open-source datasets. All the original datasets used for the training in this manuscript are now made public (<https://zenodo.org/records/10205819>).

In line 485 – 486, “Sections of mouse tissues, measuring 500 nm in thickness, were mounted onto silicon wafers.” This sentence is ambiguous. Whether it means either “serial sections of mouse tissues of 500 nm thick sections were mounted onto silicon wafers”, or “serial sections of mouse tissues of 50 nm thick section, measuring 500 nm in total thickness, were mounted onto silicon wafers”. It should be clearly described.

RESPONSE: Thank you for pointing out the ambiguity in our description. For our in-house generated datasets for the denoising and super-resolution tasks, we used semi-thin sections of 500 nm for imaging with an FE-SEM, and no serial sections were used in these experiments. We have revised the sentence in Line 508.

The GitHub page (<https://github.com/Luchixiang/EMDiffuse>) is currently inaccessible, rendering it impossible to assess the effectiveness of this technology for EM data in general. It is imperative that the GitHub page be made accessible, and the associated code should be made available.

RESPONSE: Thanks for your comments. We have made the code public (<https://github.com/Luchixiang/EMDiffuse>).

While I find this technique to be highly effective for denoising and super resolution of typical structures, I have concerns regarding potential artifacts in the case of rare structures. Specifically:

- When a new sample is trained with limited data using transfer learning.

- When the size of the structure aligns closely with the z step of the original image. Could you discuss the possibility of artifacts being introduced in these scenarios? If such a possibility exists, please elaborate on the nature of these potential artifacts.

RESPONSE: Thank you for your inquiry regarding potential artifacts. We have carefully considered these scenarios.

Transfer learning with limited training samples: Artifacts in this context are generally a consequence of a domain gap—disparities between the training and test datasets. These artifacts may arise when the model encounters previously unseen structures. For example, in our mouse bone marrow dataset, some artifacts were generated by the model trained on the mouse brain cortex dataset before fine-tuning (**Fig. 1e**). However, these artifacts were effectively flagged out by our model's uncertainty estimation, which returned high uncertainty values (exceeding the threshold of 0.12), signaling potential reliability issues. Furthermore, by freezing the encoder weight during transfer learning, we showed that the performance of the pre-trained EMDiffuse-n model could be improved with only a single data pair of 1768×1768 pixels from new domains (Extended Data Fig. 7). After fine-tuning, the predictions showed low uncertainty values and high reliability (**Fig. 1e**).

Structure size similar to z-step: There is indeed a risk that artifacts may occur with structures whose size is similar to or smaller than the z-step. If the axial resolution is insufficient, small structures may go undetected. We have tested the capability of vEMDiffuse-i to generate isotropic volumes from anisotropic volumes of different axial resolutions from 48 nm to 96 nm (Extended Data Fig. 20, *see below*). In our experiments where the z-step was large, for example, downsampled volumes with a 96 nm axial resolution from the Openorganelle mouse liver dataset, we observed the occurrence of artifacts (Extended Data Fig. 20a). vEMDiffuse-i successfully generated structures of major organelles, however, failed to accurately generate fine details of the endoplasmic reticulum and mitochondria cristae (Extended Data Fig. 20a), showing reduced FSIM metrics (Extended Data Fig. 20c). With a 64 nm axial resolution, vEMDiffuse-i was able to reconstruct most of the fine details of all organelles. Here, our uncertainty estimation played an essential role in evaluating the reliability of each generated layer (Extended Data Fig. 20b). It quantitatively indicated the confidence level in each reconstructed layer, enabling users to discern the likelihood of potential artifacts or inaccuracies in the generated images. We have revised the manuscript accordingly. Please refer to Lines 250-258 and the new Extended Data Figure 20 in the revised manuscript.

a**b****c**
Extended Data Fig. 20 Reliability of vEMDiffuse-i generated volumes from anisotropic volumes with different axial resolutions.

a. Example YZ view of the isotropic volume (8 nm x 8 nm x 8 nm resolution), vEMDiffuse-i generated volume from 8 nm x 8 nm x 48 nm resolution, 8 nm x 8 nm x 56 nm resolution, 8 nm x 8 nm x 64 nm, 8 nm x 8 nm x 72 nm, from 8 nm x 8 nm x 96 nm resolution anisotropic volumes downsampled from the Openorganelle mouse liver dataset (jrc_mus-liver). The IoU scores of ER and mitochondria segmentation results are shown in the bottom right corner. Scale bar, 0.3 μ m.

b. The violin plots of FSIM metrics of five generated volumes.

c. The violin plots of uncertainty values of five generated volumes.

Since many users will conduct 3D segmentation post-application of this method, it is crucial to validate and demonstrate the efficacy of this method for 3D segmentation. I am particularly interested in understanding its effectiveness when applied to SSTEM data. While it may be highly effective for mitochondrial segmentation, how about its application to mitochondria-associated membranes? Addressing these questions and discussing the effectiveness and limitations of this method in these contexts is essential.

RESPONSE: Thank you for your comment. To validate the application of vEMDiffuse-a on SSTEM datasets for fine details of organelles, such as mitochondria-associated membranes, we conducted experiments of mitochondria membrane reconstruction on downsampled Openorganelle mouse liver volumes (*jrc_mus-liver*). A 3D U-Net was trained on the isotropic volume of the liver dataset and its corresponding mitochondria membrane ground truth mask. This trained model was subsequently applied to three different volumes: an interpolated volume using ITK-cubic interpolation, a volume generated by vEMDiffuse-a, and the ground truth isotropic volume. The 3D rendering (Extended Data Fig. 24, *see below*) shows that the segmentation masks of the vEMDiffuse-a generated volume significantly improve the reconstruction of mitochondria membranes compared to anisotropic masks and the segmentation masks of the interpolated volume, further demonstrating that vEMDiffuse-a's capability to facilitate ultrastructural reconstruction and analysis. As discussed above, the capability of vEMDiffuse to accurately generate fine details of organelles will be limited by the axial resolution of the vEM datasets. Nevertheless, our uncertainty estimation will assess the reliability of generated volumes. Please refer to Lines 297-299 and the new Extended Data Figure 24 in the revised manuscript.

Extended Data Fig. 24 vEMDiffuse-a enhances mitochondria membrane reconstruction of the Openorganelle mouse liver dataset (*jrc_mus-liver*).

Comparison of the 3D mitochondria membrane reconstruction results between the anisotropic mask (8 nm x 8 nm x 48 nm resolution), segmentation masks of interpolated volume (ITK-cubic), segmentation masks of vEMDiffuse-a generated volume and segmentation masks of the ground truth isotropic volume (8 nm x 8 nm x 8 nm resolution). The IoU score of segmentation masks is calculated against the segmentation mask of the ground truth isotropic volume to avoid the potential influence of the segmentation model.

One of the motivations for applying this method is automating neurite tracing. If possible, could you provide insights into how the accuracy of neurite auto-tracing improves with and without the

application of this method? Additionally, I would appreciate understanding the effectiveness of this method for SSTEM data with large z steps.

RESPONSE: Thank you for your interest in the applications of our method for neurite tracing and its effectiveness on SSTEM data with large z steps. We have now included additional results in the revised manuscript regarding vEMDiffuse's application in neurite tracing in Lines 317-322 and methods used for neurite tracing in the Methods section Lines 1018-1029.

Specifically, the application of vEMDiffuse-a to neurite tracing was demonstrated on the downsampled MANC dataset with an 8 nm x 8 nm x 48 nm resolution, using a Flood-Filling Network (FFN) model. Indicated by high IoU scores, vEMDiffuse-a generated volume (Extended Data Fig. 27a) showed improved capabilities in preserving fine neurite details and capturing complex branching

structures, when compared to the over-smoothed anisotropic masks and the segmentation masks of interpolated volume (Extended Data Fig. 27b).

As discussed above, we have tested the capability of vEMDiffuse to generate isotropic volumes from downsampled volumes of different axial resolutions, and a new Extended Data Fig. 20 is now included in the manuscript.

The manuscript mentions SEM imaging magnification but lacks information on the pixel size (nm/pixel), which is now standard manner to describe a magnification of electron micrograph. Please include details about the pixel size.

RESPONSE: Thanks for your comments. We have added the pixel size information in the Methods section, main text and figure legends for clarity.

Acceleration voltage is a matter for scanning electron micrograph to collect tissue image signals from the depth. It should be described in Method.

RESPONSE: Thank you for highlighting the importance of the acceleration voltage when discussing scanning electron micrographs. We have included this information in the Methods section. The acceleration voltage utilized for all our in-house generated datasets was set at 2 kV.

The authors compare their proposed methods, EMDiffuse-n and EMDiffuse-r, with state-of-the-art methods such as CARE in Figures 1 and 2. To clarify the effectiveness of vEMDiffuse-i for vEM image segmentation, it would be helpful to provide the following:

- The intersection over union (IoU) value of segmentation using "Interpolation" in Figure 3e.
- A segmentation example in Figure 3f.

RESPONSE: Thank you for highlighting the need for clarity regarding our segmentation comparison. We have included the IoU score of the segmentation masks of interpolated volume (as baseline) and the segmentation masks of ground truth isotropic volume (as upper bound) in **Fig. 3e**, *see above*, and included a reconstruction example in **Fig. 3f**, *see above*. The vEMDiffuse-i generated volume achieved a similar IoU score as isotropic volume (**Fig. 3e**), indicating precise organelle prediction by vEMDiffuse-i. In contrast, the IoU scores of the anisotropic volume and interpolated volume were low (**Fig. 3e**). The 3D rendering visualizations (**Fig. 3f**) demonstrated the accurate reconstruction of organelle ultrastructure in vEMDiffuse-i generated volume. In contrast, anisotropic volume and interpolated volume failed the task, which represents a major limitation of serial section-based vEM techniques. Accordingly, we have updated the manuscript and figures. Please refer to new **Fig. 3e** and **Fig. 3f**, and Lines 238-248 and Lines 997-1016 of the manuscript.

The authors use channel embedding to generate images in vEMDiffuse-i. Is there any relationship between the number of interpolated images R and the segmentation accuracy?

The users of vEMDiffuse-i would be appreciated if you provide the relationship. Is there a risk of overfitting if R is too large?

RESPONSE: Thank you for this comment. As discussed above, we have included new experiments assessing the relationship between axial resolution, or the number of interpolated images R, and the segmentation accuracy (IoU scores) and a new Extended Data Fig. 20 is now included in the manuscript.

With regard to overfitting, it is important to clarify that R represents the interpolation factor, not the complexity of the model itself. Therefore, the risk of overfitting is not inherently tied to the value of R. The observed limitations are more attributable to information loss due to low axial resolution than overfitting due to a large R.

Non-machine learning experts would appreciate a more detailed explanation of the difficulty assessment map (L573), including equations, in the main text.

RESPONSE: Thank you for your insightful feedback. we have explained the difficulty-aware loss function in the main text Lines 100-106.

In Extended Data Figure 16c, the input appears to be the same image as the input for Case 1 in Figure 16b. Is this correct?

RESPONSE: Thanks for your comments. We have corrected Extended Data Figure 16c (Extended Data Fig. 18c in the revised manuscript) to the correct input image.

The loss function is stated to follow Ho et al. (2020) in L565. However, the equation in L569 specifies that the noise is a function of time t . Is it correct to understand that $\epsilon_t = \mathcal{N}(0, 1)$? If so, for non-expert readers in machine learning, it would be appreciated if t were removed.

RESPONSE: Thanks for your feedback. We have revised the equation to improve clarity by removing t . Below is the updated formula:

$$\begin{aligned} L_{\text{simple}} &= \mathbb{E}_{\mathbf{x}_0, t \sim [1, T], \epsilon} \|\epsilon - \epsilon_\theta(\mathbf{x}_t, t, \mathbf{c}_r)\|^2 \\ &= \mathbb{E}_{\mathbf{x}_0, t \sim [1, T], \epsilon} \|\epsilon - \epsilon_\theta(\sqrt{\bar{\alpha}_t} \mathbf{x}_0 + \sqrt{1 - \bar{\alpha}_t} \epsilon, t, \mathbf{c}_r)\|^2. \end{aligned}$$

where $\epsilon \sim \mathcal{N}(0, 1)$.

Non-expert readers in machine learning would appreciate it if you explicitly stated the objective functions for EMDiffuse-n and EMDiffuse-r, as is done for vEMDiffuse-i in L826.

RESPONSE: Thank you for your suggestions. We have included the training objective functions for EMDiffuse-n and EMDiffuse-r in the revised manuscript Line 691 and Line 863. For EMDiffuse-n, given paired high-quality image \mathbf{x}^{gt} and noisy image \mathbf{c}_n , the training objective was:

$$L = \frac{1}{N} \sum_i \left(\frac{|\epsilon - \epsilon_\theta(\sqrt{\bar{\alpha}_t} \mathbf{x}_0^{gt} + \sqrt{1 - \bar{\alpha}_t} \epsilon, t, \mathbf{c}_n)|^2}{2\varphi_i^2} + \log(\varphi_i) \right)$$

φ_i denoted the difficulty map predicted by the model.

The loss function for EMDiffuse-r was:

$$L = \frac{1}{N} \sum_i \left(\frac{|\epsilon - \epsilon_\theta(\sqrt{\bar{\alpha}_t} \mathbf{x}_0^{gt} + \sqrt{1 - \bar{\alpha}_t} \epsilon, t, \mathbf{c}_n)|^2}{2\varphi_i^2} + \log(\varphi_i) \right)$$

\mathbf{x}^{gt} was high-resolution image and \mathbf{c}_n was rescaled low-resolution image.

In Extended Data Figure 4, the PSNR values are higher for 6Mean than for Single. This suggests that PSNR is a good metric for evaluating the quality of denoised images. Is this correct?

RESPONSE: Thank you for your comment. PSNR is known to favor blurry images. However, in EM applications, blur-free high-resolution images are preferred. Given that PSNR does not account for

blurring artifacts, it is not a suitable metric for evaluating the quality of denoised EM images (Extended Data Fig. 3c and Extended Data Fig. 4a).

Reviewer #4 (Remarks to the Author):

EMDiffuse: a diffusion-based deep learning method augmenting ultrastructural imaging and volume electron microscopy

Summary:

=====
This manuscript proposes ways of applying diffusion models on Electron Microscopy (EM) and volume EM (vEM) tasks. The authors claim that by applying their method, EM and vEM tasks can significantly increase the imaging speed without compromising on the quality of acquired data and also enable the isotropic imaging of large biological volumes.

In particular, the paper shows the application of their diffusion model setup on (i) denoising, (ii) super-resolution (SR), and (iii) volume reconstruction without using isotropic training data. The manuscript demonstrates the applicability of the proposed method by performing experiments on public data and introduces a self-assessment method (based on uncertainty readouts) that users can utilize to judge the quality and reliability of obtained results.

The proposed method (the DiReP Network) relies on a combination of existing diffusion model training strategies, which to the best of our knowledge are for the first time applied to the presented tasks and data.

In terms of validation of their results, the manuscript notes that the PSNR metric should not be used, since it favors over-smoothed predictions. As an alternative, the authors propose to use FSIM (Feature Similarity Index), BRISQUE (Blind/Referenceless Image Spatial Quality Evaluator), and LPIPS (Learned Perceptual Image Patch Similarity). With respect to those metrics, the proposed method outperforms several baselines, i.e. CARE, RCAN PSSR, but when suitable also N2N and N2V.

All presented approaches (quickly summarize below) are based on well established Diffusion Model training and inference procedures. In particular, solutions for the following four tasks are presented:

1. Image Denoising:

A variant of DiReP Network, called EMDiffuse-n, is used in their denoising pipeline. During inference, an input is fed to the trained DiReP network and two predictions are sampled from the diffusion posterior. Those two samples are averaged (pixel-wise) to produce the final prediction. For quality assessment FSIM (compared against imaged ground truth data) was used. The presented method is reported to improve 30% over other baselines.

Additionally, EMDiffuse-n also outputs an uncertainty map, i.e. the pixel-wise difference between the 2 above mentioned samples (the std if more than 2 samples are used). The uncertainty map can be used to reject predictions whenever an uncertainty threshold (in their case 0.12) in the average of all pixel-wise uncertainties is superseded.

The authors also show that EMDiffuse-n can be used in a transfer learning setting, where a model trained on mouse brain cortex data was finetuned to denoise data from mouse liver, heart, or bone marrow data. Finetuning on a single image reduces the uncertainty score mentioned above considerably.

2. Image Super-Resolution:

The paper then applied a variant of the same diffusion model pipeline called the EMDiffuse-r to

perform a SR task. First, the lower resolution images gets resized to its desired dimensions, then a trained DiReP network is used to predict the higher resolution result. Again, the mean of two samples is the final prediction.

The authors demonstrated superior performance in restoring fine structures when compared with other baseline methods, i.e. CARE, RCAN, and PSSR.

Additionally, the authors calculate the uncertainty for the predictions and show that the outputs are reliable.

Finally, the authors show that the model trained on mouse cortex data is able to generalize to other datasets, again finetuned on a single additional image, i.e. from the mouse heart, liver, and bone marrow data used throughout the presented work.

3. Volume reconstruction with isotropic vEM data:

The manuscript proposes to reconstruct the axial volume of a section using the first and n-th slice of the section using another variant called the vEMDiffuse-i.

To this end, the model takes two consecutive raw data slices and a suitably encoded “channel embedding”, i.e. the index of the slice $[1,R]$ between the two given raw slices. The uncertainty calculation method previously mentioned is also used here.

The authors use no direct metrics to asses the reached performance, but go on showing that on a downstream segmentation task, evaluated in terms of IoU, their predicted isotropic volume outperforms the same segmentation applied on the raw (anisotropic) data.

4. Volume reconstruction with anisotropic vEM data:

While for vEMDiffuse-i isotropic data is required during training, the authors finally introduce a variation called vEMDiffuse-a, which can operate on anisotropic training data alone. This is accomplished by training on XZ slices instead of XY slices, the advantage being, that in the remaining Y direction, we have isotropic resolution and hence the required training data to replicate the vEMDiffuse-i procedure (on a surrogate XY interpolation task). Once trained, the trained model is then applied directly to XY slices, thereby predicting the missing data to make the original volume isotropic.

As before, the authors use no direct metrics to asses the reached performance, but go on showing that on a downstream segmentation task, evaluated in terms of IoU, their predicted isotropic volume outperforms the same segmentation applied on the raw (anisotropic) data.

Strengths:

=====

1. The manuscript addresses an important area in EM and vEM reconstruction and has the potential to find broad application.
2. The authors propose a single diffusion model-based setup that is applicable to multiple practical tasks.
3. The authors provide a way for users to assess the uncertainty in the predicted data.
4. The authors show that their method is good at transfer-learning to other (yet somewhat similar) data.
5. The authors compare their method against several relevant baselines and show that their method outperforms them.

6. The authors also use suitable perceptual metrics such as FSIM, BRISQUE, and LPIPS for measuring the performance of their method. It is regrettable though, that PSNR and/or SSIM evaluations are not at all available in the manuscript.

Weaknesses/Open Questions:

1. The authors propose a method based on existing DL technologies that promises to be of great utility for life-scientists that acquire EM data. In the light of this setup, the reviewer thinks it is required to make the full source of the method publicly available. The link given in the section “Code Availability” leads to an error 404, making it impossible for this reviewer to evaluate the quality of the codebase to be made available. I consider this to be the biggest problem to propose acceptance of this manuscript.

RESPONSE: Thanks for your comments. We have made the code public (<https://github.com/Luchixiang/EMDiffuse>).

2. From a purely DL-engineering perspective, it is very regrettable that there are no ablation studies presented. The reviewer is certain that multiple slightly different approaches have been tried and it would be very interesting to hear how much worse they have performed. One example: instead of interpolating R-fold (in vEMDiffuse models), the “channel embedding” could encode the relative position between the two given slices, which would even make it possible to transfer learn between datasets that obey a different value for R. Or: how much worse are results without the difficulty-aware loss term? In the CV field, such ablations are considered essential to tell a complete story about a proposed method. The reviewer will leave it to the editor to decide how important such information is for a venue like Nature Communication.

RESPONSE: Thanks for highlighting the importance of the ablation study. Indeed, we have tested and optimized approaches for the development of EMDiffuse, and we have included more details of the ablation studies here and a new Supplementary Table in the revised manuscript.

Encoding the relative distance into channel embedding: To generate isotropic volumes in vEM, the distance between two layers is the same as the pixel size in the XY plane of the dataset. EMDiffuse can learn these distances from input images. To quantitatively validate this, we conducted two additional experiments.

Experiment 1: We compared encoding the relative distance with the R-fold interpolation approach on the generation of isotropic volumes. We trained and tested the two models on the Openorganelle mouse kidney dataset (jur_mus-kidney), which showed comparable performances.

	FSIM	Resolution Ratio	LPIPS
Interpolating R	0.974	1.024	0.094
Relative distance	0.973	1.025	0.093

Experiment 2: We assessed the transferability of encoding the relative distance and R-fold interpolation approaches. We applied the trained vEMDiffuse-i model from the Openorganelle mouse kidney dataset (8 nm x 8 nm x 48 nm resolution) to the Openorganelle mouse liver dataset (jrc_mus-liver, 8 nm x 8 nm x 64 nm resolution) with minimal fine-tuning (100 epochs), yielding similarly close performance metrics between these two approaches.

	FSIM	Resolution Ratio	LPIPS
Interpolating R	0.958	1.011	0.091
Relative distance	0.959	1.010	0.090

Overall, encoding relative distance and R-fold interpolating approaches have similar performances in isotropic volume generation and transferability.

Ablation study on EMDiffuse-n components: We have conducted ablation studies that examine the contributions of global attention, difficulty-aware loss, and Gaussian blur data augmentation to our model's performance. The study highlights that each component individually and collectively enhances the model's performance. The results of the ablation study have been added as Supplementary Table 1 in the revised manuscript.

	FSIM	LPIPS	Resolution Ratio
DDPM	0.84	0.14	0.96
DDPM + global attention	0.86	0.11	0.98
DDPM + difficulty aware loss	0.87	0.13	0.97
DDPM + global attention + difficulty aware loss	0.88	0.11	0.99
EMDiffuse (DDPM + global attention + difficulty aware loss + Gaussian blur)	0.89	0.10	1.01

Supplementary Table 1: Ablation study of EMDiffuse-n on mouse brain cortex denoise dataset. For each study, four experiments were repeated, with the mean score reported.

3. The authors rely on an uncertainty metric threshold that is determined on some given data. Will 0.12 now work for all data users would like to use? The reviewer thinks this is not the case, posing the question how future users should go about evaluating a suitable threshold for their own data. Being more explicit about this in the manuscript seems desirable.

RESPONSE: Thank you for your comment regarding the generalizability of the uncertainty metric threshold. We believe that our threshold is robust across various datasets. The basis of our uncertainty estimation is that the prediction becomes increasingly trustworthy if multiple outputs agree. This model-centric approach is less affected by the inherent variability of different datasets. Also, given that the standard deviation calculation was normalized across a [0, 1] scale using min-max normalization, contrast variations do not affect this metric. Furthermore, in our experiments, this threshold was tested across various datasets, including both vEM and 2D EM data, and the 0.12 threshold consistently identified reliable predictions (Extended Data Fig. 20, *see above*).

However, we acknowledge that different users or applications may have different standards of trustworthy predictions, a detailed methodology for determining the uncertainty threshold was also included in the Methods section (Lines 817-825). Future users could generate their custom threshold if desired for specific applications.

4. In Figure 3(e), quantitative results are only shown for the anisotropic volume and the prediction using vEMDiffuse-i. The reviewer would appreciate to also see the same quantification for GT (as an upper bound) and the interpolated volume (as a baseline).

RESPONSE: Thank you for this comment. We have revised **Fig. 3e**, *see above* and added the IoU scores of segmentation results of the interpolated volume (as baseline) and ground truth isotropic volume (as upper bound). The vEMDiffuse-i generated volume achieved a similar IoU score as isotropic volume (**Fig. 3e**), indicating precise organelle prediction by vEMDiffuse-i. In contrast, the IoU scores of the anisotropic volume and interpolated volume were low (**Fig. 3e**). The 3D rendering visualizations (**Fig. 3f**) demonstrated the accurate reconstruction of organelle ultrastructure in

vEMDiffuse-i generated volume. In contrast, anisotropic volume and interpolated volume failed the task, which represents a major limitation of serial section-based vEM techniques. Accordingly, we have updated the manuscript and figures. Please refer to new **Fig. 3e** and **Fig. 3f**, and Lines 238-248 and Lines 997-1016 of the manuscript.

5. The paper deals with various mouse datasets, presumably being more similar than other samples where cells and organelles have much diverging sizes and the subcellular milieu can even contain structures not present in mice. It would be interesting to see how trained models would transfer and how much finetuning would improve predictions. Side note: the fact that vEMDiffuse-a works well indicates that a trained model is relatively robust to quite drastic changes in data appearance. Still, a more quantitative approach would be desirable.

RESPONSE: We appreciate this suggestion. We have now included an additional test dataset of cultured human HeLa cells to further validate the transferability of our model. EMDiffuse-n and EMDiffuse-r trained on mouse brain cortex dataset showed proficiency in denoising and super-resolving cultured HeLa cell images (**Fig. 1e**, Extended Data Fig. 10) before fine-tuning and after fine-tuned with one pair of images.

We have quantitatively evaluated the performance of the EMDiffuse-n model when transferring from the mouse brain cortex dataset to four datasets: mouse liver, heart, bone marrow and HeLa cells using different numbers of fine-tuning images. The results showed that the performance of the pre-trained EMDiffuse-n model could be easily improved by fine-tuning the decoder with only a single data pair of 1768×1768 pixels from new domains (Extended Data Fig. 7b).

6. Reporting an improvement in percent (e.g. the reported 30% in line 130) was more confusing to the reviewer than helpful. 30% in what

RESPONSE: Thank you for your comment. We have revised the statements in the improvement in percent in the abstract and results section.

7. The image quality assessment metric LPIPS, the pre-trained neural network is usually trained with natural images. In this case, it is unclear how this would affect the performance reporting of the metric.

RESPONSE: Thank you for your comment. LPIPS has been designed to emulate human visual perception by assessing image quality based on learned high-level features. Although its training on natural images might initially seem to limit its direct applicability to bioimages, the core perceptual factors it measures, such as contrast, distortion, and blurring, are indeed relevant across image domains. In EM imaging, it is often these very attributes—the contrast between structures, the presence of noise and distortion, blurring, artifacts, and the clarity of fine details—that critically affect the EM images for ultrastructure analysis and scientific discovery. Thus, we believe that LPIPS is also suitable for measuring the quality of EM images. Notably, the distortion types included in the LPIPS training

	Patch 0	Reference	Patch 1	Patch 0	Reference	Patch 1
						PSNR	24.8 ✓		23.1	31.6 ✓		29.4
LPIPS	0.36		0.25 ✓	0.30		0.21 ✓
Human			✓			✓

Evaluation of the use of LPIPS and PSNR on EM images.

datasets, such as blurring, noise, superresolution, and frame interpolation, are also congruent to our denoising, super-resolution and vEM reconstruction tasks. To consolidate our assumption, we also show two examples (*see below*). In each case, PSNR disagrees with human judgments, while LPIPS aligns well with humans.

To ensure a comprehensive evaluation, we complemented LPIPS with FSIM and resolution ratio metrics. FSIM offers an additional measure of structural fidelity, focusing on significant features and their phase congruency. The resolution ratio further allows us to quantify the improvement in resolution that our methodology provides. This multi-metric approach ensures that our assessment captures both the perceptual quality and ultrastructural accuracy of generated images.

8. Not at all reporting on classical and widely used metrics such as PSNR or SSIM is regrettable.

RESPONSE: Thank you for your comment. In the revised manuscript, we have included the comparison between EMDiffuse and other methods using the PSNR metric (Extended Data Fig. 4).

9. The text is overall well written, but slightly deteriorates in later sections (and in more technical sections). A thorough pass over all text sections is highly recommended.

RESPONSE: Thank you for your feedback. We have revised the manuscript and particularly the Methods section to enhance the clarity of the manuscript.

10. Please correct derivation to deviation on line 729

RESPONSE: We have corrected "derivation" to "deviation".

11. Please correct LIPSI to LPIPS on line 748.

RESPONSE: Thank you for pointing out this typo. It has now been corrected to 'LPIPS'.

REVIEWER COMMENTS

Reviewer #1 (Remarks to the Author):

Thank you for releasing the data and software. (I believe that the 3D-SRU-Net code may have been committed in a non-functional state, it seems that a and b are confused in the main call and that there are two competing scaling factors 6 and 8?) Also thanks for adding additional references and comparison. To me, those changes make the paper stronger and support that this work is interesting and contributes to the field. I really like this work.

I am still concerned about some presentations in the main paper (the methods part is straight forward and easy to read). Particularly the qualitative statements about the high resolution that the diffusion models generate requires some attention (see below), e.g. 99-100 "to produce the denoised image." is somehow misleading, because the strength of the method is that it learns to produce many plausible denoised images. This comes back in several other places.

This work uses 2D architectures to do 2D denoising and slice in-painting to improve 3D resolution. The manuscript talks a lot about 3D, yet never about that the architecture being 2D. The fact that the architectures are 2D transpires on the side of patch sizes discussed with two numbers. I find it interesting that a 2D method achieves results superior to a 3D method which should be discussed?

74--76 CARE implements a reliability assessment, please discuss or rephrase
87 propose to change to: "[...]the inherent ability of diffusion models to generate an arbitrary number of diverse samples..."

111 and below do not address appropriately that each individual generated image is just a sample from the distribution of plausible images given the noisy data. The average of 2 may optimize the metrics used for comparison but has no other meaning (see below).

Extended Data Fig. 3 I think it would be less confusing if the arrow "intricate Structure" would point to the same location in all images, including GT. The input crop is not aligned

with the output crops and GT.

Language: "EMDiffuse can adopt the mean result of different testing times prediction as the ultimate prediction to increase fidelity" what does this mean? What even is "fidelity"? I suggest to stick to the facts, e.g.: "EMDiffuse generates plausible super-resolution images with convincing high-resolution details. However, super-resolution from noisy images is an ill-posed problem, and each super-resolution output is just one possible solution to this problem. To account for this, we can generate an arbitrary number of outputs for the same input from different initial noise. One can use these outputs to assess the variance at each pixel, or the variance of interpretations of the data (e.g. number of pre-synaptic vesicles)."

Following up on this: Averaging just two predictions optimizes your error metrics but feels wrong. It means that you do not create a statistically meaningful average nor account correctly for the distribution of your solutions. Creating the mean of all plausible solutions does not do justice to the fact that all solutions are likely plausible themselves. E.g. in the example in EDFig. 3, the network has learned that the noisy input near the synaptic cleft indicates the presence of a number of vesicles. Each output has a number of vesicles and this number seems highly consistent. Averaging any number of outputs will smear vesicles away, taking any output as the one true solution is similarly incorrect. This is interesting though, because it enables you to e.g. count vesicles in all outputs, and present the average count and variance as a result, since all outputs agree that there are vesicles. Averaging any number of outputs compromises this and is therefore much less interesting than aggregating over the structures embedded in the learned complex distribution that the network allows you to sample.

The figure title "Optimization of prediction generation methods using multiple outputs enhances reliability while reserving resolution" is meaningless because it does not say any of those interesting things and makes generic qualitative statements about "reliability"(?) and "resolution", none of which is supported by this figure.

129 "preserved the high resolution" is IMHO not a correct interpretation. As above, "it generates outputs with high resolution details". The variance of those high-resolution

details can be assessed because an arbitrary number of outputs can be generated. This is where the diffusion model based approach excels. The other methods have all been trained to generate the pixel-based average which is inferior.

For the per-pixel uncertainty estimation, I believe that images with diverse contrast levels will have lower uncertainty values for lower contrast regions than for higher contrast regions. This makes a single global threshold difficult to interpret. I find this worth mentioning. I also think that it could be addressed by some form of local contrast normalization ([CL]AHE?) which would boost edges?

Reviewer #1 (Remarks on code availability):

Partial review, there seem to be some bugs here and there, but that can be fixed. I have not run the experiments to reproduce the results because that would have required me to do some amount of plumbing.

Reviewer #2 (Remarks to the Author):

The review addresses my concerns regarding the manuscript. It now includes the references to prior work and extends experiments and method descriptions significantly to explain the comparison to these methods.

The authors also make the code publicly available. The code is currently in a "research code" state. It thus likely enables replication of the results, but is not in a sufficient advanced state to be used by life scientists without significant expertise. See the section on code review below for details.

Overall I can recommend this manuscript for publication as it shows the potential of using diffusion models for EM image reconstruction tasks by convincingly showing that these methods outperform the previous state-of-the-art. However, I will add that I think it that it is unlikely that the tool will gain broad usage in the volume EM community in its current form due to the lack of a user-friendly implementation.

Reviewer #2 (Remarks on code availability):

I have reviewed the software at <https://github.com/Luchixiang/EMDiffuse> and tried to

install it on my laptop in order to test it (see below). Overall, the software is comprised of python scripts without any user interface to help applying it to new data. It has a README that explains how to use the software; however, it is not written well (in several cases sentences are incorrect).

It is thus in a state where users with significant computational experience and specialized hardware (access to a GPU) can run the tool. However, this means the tool will be difficult to use for many researchers in the volume EM community, who are biologists by training and rely on user-friendly software with a GUI.

In order to build such user friendly software I can recommend looking into bioimage.io (<https://bioimage.io/#/>) and/or napari (<https://napari.org/stable/>). While I don't think that this is a prerequisite for publication, it would increase the potential impact of the tool significantly.

As for my own attempts with the software: I tried to run inference for EM denoising. Here is a step by step explanation with comments that may also help to improve it:

- I installed the software via conda + pip as described. This worked, however the 'tqdm' dependency was missing. This was easy to fix (for me), but could already be a hurdle for computationally inexperienced users.
- I tried to run inference according to <https://github.com/Luchixiang/EMDiffuse#inference>.
- Downloading the model and example dataset worked as described.
- The cropping step worked.
- Running the actual inference step fails on my laptop because I don't have a GPU. It's unclear why you hard-code the model to run on the GPU here. Normally inference works fine on the CPU (training of course is a different story), although taking longer, and implementing GPU and CPU support is trivial in PyTorch. Note that a lot of potential users may not have direct access to a GPU. Even if they do, the machine with the GPU may not have support for visualization so training with GPU and inference with CPU may be a common use-case.

At this point I stopped because setting up the software on our cluster with GPU support would be too much effort for me right now.

Reviewer #3 (Remarks to the Author):

Thank you for revising manuscript. The authors have adequately addressed the reviewer's comments, and overall, the manuscript is well-crafted. However, please consider followings.

Thank you for providing a thorough elucidation of "transfer learning with limited training samples" and "structure size similar to z-step" in the context of artifacts.

Recognizing the broader implications of your technology beyond the machine learning expert community, it becomes crucial to tailor your explanations for a more diverse readership, including non-experts. In relation to "transfer learning with limited training samples," I encourage you to expound on the concept of domain gap, elucidating its significance and emphasizing the necessity of incorporating diverse structures to mitigate the impact of the domain gap in the learning or transfer learning process. Such clarifications would enhance the accessibility of your work to a wider readership.

With regards to "structure size similar to z-step," I appreciate the inclusion of the extended Figure 20, which effectively elucidates the application of vEMDiffuse-i to images featuring ERs and mitochondria. However, the absence of information related to synapses and synaptic vesicles creates a noticeable void in the narrative. Readers are likely eager to comprehend the effects on these structures. Hence, I propose expanding the discussion to include the outcomes of applying vEMDiffuse-i to synapses and synaptic vesicles. Additionally, shedding light on the results of employing vEMDiffuse-a on these structures would enhance the overall comprehensiveness of your findings.

In the interest of transparency, it is crucial to address any limitations associated with your technique. If there are constraints or shortcomings, kindly elucidate them to provide a balanced perspective on the applicability and scope of your approach."

Thank you for your attention to these matters, and I look forward to seeing these refinements in your manuscript.

Reviewer #4 (Remarks to the Author):

The authors did a good job taking our reviewer feedback into account while preparing their resubmission. The manuscript did improve in many respects and I have only a few but rather fundamental remaining issues to point out.

I believe that after addressing those, the manuscript merits publication. I hope, though, the authors will take my feedback as serious as I believe it is.

Major Points of Criticism

1. The current manuscript and supplementary information draws a misleading picture about what PSNR measures and why it should or should not be used.

After reading the authors arguments for not using PSNR more prominently, I believe the authors are either not fully aware of what I will describe below or are not educating readers in an appropriate way in the current manuscript.

- Scientific image data is not meant to perceptually “look good” for human observers.

Instead, it must show the imaged content as scientifically precisely as possible. Pixel metrics such as MSE, MAE, PSNR (or even SSIM) do NOT have a preference for smooth images, as the authors claim in multiple occasions throughout the manuscript. Best quality values will ALWAYS be achieved when the predicted image is exactly like the GT image and smoothing results will not improve the measured similarity.

- What I believe the authors observe (and potentially misinterpret) is the fact that smoother looking predictions can at times lead to higher PSNR (or MSE, etc.) values. This is NOT the case if more accurate predictions would be made (see above), which would lead to better PSNR values. Smoother looking predictions are better only when less smooth predictions show more contrasted structures in the WRONG places, hence, in occasions when the prediction is actually NOT perfectly in-line with the given GT image. Since we are dealing with scientific image data where scientist users might actually measure and quantify the precise location, width, size, angle, etc. of the smallest biological structures, I believe that a realistic looking (non smoothed) structure that is not in-line with GT is a bigger problem than some smoothness introduced by uncertainty in the given input image (more details below).

- Smooth predictions in well trained networks (as the baselines used in the manuscript) are typically the result of using L1 or L2 losses. Those losses deal with data uncertainty (aleatoric uncertainties) by predicting outputs that approximate the MMSE of the distribution of possible true predictions, i.e. the average/median of all possible data interpretations that are in-line with the learned posterior. Such an average becomes blurry by definition. We can also observe this phenomenon in the submitted manuscript, e.g. in Extended Data Figure 4. (Side note: using an L1 loss will converge to the median prediction, not the average prediction as L2 losses do [see for example the Noise2Noise paper for a more in-depth explanation].)
- Other interesting metrics are not currently used: The EM and cryo-EM field is very successfully using Fourier Ring/Shell Correlation (FRC/FSC) to assess the resolution in their images/tomograms/averages etc. At the very least for the super-res part of this manuscript this would make a lot of sense to be adopted by the authors. While it is nice to show Fourier Spectra (as for example in Figure 1b), if and to what degree the higher frequency values predicted by EMDiffuse actually correspond to the correct values would be evaluated using FRC/FSC.
- Diffusion models do not minimize an L1 or L2 loss and are therefore not producing the median or mean prediction of all possible solutions, but are instead sampling one possible solution that is not smooth, even in places where input data comes with uncertainty. Diffusion models are applied in iterations and can therefore potentially lead to higher quality predictions. Still, it is of utmost importance to educate future users that a single prediction is not the only valid interpretation of the input data. A second prediction of the same diffusion model might be different in ways critical to the objective of the user (mainly if small structural differences ought to be evaluated). I think this point must be made crystal clear in the manuscript, since an obligation of developing and publishing powerful new methods is to educate future users. This will prevent (or minimize) problems that will otherwise arise when biologists will use results obtained by EMDiffuse for downstream data quantifications in the context of their biological research projects.
- ****In summary:**** instead of claiming that PSNR is not measuring a desired property of predictions (which it clearly does), I would prefer if the authors educate readers on what PSNR measures, why it can lead to better values for smoother predictions (see above) and then go on to explain why diffusion models are doing something different than other

methods that are trained to minimize L1 or L2 losses.

2. The uncertainty readout introduced by the authors could be improved and better explained.

- In its current form, the manuscript is not explaining well enough the uncertainty readout the authors are introducing. Despite new text being introduced (Lines 797-815), some variables remain undefined, normalization is not discussed but likely required, the choice of removing 1% top uncertain pixels is not justified, and the choice of 0.12 as the image-level threshold for good vs. “bad” predictions still lacks justification and will lead to problems as the one I explain next:

- Example: Lets assume I predict results for four 500x500 pixels sized images that would lead to average uncertainty values of 0.2, 0.07, 0.07, and 0.06, respectively. According to the authors, we would reject the first image but not the other three. What I did so far not share is that those 4 images actually are 4 pieces of a 1000x1000 sized image that I simply chopped in four pieces. After re-assembling the original image, we will predict an average uncertainty value of 0.1 $((.2+.07+.07+.06)/4)$ and will not flag the prediction as problematic and therefore use the entire image (including the previously rejected quarter). I hope this makes it clear that the uncertainty measurement introduced by the authors is not ready for prime time and the way it is introduced and “sold” in the manuscript misleading and unscientific.

- The literature distinguishes data and model uncertainty (aka aleatoric and epistemic uncertainty). I think it would be important to discuss the uncertainty readout introduced in this manuscript in terms of those types of uncertainty. What is it that the authors are evaluating? I have made some more detailed comments below.

Heartfelt Suggestions

- To enable others to benefit from the work of the authors, sharing source code is critical. While the authors are now pointing in their response to a publicly available GitHub repository, the manuscript itself does not, but should.

- Additionally, a two not very time-consuming steps for the authors can help to save a lot of work for future users.

- Sources could contain example notebooks that download and prepare training and

validation data in the way EMDiffuse requires them and then starts a training run that re-creates one (or even all) of the models used to create the figures in the manuscript.

- The models used to create the figures could be offered to be downloaded (enabling reproducibility). This can happen in many ways, one of which would be to upload the trained models to bioimage.io. While this might come with a bit of overhead for the authors (when done the first time), it has the big advantage that future readers of the manuscript can download and use the uploaded models in various user-facing software tools and even test the model directly online in the browser UI provided at bioimage.io. If the authors decide to upload to the Bioimage Model Zoo or not is their choice, of course, but I would strongly suggest to make the trained models used to create the figures for the paper publicly available in one way or another.

Main Figures

- Figure 1a: What is the Gaussian shown in the Inference panel? This remains entirely unexplained in caption (and I believe in the main text as well). Additionally, the Uncertainty map seems to not fit the shown prediction (there are strong red structures visible that do NOT correspond to visible diversity in the images shown anywhere else in this figure). Also, I find the “Self-assess” arrow unclear. Why is it pointing up? Does “Self-asses” mean the user must self assess?

- Figure 1b: the line-plots seem cherry-picked and would look more convincing when longer than 10 pixels. Also, why is the y-axis normalized and not showing native intensity values? What is the nature of the normalization (cannot find this anywhere).

- Figure 1d: it is interesting that the “Uncertainty” column of images does not show high uncertainty values in places where the predicted image is actually wrong (wrt. the higher quality images shown in the same figure). For example, the top right panel does not highlight the missing vesicles that become apparent in the second row and below. I would suggest the authors to explain why this is not the case and spell out what the uncertainty measures instead.

- Figure 2: this would be an ideal place to add FRC/FSC measurements (see major criticisms).

Other Comments

- Line 65: I'm not sure the sentence/reasoning here is correct. In [20], for example, GANs are used. If they are labeled a "regression-based deep learning model" I do not see why an iteratively applied U-Net in EMDiffuse is not as well. The statement of "excessively smooth predictions" is one of the instances where the author's understanding regarding data uncertainty and networks trained with L1 or L2 losses shows (see major criticisms).
- Line 113: PSNR prefers over-smoothed results. No, it does not (see major criticisms).
- Noise2Noise and Noise2Void are typically written with capital letters (as I did here).
- Line 130: Smoothness is never "prioritized". It is a consequence of uncertainty in the input data with respect to the trained model (see major criticisms).
- Line 146: what does "should be used with caution" mean?
- Line 158 and Ext. Data Fig. 7: while the main text mentions the size of the one image used during fine-tuning, the Ext. Data Fig. does not. The fact that one image is used is therefore a bit imprecise/ misleading — was a single image of size 256x256 being used, results would likely be much worse, while an image of 4000x4000 would likely lead to better results. Why not talking about kilo- or megapixels being used for fine-tuning instead?
- Line 679: it appears that the Noise2Void results were obtained using a codebase that is not the official Noise2Void code. This is not ideal since a diligent reader would have to check if the implementation actually performs equally well as the original Noise2Void implementation.
- Line 711: This formula is not 100% in-line with what I believe is the default definition for PSNR and is additionally using confusing nomenclature. What are the parameters of PSNR or MSE (which I assume are functions of something). What are m, n, l , and R ? Additionally, most image restoration publications actually use an intensity shifted PSNR, i.e. allowing the predicted image intensities (R) to be shifted by a constant in order to minimize the computed PSNR value. The reasoning is to evaluate the predicted structure, not some systematic intensity shift throughout the image.
- Line 719: PSNR does not "favor" blurry images (see major criticisms).
- Line 797: the current manuscript never distinguishes Data and Model Uncertainty (also known as aleatoric and epistemic uncertainty). What is the uncertainty in this manuscript measuring?
- Line 803: are the intensities X_0^k normalized? If so, how? If not, the constant threshold of 0.12 suggested to be used would have to be adjusted when the same image with 100-fold

scaled pixel intensities was to to be used.

- Line 804: x_0 is not introduced - is this the GT pixel intensity?
- Line 811: “density” appears to be the wrong word. Should it be “intensity”?
- Line 813: Why 1%? This appears to be such a random choice and might be entirely wrong for other datasets than the ones used by the authors for the sake of the given manuscript.
- Line 822: I cannot see the “transition valley” the authors point to in Fig. 1c.
- Line 1254 (Ext. Data Fig. 4): I’m having a hard time seeing what “human perception” should want me to see here. Also, when talking about scientific image data quantification, maybe human perception is not what we want to optimize for (see major criticisms). The computed mean image of multiple predictions showing higher and higher PSNR values is testament of the major criticism I voiced at the very beginning. The mean of many predictions converges to the MMSE of the distribution of possible predictions and is therefore the minimizer of an L2 loss and of PSNR. The fact that a single prediction has an higher PSNR also means that the high frequency components of the predicted images (outer parts in a Fourier spectrum of that prediction) are not in-line with the GT, and the MMSE is smooth because of the uncertainty in the given input data. Hence, this result is not undesired but highly informative.
- Line 1260: PSNR does not prefer over-smoothed results. Try to take any prediction and smooth it with some small gaussian kernel. PSNR will be MUCH worse instantly!
- Ext. Data Figure 6 (and likely some other places): it would be very informative to explicitly show the 1% discarded uncertainty pixels. Currently the fact that the 1% highest uncertainty pixels are discarded is buried in the methods and not every reader will pick this up. It is an important fact though, because users will have to adjust this value if their data would contain more or less strong image gradients, wouldn’t they?
- Ext. Data Figure 7: as mentioned before, speaking about “one training pair” is not as informative as pointing at the total number of pixels available during fine-tuning. Also, if the new domain is more or less different from the one the network is trained on, the amount of pixels will needed will naturally vary. While it is too much to ask to quantify this, the authors could at least acknowledge this fact by spelling it out.

Reviewer #4 (Remarks on code availability):

- Code should be linked from the main text.

- Tutorials (e.g. example notebooks that reproduce results from the paper) would be highly recommended and useful.
- Uploading trained models in one way or another would be very useful and highly recommended. Bioimage.io would be ideal since it would allow future users to test the model in their browser and re-use them in other scientific image analysis tools.

Point-by-point response to the reviewers' comments

REVIEWER COMMENT

Reviewer #1 (Remarks to the Author):

Thank you for releasing the data and software. (I believe that the 3D-SRU-Net code may have been committed in a non-functional state, it seems that a and b are confused in the main call and that there are two competing scaling factors 6 and 8?) Also thanks for adding additional references and comparison. To me, those changes make the paper stronger and support that this work is interesting and contributes to the field. I really like this work.

RESPONSE: We thank the reviewer for acknowledging the strength of the manuscript and revision. We have uploaded the 3D-SR-Unet code in a functional state for reference (<https://github.com/Luchixiang/EMDiffuse/tree/master/3D-SR-Unet>).

I am still concerned about some presentations in the main paper (the methods part is straight forward and easy to read). Particularly the qualitative statements about the high resolution that the diffusion models generate requires some attention (see below), e.g. 99-100 "to produce the denoised image." is somehow misleading, because the strength of the method is that it learns to produce many plausible denoised images. This comes back in several other places.

RESPONSE: Thank you for your suggestions regarding the presentation of the generative capability of EMDiffuse. We appreciate your emphasis on the importance of clarity when discussing the model's ability to generate multiple plausible images since denoise and super-resolution are ill-posed one-to-many problems. We have revised the manuscript according to the comments below.

This work uses 2D architectures to do 2D denoising and slice in-painting to improve 3D resolution. The manuscript talks a lot about 3D, yet never about that the architecture being 2D. The fact that the architectures are 2D transpires on the side of patch sizes discussed with two numbers. I find it interesting that a 2D method achieves results superior to a 3D method which should be discussed?

RESPONSE: Thanks for your suggestion on discussing the 3D and 2D models. There are three main reasons why a 2D diffusion model can outperform the 3D model on vEM datasets. Firstly, we incorporated axial information into the 2D diffusion model through channel embedding. This technique enables the model to accurately discern and generate specific layers within the 3D structure. By doing so, we effectively instruct the 2D model in handling 3D data, thereby overcoming the inherent limitations of its 2D nature. Secondly, the strong generative capability of the diffusion model empowered it to reconstruct fine structures. Thirdly, compared to 3D-SRU-Net, the 2D network is much more computation and memory efficient, which makes it feasible to increase the size of the network (e.g., the number of layers) and enhance the capability of the model to process complex biological structures. We have included the discussion in Lines 397-402.

74--76 CARE implements a reliability assessment, please discuss or rephrase.

RESPONSE: Thanks for highlighting CARE's reliability assessment. CARE assesses aleatoric uncertainty by training networks to predict the mean and variance and assesses epistemic uncertainty by evaluating the agreement across five distinct CARE networks. Thus, training five separate neural networks is required. While effective, this approach introduces a significant

increase in training overhead. In contrast, EMDiffuse simplifies this process by relying on a single model. The uncertainty in EMDiffuse is determined by calculating the variance among outputs produced during different test runs. We have revised the manuscript and added the discussion in manuscript Lines 79-82.

87 propose to change to: "[...]the inherent ability of diffusion models to generate an arbitrary number of diverse samples..."

RESPONSE: Thanks for your pointing it out. We have modified the manuscript accordingly. Please refer to Lines 94-95.

111 and below do not address appropriately that each individual generated image is just a sample from the distribution of plausible images given the noisy data. The average of 2 may optimize the metrics used for comparison but has no other meaning (see below).

RESPONSE: Thanks for your comment. We have clarified that each individual result is a sample from the distribution in Lines 94-95, 113-123, 348-352, 430-431, 1305-1307 in the revised manuscript. The average of 2 outputs is simply because we found it can enhance the reliability of the results, as reflected by the high similarity between the prediction and ground truth assessed by the FSIM metric. We have also configured the code to generate both individual outputs and averaged predictions.

Extended Data Fig. 3 I think it would be less confusing if the arrow "intricate Structure" would point to the same location in all images, including GT. The input crop is not aligned with the output crops and GT.

RESPONSE: Thank you for pointing out this confusion and misalignment between input and output. We have revised the Extended Data Fig. 3 accordingly (*see figure below*).

Extended Data Fig. 3 EMDiffuse samples one plausible solution at each test time, and the prediction can be optimized using the means of different outputs.

a, b. EMDiffuse samples one plausible solution from distribution. In cases where the intricate structure is heavily dominated by noise (**a**), variance exists in sampled images (*arrows*). For less noisy cases (**b**), outputs are highly consistent. Multiple outputs can be sampled, analyzed individually, or averaged to improve the final prediction. Scale bar, 0.1 μm .

c. Assessments of EMDiffuse performance based on FSIM. Shown are the FSIM of EMDiffuse-processed images of different noise levels and the mean of different numbers of outputs. The number of outputs for generating the mean result that yields the highest FSIM is selected for EMDiffuse prediction.

d. Quantification of resolution ratio (resolution of GT/resolution of EMDiffuse prediction) from the mean of different numbers of EMDiffuse outputs and different input noise levels. Scale bar, 0.1 μm .

Language: "EMDiffuse can adopt the mean result of different testing times prediction as the ultimate prediction to increase fidelity" what does this mean? What even is "fidelity"? I suggest to stick to the facts, e.g.: "EMDiffuse generates plausible super-resolution images with convincing high-resolution details. However, super-resolution from noisy images is an ill-posed problem, and each super-resolution output is just one possible solution to this problem. To account for this, we can generate an arbitrary number of outputs for the same input from different initial noise. One can use these outputs to assess the variance at each pixel, or the variance of interpretations of the data (e.g. number of pre-synaptic vesicles)."

Following up on this: Averaging just two predictions optimizes your error metrics but feels wrong. It means that you do not create a statistically meaningful average nor account correctly for the distribution of your solutions. Creating the mean of all plausible solutions does not do justice to the fact that all solutions are likely plausible themselves. E.g. in the example in EDFig. 3, the network has learned that the noisy input near the synaptic cleft indicates the presence of a number of vesicles. Each output has a number of vesicles and this number seems highly consistent. Averaging any number of outputs will smear vesicles away, taking any output as the one true solution is similarly incorrect. This is interesting though, because it enables you to e.g. count vesicles in all outputs, and present the average count and variance as a result, since all outputs agree that there are vesicles. Averaging any number of outputs compromises this and is therefore much less interesting than aggregating over the structures embedded in the learned complex distribution that the network allows you to sample.

RESPONSE: Thanks for your comment on whether to generate each plausible solution or average output. As we mentioned above, we have clarified that each individual output is a sample from the distribution in Lines 94-95, 113-123, 348-352, 430-431, 1305-1307 in the revised manuscript.

For highly noisy inputs, we agree that average can lead to inaccuracies (e.g., smearing pre-synaptic vesicles). We have now configured EMDiffuse to generate multiple plausible images for the investigation of areas of interest (Extended Data Fig. 3a, *see figure above*). Conversely, for predictions from less-noisy inputs, we observed outputs from EMDiffuse with highly consistent ultrastructural details (Extended Data Fig. 3b, *see figure above*). For such cases, averaging two outputs is less likely to induce inaccuracy and can further improve the similarity between predictions and ground truth, reflected by the FSIM metric. We recommend the application of

EMDiffuse in later scenarios where the uncertainty value is below the threshold. To provide users with flexibility, our codes now include the option to choose between accepting averaged predictions or exploring each plausible output individually.

The figure title "Optimization of prediction generation methods using multiple outputs enhances reliability while reserving resolution" is meaningless because it does not say any of those interesting things and makes generic qualitative statements about "reliability"(?) and "resolution", none of which is supported by this figure.

RESPONSE: Thanks for your comment on the figure legend Extended Data Fig. 3. We have revised the Extended Data Fig. 3 and its figure legend (*see figure above*).

129 "preserved the high resolution" is IMHO not a correct interpretation. As above, "it generates outputs with high resolution details". The variance of those high-resolution details can be assessed because an arbitrary number of outputs can be generated. This is where the diffusion model based approach excels. The other methods have all been trained to generate the pixel-based average which is inferior.

RESPONSE: Thank you for pointing it out. We have revised all related sentence in the manuscript. Please refer to Lines 130-131, 138, 426-427.

For the per-pixel uncertainty estimation, I believe that images with diverse contrast levels will have lower uncertainty values for lower contrast regions than for higher contrast regions. This makes a single global threshold difficult to interpret. I find this worth mentioning. I also think that it could be addressed by some form of local contrast normalization ([CL]AHE?) which would boost edges?

RESPONSE: Thank you for highlighting the challenge of interpreting global uncertainty thresholds in images and for suggesting the implementation of CLAHE. We indeed incorporated a normalization step into our uncertainty calculation process to mitigate the potential bias introduced by varying contrast levels. Specifically, we applied min-max normalization to each 256x256 pixels patch of a large image independently, enhancing the local contrast. We have documented the details of normalization in uncertainty estimation in the manuscript Lines 860-862.

We further demonstrated that min-max normalization achieves comparable outcomes to CLAHE normalization using our mouse brain cortex dataset (*see figure below*). We have highlighted patches of high uncertainty with min-max normalization in red, and the same regions, also highlighted in red, also showed high uncertainty with CLAHE normalization despite their differences in absolute values.

The violin plots of uncertainty value on mouse brain cortex test datasets (n=960).

Reviewer #1 (Remarks on code availability):

Partial review, there seem to be some bugs here and there, but that can be fixed. I have not run the experiments to reproduce the results because that would have required me to do some amount of plumbing.

RESPONSE: Thank you for pointing it out. We have carefully revised our code and included example notebooks that can be executed directly on Google Colab or a local computer server (<https://github.com/Luchixiang/EMDiffuse/tree/master/example>). These notebooks provide step-by-step instructions and pre-configured settings to facilitate the use of EMDiffuse, lowering the barrier to entry for those interested in applying our method to their data.

Reviewer #2 (Remarks to the Author):

The review addresses my concerns regarding the manuscript. It now includes the references to prior work and extends experiments and method descriptions significantly to explain the comparison to these methods.

The authors also make the code publicly available. The code is currently in a "research code" state. It thus likely enables replication of the results, but is not in a sufficient advanced state to be used by life scientists without significant expertise. See the section on code review below for details. Overall I can recommend this manuscript for publication as it shows the potential of using diffusion models for EM image reconstruction tasks by convincingly showing that these methods outperform the previous state-of-the-art. However, I will add that I think it that it is unlikely that the tool will gain broad usage in the volume EM community in its current form due to the lack of a user-friendly implementation.

RESPONSE: We appreciate the reviewer for recognizing the strength of the manuscript and the revisions. In response to your concerns about user-friendliness, we have developed and included example notebooks that can be executed directly on Google Colab or a local computer server (<https://github.com/Luchixiang/EMDiffuse/tree/master/example>). These notebooks provide step-by-step instructions and pre-configured settings to facilitate the use of EMDiffuse, lowering the barrier to entry for those interested in applying our method to their data.

Reviewer #2 (Remarks on code availability):

I have reviewed the software at <https://github.com/Luchixiang/EMDiffuse> and tried to install it on my laptop in order to test it (see below). Overall, the software is comprised of python scripts without any user interface to help applying it to new data. It has a README that explains how to use the software; however, it is not written well (in several cases sentences are incorrect). It is thus in a state where users with significant computational experience and specialized hardware (access to a GPU) can run the tool. However, this means the tool will be difficult to use for many researchers in the volume EM community, who are biologists by training and rely on user-friendly software with a GUI.

In order to build such user friendly software I can recommend looking into bioimage.io (<https://bioimage.io/#/>) and/or [napari](https://napari.org/stable/) (<https://napari.org/stable/>). While I don't think that this is a prerequisite for publication, it would increase the potential impact of the tool significantly.

As for my own attempts with the software: I tried to run inference for EM denoising. Here is a step by step explanation with comments that may also help to improve it:

- I installed the software via conda + pip as described. This worked, however the 'tqdm' dependency was missing. This was easy to fix (for me), but could already be a hurdle for computationally inexperienced users.
- I tried to run inference according to <https://github.com/Luchixiang/EMDiffuse#inference>.
- Downloading the model and example dataset worked as described.
- The cropping step worked.
- Running the actual inference step fails on my laptop because I don't have a GPU. It's unclear why you hard-code the model to run on the GPU here. Normally inference works fine on the CPU (training of course is a different story), although taking longer, and implementing GPU and CPU support is trivial in PyTorch. Note that a lot of potential users may not have direct access to a GPU. Even if they do, the machine with the GPU may not have support for visualization so training with GPU and inference with CPU may be a common use-case.

At this point I stopped because setting up the software on our cluster with GPU support would be too much effort for me right now.

RESPONSE: Thank you for the feedback on the code of EMDiffuse. We have carefully revised the README file. Regarding the integration of EMDiffuse into bioimage-zoo, we appreciate your suggestion and have investigated this option. The current framework of bioimage-zoo presents specific challenges for models like EMDiffuse that generate outputs with inherent variability, a feature critical for our model's uncertainty assessment capability. The requirement for an exact match between model output and a provided output example (<https://github.com/bioimage-io/spec-bioimage-io/issues/525>) does not accommodate EMDiffuse. We have engaged with the bioimage-zoo community to discuss potential support for generative models and will continue to work towards making EMDiffuse compatible with their platform.

For installing and running EMDiffuse from the source code, we have fixed the installation bug. Since the diffusion model runs slowly on the CPU, we hard-coded it on the GPU. However, we

have enabled the inference on the CPU in the updated code. As mentioned above, we have also added the notebook to facilitate using EMDiffuse. We are keen to develop the GUI, such as the Napari plugin, to promote EMDiffuse in EM and vEM applications in the future.

Reviewer #3 (Remarks to the Author):

Thank you for revising manuscript. The authors have adequately addressed the reviewer's comments, and overall, the manuscript is well-crafted. However, please consider followings.

Thank you for providing a thorough elucidation of "transfer learning with limited training samples" and "structure size similar to z-step" in the context of artifacts.

RESPONSE: We express our gratitude to the reviewer for acknowledging the revisions made.

Recognizing the broader implications of your technology beyond the machine learning expert community, it becomes crucial to tailor your explanations for a more diverse readership, including non-experts. In relation to "transfer learning with limited training samples," I encourage you to expound on the concept of domain gap, elucidating its significance and emphasizing the necessity of incorporating diverse structures to mitigate the impact of the domain gap in the learning or transfer learning process. Such clarifications would enhance the accessibility of your work to a wider readership.

RESPONSE: We appreciate the suggestion to expand on the concept of domain gap. We have revised our manuscript to include an explanation of the domain gap and elucidate the necessity of incorporating diverse structures. Please refer to Lines 158-163.

With regards to "structure size similar to z-step," I appreciate the inclusion of the extended Figure 20, which effectively elucidates the application of vEMDiffuse-i to images featuring ERs and mitochondria. However, the absence of information related to synapses and synaptic vesicles creates a noticeable void in the narrative. Readers are likely eager to comprehend the effects on these structures. Hence, I propose expanding the discussion to include the outcomes of applying vEMDiffuse-i to synapses and synaptic vesicles. Additionally, shedding light on the results of employing vEMDiffuse-a on these structures would enhance the overall comprehensiveness of your findings.

RESPONSE: Thank you for the suggestion. We have included the discussion on the structure size similar to z-step and the potential application of vEMDiffuse-i and vEMDiffuse-a in synapses and synaptic vesicles in Lines 404-412.

The axial resolution of the anisotropic volume is recommended to be as fine as the diameter of interested structures. A greater axial resolution significantly risks fine structures being indiscernible or inaccurately reconstructed, appearing smeared across successive layers, thus undermining the reconstruction. We have expanded this discussion in Lines 404-412 in the revised manuscript. We expect that vEMDiffuse-i and vEMDiffuse-a can reconstruct other fine structures accurately, including synaptic vesicles (shown in Extended Data Fig. 18) and synapses (partially shown in neurite reconstruction experiments in Extended Data Fig. 27).

In the interest of transparency, it is crucial to address any limitations associated with your technique. If there are constraints or shortcomings, kindly elucidate them to provide a balanced perspective on the applicability and scope of your approach."

RESPONSE: Thank you for suggesting elucidating the constraints and shortcomings of EMDiffuse.

Firstly, as mentioned above, the axial resolution of anisotropic volume will limit the capability of vEMDiffuse-i and vEMDiffuse-a. We have expanded this discussion in Lines 404-412. Secondly, EMDiffuse is still computationally heavier than that of traditional neural networks, which could be further refined by the recently developed stable diffusion model. This limitation is mentioned in Lines 414-419.

Thank you for your attention to these matters, and I look forward to seeing these refinements in your manuscript.

Reviewer #4 (Remarks to the Author):

The authors did a good job taking our reviewer feedback into account while preparing their resubmission. The manuscript did improve in many respects and I have only a few but rather fundamental remaining issues to point out.

I believe that after addressing those, the manuscript merits publication. I hope, though, the authors will take my feedback as serious as I believe it is.

RESPONSE: Thank you for the suggestions. We have revised the manuscript according to the detailed comments below.

Major Points of Criticism

1. The current manuscript and supplementary information draws a misleading picture about what PSNR measures and why it should or should not be used.

After reading the authors arguments for not using PSNR more prominently, I believe the authors are either not fully aware of what I will describe below or are not educating readers in an appropriate way in the current manuscript.

- Scientific image data is not meant to perceptually "look good" for human observers. Instead, it must show the imaged content as scientifically precisely as possible. Pixel metrics such as MSE, MAE, PSNR (or even SSIM) do NOT have a preference for smooth images, as the authors claim in multiple occasions throughout the manuscript. Best quality values will ALWAYS be achieved when the predicted image is exactly like the GT image and smoothing results will not improve the measured similarity.

- What I believe the authors observe (and potentially misinterpret) is the fact that smoother looking predictions can at times lead to higher PSNR (or MSE, etc.) values. This is NOT the case if more accurate predictions would be made (see above), which would lead to better PSNR values. Smoother looking predictions are better only when less smooth predictions show more contrasted structures in the WRONG places, hence, in occasions when the prediction is actually NOT perfectly in-line with the given GT image. Since we are dealing with scientific image data where scientist users might actually measure and quantify the precise location, width, size, angle, etc. of

the smallest biological structures, I believe that a realistic looking (non smoothed) structure that is not in-line with GT is a bigger problem than some smoothness introduced by uncertainty in the given input image (more details below).

- Smooth predictions in well trained networks (as the baselines used in the manuscript) are typically the result of using L1 or L2 losses. Those losses deal with data uncertainty (aleatoric uncertainties) by predicting outputs that approximate the MMSE of the distribution of possible true predictions, i.e. the average/median of all possible data interpretations that are in-line with the learned posterior. Such an average becomes blurry by definition. We can also observe this phenomenon in the submitted manuscript, e.g. in Extended Data Figure 4. (Side note: using an L1 loss will converge to the median prediction, not the average prediction as L2 losses do [see for example the Noise2Noise paper for a more in-depth explanation].)

RESPONSE: Thank you for your detailed critique regarding our discussion and application of PSNR within our manuscript and reason of smoothness of baseline models. We acknowledge the error in our previous assertions that PSNR inherently favors smoother images. We have removed all the related descriptions in the revised manuscript. As you rightly pointed out, the best PSNR value is always achieved when the prediction perfectly matches the ground truth (both the structure position and intensity). However, in electron microscopy, it is not feasible to capture noise-free images as ground truth. Thus, the pixel-wise mean square error and PSNR scores (error maps in Extended Data Fig. 4) cannot necessarily account for high-quality results due to the impact of random noise. Instead, a smooth result, due to the approximation of MMSE of possible distribution, will tend to alleviate the influence of noise in ground truth image by taking an averaged or median value and thus lead to a higher result in PSNR calculation. Our claim should be put into the context of electron microscopy images where random noise exists in ground truth images. We have carefully revised our manuscript to correct the misleading statements regarding PSNR and clarify what PSNR measures. We have also declared in the manuscript the source of smoothness in baseline model outputs. Please refer to Lines 65-69, 113-117, 121-123, 140-141, 345-352, 753-768, 1320-1332.

- Other interesting metrics are not currently used: The EM and cryo-EM field is very successfully using Fourier Ring/Shell Correlation (FRC/FSC) to assess the resolution in their images/tomograms/averages etc. At the very least for the super-res part of this manuscript this would make a lot of sense to be adopted by the authors. While it is nice to show Fourier Spectra (as for example in Figure 1b), if and to what degree the higher frequency values predicted by EMDiffuse actually correspond to the correct values would be evaluated using FRC/FSC.

RESPONSE: Thank you for highlighting the need to incorporate Fourier Ring Correlation (FRC) as an additional metric. We have now included it in the revised manuscript. The Fourier ring correlation between predictions and ground truth images is plotted using the FIJI plugin (<https://imagej.net/plugins/fourier-ring-correlation>). The Fourier ring correlation plot reflects the correlation between two images in different spatial frequency components.

$$FRC(r) = \frac{\sum_{r_i \in r} F_1(r_i) * F_2(r_i)^*}{\sqrt{\sum_{r_i \in r} |F_1(r_i)|^2 * \sum_{r_i \in r} |F_2(r_i)|^2}}$$

where F_1 and F_2 are the Fourier transform of generated image and ground truth image. $F_1(r_i)$ refers to pixels on the perimeter of circles of constant spatial frequency with magnitude r .

Here, we use FRC to measure the correlation between predictions from our method and the ground truth reference. The Fourier ring correlation plot (Fig. 2d) of EMDiffuse’s prediction shows a stronger correlation with the ground truth in the high spatial frequency domain, indicating that EMDiffuse captures the intricate details present in the high-frequency components. We have modified Figure 2 and the manuscript accordingly. Please refer to Lines 194-195. Methods details about the Fourier ring correlation have been included in Lines 818-826.

Fig. 2: EMDiffuse achieves superior performance in super-resolution. **a.** Schematic of EMDiffuse-r for super-resolution of EM images. **b.** Representative region from mouse brain sections (input with a pixel size of 6.6 nm, GT with a pixel size of 3.3 nm) for comparing the super-resolution capability of EMDiffuse-r with CARE, PSSR, and RCAN. Top-right of each panel is the Fourier power spectrum. The resolution of processed images is shown on the Fourier transform image. The uncertainty value is shown in the bottom right corner. **c.** The 3D scatter plot and distribution plots show the quantitative performance assessment of EMDiffuse, CARE, PSSR, and RCAN for the super-resolution task. Each point represents a unique mouse brain cortex, $n = 960$. **d.** The Fourier ring correlation plot between predictions in **b** and ground truth image. **e.** EMDiffuse for super-resolution is robust and transferable to other tissues, including the liver, heart, and bone marrow. Resolution values and uncertainty values are indicated on each panel. Scale bar, (b), 0.1 μm ; (e), 0.3 μm .

- Diffusion models do not minimize an L1 or L2 loss and are therefore not producing the median or mean prediction of all possible solutions, but are instead sampling one possible solution that is not smooth, even in places where input data comes with uncertainty. Diffusion models are applied in iterations and can therefore potentially lead to higher quality predictions. Still, it is of utmost importance to educate future users that a single prediction is not the only valid interpretation of

the input data. A second prediction of the same diffusion model might be different in ways critical to the objective of the user (mainly if small structural differences ought to be evaluated). I think this point must be made crystal clear in the manuscript, since an obligation of developing and publishing powerful new methods is to educate future users. This will prevent (or minimize) problems that will otherwise arise when biologists will use results obtained by EMDiffuse for downstream data quantifications in the context of their biological research projects.

RESPONSE: Thank you for emphasizing the crucial distinction in how diffusion models operate compared to traditional models that minimize L1 or L2 losses. We have revised our manuscript accordingly to recognize the importance of conveying this information to future users. We have also educated the user about the potential variability between different outputs. Please refer to Lines 113-117, 345-352, 430-431, 832, 1305-1307. This is also the reason that we incorporate the uncertainty assessment for users to refer to.

- ****In summary:**** instead of claiming that PSNR is not measuring a desired property of predictions (which it clearly does), I would prefer if the authors educate readers on what PSNR measures, why it can lead to better values for smoother predictions (see above) and then go on to explain why diffusion models are doing something different than other methods that are trained to minimize L1 or L2 losses.

RESPONSE: Thank you for your feedback on our presentation of PSNR in our manuscript. As mentioned above, we have undertaken a comprehensive revision to clarify what PSNR precisely measures and expanded our discussion to contrast the diffusion models with those of baseline models (i.e., the difference between approximating the MMSE of possible solution distribution and sampling one plausible solution). Please refer to Lines 65-69, 113-117, 121-123, 140-141, 345-352, 430-431, 753-768, 832, 1305-1307, 1320-1332.

2. The uncertainty readout introduced by the authors could be improved and better explained.

- In its current form, the manuscript is not explaining well enough the uncertainty readout the authors are introducing. Despite new text being introduced (Lines 797-815), some variables remain undefined, normalization is not discussed but likely required, the choice of removing 1% top uncertain pixels is not justified, and the choice of 0.12 as the image-level threshold for good vs. “bad” predictions still lacks justification and will lead to problems as the one I explain next:

- Example: Lets assume I predict results for four 500x500 pixels sized images that would lead to average uncertainty values of 0.2, 0.07, 0.07, and 0.06, respectively. According to the authors, we would reject the first image but not the other three. What I did so far not share is that those 4 images actually are 4 pieces of a 1000x1000 sized image that I simply chopped in four pieces. After re-assembling the original image, we will predict an average uncertainty value of 0.1 $((.2+.07+.07+.06)/4)$ and will not flag the prediction as problematic and therefore use the entire image (including the previously rejected quarter). I hope this makes it clear that the uncertainty measurement introduced by the authors is not ready for prime time and the way it is introduced and “sold” in the manuscript misleading and unscientific.

RESPONSE: Thank you for your feedback regarding the uncertainty. Our methodology evaluates uncertainty on a patch-by-patch basis for large images. Specifically, we divide large images into smaller patches of 256 x 256 pixels. Min-max normalization was applied to each patch to enhance local contrast, a critical step for ensuring that variance calculations reflect local uncertainty accurately. We have included it in Lines 860-862 in the revised manuscript.

The decision to remove the top 1% of uncertain pixels was based on an empirical analysis of approximately 200 image predictions and their corresponding ground truths from our mouse brain cortex denoising dataset. This analysis aimed to identify an appropriate threshold for distinguishing outliers and high-uncertainty values in the structure boundaries, which does not necessarily mean incorrect predictions (Extended Data Fig. 6, *see figure below*). Our findings indicated that excluding the top 1% of uncertainty values helps to align our uncertainty readout more closely with human judgments of image reliability. Nevertheless, because electron microscopy has a wide range of applications, the quantity of disregarded pixels may vary across different datasets and can be modified by users through examination of the uncertainty map and estimation of the unreasonably high number of uncertain pixels. Please refer to Line 357-360, 874-880.

EMDiffuse does not average uncertainty values across patches to determine the overall image quality. Instead, if any individual patch exceeds the predetermined uncertainty threshold, EMDiffuse flags the entire image and the specific patch as potentially problematic upon reassembly. We have clarified our uncertainty estimation method in the revised manuscript. Please refer to Lines 880-882.

Extended Data Fig. 6 Uncertainty map for self-assessment of prediction reliability.

EMDiffuse denoising predictions and uncertainty maps of two example regions with different noise levels. Each row consists of the noise levels, input image, EMDiffuse prediction, uncertainty map and 1% ignored uncertainty pixels consisting of outliers and pixels in the structure boundary. A threshold of 0.12 for the total uncertainty is set for assessing the reliability of the final prediction, as shown in the last column. The caution signifies the potential inaccuracies in the predicted structure, while yes denotes that the prediction is reliable. Scale bar, 0.1 μm .

- The literature distinguishes data and model uncertainty (aka aleatoric and epistemic uncertainty). I think it would be important to discuss the uncertainty readout introduced in this manuscript in terms of those types of uncertainty. What is it that the authors are evaluating? I have made some more detailed comments below.

RESPONSE: Thank you for emphasizing the importance of distinguishing between data uncertainty (aleatoric) and model uncertainty (epistemic) in our manuscript. EMDiffuse primarily evaluates model uncertainty by analyzing the variance in distinct predictions for a given input. We have included it in the manuscript. Please refer to Line 856, 858.

Heartfelt Suggestions

- To enable others to benefit from the work of the authors, sharing source code is critical. While the authors are now pointing in their response to a publicly available GitHub repository, the manuscript itself does not, but should.

RESPONSE: Thanks for your comment. We have linked the code in the manuscript. Please refer to Lines 95-96.

- Additionally, a two not very time-consuming steps for the authors can help to save a lot of work for future users.

- Sources could contain example notebooks that download and prepare training and validation data in the way EMDiffuse requires them and then starts a training run that re-creates one (or even all) of the models used to create the figures in the manuscript.

- The models used to create the figures could be offered to be downloaded (enabling reproducibility). This can happen in many ways, one of which would be to upload the trained models to bioimage.io. While this might come with a bit of overhead for the authors (when done the first time), it has the big advantage that future readers of the manuscript can download and use the uploaded models in various user-facing software tools and even test the model directly online in the browser UI provided at bioimage.io. If the authors decide to upload to the Bioimage Model Zoo or not is their choice, of course, but I would strongly suggest to make the trained models used to create the figures for the paper publicly available in one way or another.

RESPONSE: We appreciate your suggestion and have thoroughly investigated this option. We have developed and included example notebooks that can be executed directly on Google Colab or a local computer server (<https://github.com/Luchixiang/EMDiffuse/tree/master/example>). These notebooks provide step-by-step instructions and pre-configured settings to facilitate the use of EMDiffuse, lowering the barrier to entry for those interested in applying our method to their data.

The current framework of bioimage-zoo presents specific challenges for generative models like EMDiffuse that generate outputs with inherent variability, a feature critical for our model's uncertainty assessment capability. The requirement for an exact match between model output and a provided output example (<https://github.com/bioimage-io/spec-bioimage-io/issues/525>) does not accommodate EMDiffuse. We have engaged with the bioimage-zoo community to discuss potential support for generative models and will continue to work towards making EMDiffuse compatible with their platform. We have made the model weights used to create the figure publicly available

(https://connecthkuhk-my.sharepoint.com/:f/g/personal/u3590540_connect_hku_hk/EtSvqrIyrNREim5dJfabx2ABMLNhwk2Z9EsJDD4w6mls8g?e=OdP4Vq).

Main Figures

- Figure 1a: What is the Gaussian shown in the Inference panel? This remains entirely unexplained in caption (and I believe in the main text as well). Additionally, the Uncertainty map seems to not fit the shown prediction (there are strong red structures visible that do NOT correspond to visible diversity in the images shown anywhere else in this figure). Also, I find the “Self-assess” arrow unclear. Why is it pointing up? Does “Self-asses” mean the user must self assess?

RESPONSE: Thank you for your suggestion on Figure 1a. We've replaced the term "Gaussian" with "solution distribution" to more accurately reflect the inference stage. This change is intended to clarify that EMDiffuse-n samples a single potential output from the learned data distribution at each test time given an input image. We have updated the uncertainty map to ensure that it accurately corresponds with the prediction displayed in Figure 1a.

The term "self-assess" refers to EMDiffuse's capability to evaluate the reliability of its predictions internally by assessing the variance between different outputs, without the need for external validation or the incorporation of additional parameters during training. The arrow pointing upwards in the diagram symbolizes this process, where EMDiffuse utilizes its internal variance to measure prediction reliability. We have revised the manuscript and figure legend to avoid misunderstanding. Please refer to Lines 426-433.

- Figure 1b: the line-plots seem cherry-picked and would look more convincing when longer than 10 pixels. Also, why is the y-axis normalized and not showing native intensity values? What is the nature of the normalization (cannot find this anywhere).

RESPONSE: Thank you for your constructive feedback on the line plots in Figure 1b. We have expanded the length of each line profile from 10 pixels to 30 pixels. This extension allows for a more comprehensive comparison between EMDiffuse and ground truth. We employed min-max normalization to scale the intensity values within each line plot for an easier visual assessment of the relative differences and similarities in pattern and structure. The choice of min-max normalization has now been explicitly described in the revised figure legend. Please refer to Lines 440-441.

- Figure 1d: it is interesting that the “Uncertainty” column of images does not show high uncertainty values in places where the predicted image is actually wrong (wrt. the higher quality

images shown in the same figure). For example, the top right panel does not highlight the missing vesicles that become apparent in the second row and below. I would suggest the authors to explain why this is not the case and spell out what the uncertainty measures instead.

RESPONSE: Thank you for pointing out that the missing vesicles are not being explicitly highlighted in the uncertainty maps. In the first row of Figure 1d, due to the high noise levels present in the input images, EMDiffuse faces significant challenges in accurately predicting specific structures like vesicles. Consequently, the model may generate various plausible structures within these highly uncertain regions, rather than reproducing the exact vesicles. The uncertainty map, therefore, reflects high variance (indicated by red or yellow colors) in these regions, signaling the model's lack of confidence in its predictions. However, this high variance is not directly tied to the specific absence or presence of vesicles but rather to the overall unreliability of predictions in that area due to the noise. As such, the map does not delineate the shape of missing vesicles but instead shows high uncertainty areas.

- Figure 2: this would be an ideal place to add FRC/FSC measurements (see major criticisms).

RESPONSE: Thank you for your comments. As mentioned above, we have incorporated an FRC plot into Figure 2.

Other Comments

- Line 65: I'm not sure the sentence/reasoning here is correct. In [20], for example, GANs are used. If they are labeled a "regression-based deep learning model" I do not see why an iteratively applied U-Net in EMDiffuse is not as well. The statement of "excessively smooth predictions" is one of the instances where the author's understanding regarding data uncertainty and networks trained with L1 or L2 losses shows (see major criticisms).

RESPONSE: Thank you for pointing it out. We have revised the reference to remove the GAN-based methods and explained the reason for smoothed predictions. Please refer to Lines 65-69.

- Line 113: PSNR prefers over-smoothed results. No, it does not (see major criticisms).

RESPONSE: Thank you for your comments. As mentioned above, we have revised the sentences. Please refer to Lines 122-123.

- Noise2Noise and Noise2Void are typically written with capital letters (as I did here).

RESPONSE: Thank you for your pointing it out. We have corrected it in the manuscript and figures.

- Line 130: Smoothness is never "prioritized". It is a consequence of uncertainty in the input data with respect to the trained model (see major criticisms).

RESPONSE: Thanks for your feedback. We have revised the sentence to explain the reason (i.e., approximation of MMSE of solution distribution) for the smoothness of the prediction. Please refer to Lines 140-141.

- Line 146: what does "should be used with caution" mean?

RESPONSE: Thank you for your comment. It means that the prediction might be inaccurate as a high variance exists among distinct outputs. We have revised the manuscript in Lines 155-156.

- Line 158 and Ext. Data Fig. 7: while the main text mentions the size of the one image used during

fine-tuning, the Ext. Data Fig. does not. The fact that one image is used is therefore a bit imprecise/misleading — was a single image of size 256x256 being used, results would likely be much worse, while an image of 4000x4000 would likely lead to better results. Why not talking about kilo- or megapixels being used for fine-tuning instead?

RESPONSE: We appreciate this suggestion. All the “one-pair” mentioned in the manuscript refer to a single data pair of 1768×1768 pixels from new domains. We have revised the manuscript and figure legends accordingly and used megapixels as the unit of measurement for fine-tuning. Please refer to Lines 22, 169, 200, 364, 1366, 1395.

- Line 679: it appears that the Noise2Void results were obtained using a codebase that is not the official Noise2Void code. This is not ideal since a diligent reader would have to check if the implementation actually performs equally well as the original Noise2Void implementation.

RESPONSE: Thank you for pointing it out. We have re-trained the Noise2Void model using an official codebase (<https://github.com/juglab/n2v>), which achieves similar results. We have updated the Noise2Void results in Extended Data Figure 5 and revised the manuscript in Lines 725-728.

- Line 711: This formula is not 100% in-line with what I believe is the default definition for PSNR and is additionally using confusing nomenclature. What are the parameters of PSNR or MSE (which I assume are functions of something). What are m,n, I, and R? Additionally, most image restoration publications actually use an intensity shifted PSNR, i.e. allowing the predicted image intensities (R?) to be shifted by a constant in order to minimize the computed PSNR value. The reasoning is to evaluate the predicted structure, not some systematic intensity shift throughout the image.

RESPONSE: Thanks for your comments regarding the PSNR formula. We have modified the formula of PSNR and added an explanation of the notion used in the formula. I refers to the generated image and R refers to the ground truth. m and n are the width and height of the image. Please refer to Lines 753-768. In practice, we calculated the PSNR through the function in the `scikit-image` package (https://scikit-image.org/docs/stable/api/skimage.metrics.html#skimage.metrics.peak_signal_noise_ratio), a popular image processing package.

- Line 719: PSNR does not “favor” blurry images (see major criticisms).

RESPONSE: Thank you for your feedback. As previously stated, we have adjusted the sentence to prevent any confusion. Please see Lines 753-768.

- Line 797: the current manuscript never distinguishes Data and Model Uncertainty (also known as aleatoric and epistemic uncertainty). What is the uncertainty in this manuscript measuring?

RESPONSE: Thank you for your pointing it out. The current uncertainty in the manuscript measures model uncertainty (epistemic uncertainty). We have revised the manuscript Line 856 and 858 to declare it.

- Line 803: are the intensities X_0^k normalized? If so, how? If not, the constant threshold of 0.12 suggested to be used would have to be adjusted when the same image with 100-fold scaled pixel intensities was to be used.

RESPONSE: Yes. The image is min-max normalized on each local patch. We have supplemented normalization details in uncertainty estimation. Please refer to Lines 860-862.

- Line 804: x_0 is not introduced - is this the GT pixel intensity?

RESPONSE: x_0 refers to the average of outputs introduced in the previous formula. We have now explained it in the formula. Please refer to Line 866.

- Line 811: “density” appears to be the wrong word. Should it be “intensity”?

RESPONSE: Thank you for pointing out the typo. We have corrected it in the manuscript.

- Line 813: Why 1%? This appears to be such a random choice and might be entirely wrong for other datasets than the ones used by the authors for the sake of the given manuscript.

RESPONSE: Thanks for your comments. As we mentioned above, the 1% is an empirically determined hyper-parameter based on the study of our dataset. 1% was designed to ignore unreasonably high uncertainty pixels (outliers and pixels in the structure boundary, Extended Data Fig. 6, see figure above) of the uncertainty map to better align with the human judgment of reliability. For applications with a similar dataset, 1% ignored pixels should generally work.

However, due to the wide-range applications of electron microscopy, the number of disregarded pixels may vary across different datasets and can be modified by users through examination of the uncertainty map and estimation of the unreasonably high number of uncertain pixels. We have revised the manuscript accordingly. Please refer to Lines 874-880.

- Line 822: I cannot see the “transition valley” the authors point to in Fig. 1c.

RESPONSE: Thanks for your question. “Transition valley” refers to the area between the low LPIPS region and high LPIPS region (see figure below where an arrow points out it).

- Line 1254 (Ext. Data Fig. 4): I’m having a hard time seeing what “human perception” should want me to see here. Also, when talking about scientific image data quantification, maybe human perception is not what we want to optimize for (see major criticisms). The computed mean image of multiple predictions showing higher and higher PSNR values is testament of the major criticism I voiced at the very beginning. The mean of m of any predictions converges to the MMSE of the distribution of possible predictions and is therefore the minimizer of an L2 loss and of PSNR. The fact that a single prediction has an higher PSNR also means that the high frequency components of the predicted images (outer parts in a Fourier spectrum of that prediction) are not in-line with the GT, and the MMSE is smooth because of the uncertainty in the given input data. Hence, this result is not undesired but highly informative.

- Line 1260: PSNR does not prefer over-smoothed results. Try to take any prediction and smooth it with some small gaussian kernel. PSNR will be MUCH worse instantly!

RESPONSE: Thanks for your feedback. We have updated the figure legend to remove the statements about PSNR and clarify that the average approximates the MMSE of solution distribution and leads to a higher PSNR score. Please refer to Lines 1320-1332.

- Ext. Data Figure 6 (and likely some other places): it would be very informative to explicitly show the 1% discarded uncertainty pixels. Currently the fact that the 1% highest uncertainty pixels are discarded is buried in the methods and not every reader will pick this up. It is an important fact though, because users will have to adjust this value if their data would contain more or less strong image gradients, wouldn't they?

RESPONSE: Thanks for your suggestion. As we mentioned above, we have shown the 1% ignored pixels in the Extended Data Fig. 6. 1% is an empirically determined parameter designed to ignore the outliers and pixels in the structure boundary, which does not necessarily mean incorrect predictions, after studying the uncertainty map of our dataset. The number of ignored pixels can be varied with different datasets and adjusted by users by inspecting the uncertainty map and estimating the unreasonably high uncertainty pixels number. We have revised Lines 357-360, 874-880, 1349-1356 in the manuscript.

- Ext. Data Figure 7: as mentioned before, speaking about "one training pair" is not as informative as pointing at the total number of pixels available during fine-tuning. Also, if the new domain is more or less different from the one the network is trained on, the amount of pixels will needed will naturally vary. While it is too much to ask to quantify this, the authors could at least acknowledge this fact by spelling it out.

RESPONSE: Thanks for your comment. As mentioned above, we have added pixel number information for fine-tuning in the manuscript and figure legends. We have also acknowledged that the number of pixels or images used for fine-tuning the model will vary depending on how different the target dataset is from the original dataset. Please refer to Lines 367-369.

Reviewer #4 (Remarks on code availability):

- Code should be linked from the main text.

RESPONSE: Thanks for your comment. We have linked the code in the manuscript. Please refer to Lines 95-96.

- Tutorials (e.g. example notebooks that reproduce results from the paper) would be highly recommended and useful.

RESPONSE: Thanks for your comment. As mentioned above, we have developed and included example notebooks that are designed to be user-friendly and can be executed directly on Google Colab or a local computer server (<https://github.com/Luchixiang/EMDiffuse/tree/master/example>). These notebooks provide step-by-step instructions and pre-configured settings to facilitate the use of EMDiffuse.

- Uploading trained models in one way or another would be very useful and highly recommended. Bioimage.io would be ideal since it would allow future users to test the model in their browser and re-use them in other scientific image analysis tools.

RESPONSE: Thanks for your comment. As mentioned above, the current framework of bioimage-zoo presents specific challenges for generative models like EMDiffuse that generate outputs with inherent variability. We will keep contacted with bioimage-zoo and uploaded EMDiffuse when

bioimage-zoo supports generative models. We have made the model weights used to create figure public available (https://connecthkuhk-my.sharepoint.com/:f:/g/personal/u3590540_connect_hku_hk/EtSvqrIyrNREim5dJfabx2ABMLNhwk2Z9EsJDD4w6mls8g?e=OdP4Vq).

REVIEWERS' COMMENTS

Reviewer #1 (Remarks to the Author):

Thanks for addressing the technical issues. I suggest that the publisher provide serious editorial help to improve the still difficult to follow writing of the main manuscript. The Methods chapter is excellently written and easy to follow.

Code and data are currently released without a license, I suggest BSD/ MIT/ Apache for code and something like CC-BY for data.

Me and other reviewers suggested to better explain that the diffusion based approach generates samples from a distribution of plausible solutions to the ultimately ill-posed super-resolution or denoising problem. The consistency of these solutions can ultimately be evaluated with respect to specific analysis tasks like structural segmentation or classification (Figs 3 and 4). The added sections improve this compared with earlier versions of the manuscript. I still do not find that averaging 2 (or 6) of those samples to tune FSIM performance (113-123, Fig 2, Ext Data Fig 3 and 4) is scientifically meaningful, but leave this to the editor and the authors for consideration.

Reviewer #3 (Remarks to the Author):

Thank you for revising manuscript. I find that you have meticulously incorporated the reviewer's feedback into the revised manuscript. Consequently, I find the manuscript to be suitable for publication in Nature Communications.

The findings presented in this study hold significant promise for advancing the field of volume electron microscopy. Nevertheless, there exists a potential risk of generating misleading images if incorrect images are utilized for training or if the z-step range is inaccurate. Please continue to contribute so that this technique can be used properly after the paper is published.

Reviewer #4 (Remarks to the Author):

The re-review of the manuscript describing EMDiffuse reached me at a very inconvenient time. I apologize very much for not having had enough time to go over all the edits with the same rigor as I would usually do.

Since there are three more reviewers assigned to this submission and the rebuttal states changes that are largely in line with my suggestions from previous rounds of reviews, I feel rather confident that the current manuscript has addressed my feedback sufficiently well. I still think the confidence score might be misunderstood or misused in some cases, but I have made my point and I have full trust in the editorial team to judge the adequacy of the final manuscript.

Reviewer #4 (Remarks on code availability):

I have reviewed the code the last time around but not this time.

Adding notebooks as tutorials is a great addition and I hope the other reviewers found the time to have an in-depth look at them.

From what I saw the last time, my opinion is very much in-line with the other reviewers.